# Precision targeting of β-catenin induces tumor reprogramming and immunity in hepatocellular cancers

First-line immune checkpoint inhibitor (ICI) combinations show responses in subsets of hepatocellular carcinoma (HCC) patients. Nearly half of HCCs are Wnt-active with mutations in *CTNNB1* (encoding for β-catenin), *AXIN1/2*, or *APC*, and demonstrate heterogeneous and limited benefit to ICI due to an immune excluded tumor microenvironment. We show significant tumor responses in multiple β-catenin-mutated immunocompetent HCC models to a novel siRNA encapsulated in lipid nanoparticle targeting *CTNNB1* (LNP-CTNNB1). Both single-cell and spatial transcriptomics reveal cellular and zonal reprogramming, along with activation of immune regulatory transcription factors IRF2 and POU2F1, re-engaged type I/II interferon signaling, and alterations in both innate and adaptive immunity upon β-catenin suppression with LNP-CTNNB1 at early- and advanced-stage disease. Moreover, ICI enhances response to LNP-CTNNB1 in advanced-stage disease by preventing T cell exhaustion and through formation of lymphoid aggregates (LA). In fact, expression of an LA-like gene signature prognosticates survival for patients receiving atezolizumab plus bevacizumab in the IMbrave150 phase III trial and inversely correlates with *CTNNB1*-mutatational status in this patient cohort. In conclusion, LNP-CTNNB1 is efficacious as monotherapy and in combination with ICI in *CTNNB1*-mutated HCCs through impacting tumor cell-intrinsic signaling and remodeling global immune surveillance, providing rationale for clinical investigations.

Hepatocellular carcinoma (HCC) is the third leading cause of cancer-related death globally[1]. Despite the shift in therapeutic management of advanced disease over the last five years from multi-tyrosine kinase inhibitors (TKIs) (e.g., sorafenib) to immunotherapy with immune-checkpoint inhibitor (ICI) combinations (e.g., atezolizumab plus bevacizumab), objective response rates (ORRs) remain low at ~30% with overall survival <2 years[2–5]. Preclinical and clinical studies investigating molecular correlates of ICI response have yielded novel insights into potential mechanisms of resistance, including but not limited to immune exclusion, with Wnt/β-catenin activation contributing to this phenotype[6–8]. Wnt/β-catenin pathway activity is observed in up to 50% of tumors from patients with HCC, with mutations mostly occurring in

*CTNNB1* (26–37%), *AXIN1/2* (8–10%), and *APC* (3–5%)[9–12]. Gain-of-function (GOF) mutations in *CTNNB1* (encoding for β-catenin) are one of the major trunk mutational events in HCC and occur mostly as missense mutations in exon 3 at serine and threonine residues or the ubiquitination destruction motif, which interfere with its degradation, leading to constitutive β-catenin activation and target gene transcription[13,14]. Patients with *CTNNB1*-mutated HCC have upregulation of known Wnt/β-catenin target genes, including *GLUL, AXIN2, LGR5*, and *TBX3*[11]. In fact, glutamine synthetase (GS; encoded by GLUL) immunohistochemistry is used as a biomarker for patients with CTNNB1-mutated HCC[15]. However, targeting these downstream Wnt target genes has revealed novel negative feedback loops in the

✉e-mail: junyantao2010@gmail.com; smonga@pitt.edu

Wnt/β-catenin oncogenic circuit[16,17], necessitating the need to focus on targeting β-catenin directly for precision therapy.

Despite improved molecular stratification of HCC over the last decade, with recognition of Wnt/β-catenin-driven tumors overlapping with Hoshida S3[18] or Boyault G5/G6 subclasses[19], these different molecular stratifications have not yielded prognostic implications due to a lack of clinically approved targeted or biomarker-driven precision therapeutics. β-catenin has traditionally been an undruggable target, despite preclinical studies elucidating the molecular and metabolic addiction to β-catenin oncogenic signaling in *CTNNB1*-mutated HCC[20–23]. Thus, β-catenin is a prime target for precision therapy. Advances in RNAi technology over the last two decades have resulted in multiple approved RNAi therapies[24], and RNAi-mediated gene silencing has proven to be an excellent tool for targeting the traditionally undruggable, especially in hepatic tissue[25].

In this work, we investigate the relevance of RNAi-mediated β-catenin inhibition in patient-derived *CTNNB1*-mutated HCC organoids and multiple humanized mouse models of *CTNNB1*-mutated HCC at different treatment windows and elucidate the underlying mechanisms of response in both hepatic and immune compartments through both single-cell and spatial transcriptomic approaches. Our findings provide a mechanistic basis for clinical investigations of this RNAi therapeutic targeting β-catenin for HCC treatment as an innovative treatment paradigm in the form of monotherapy and/or in combination with ICI in human subjects belonging to the Wnt-β-catenin active-HCC subclass.

## Results

### RNAi-mediated β-catenin inhibition results in potent CTNNB1 knockdown in vitro and in vivo

To study the effects of RNAi-mediated inhibition in β-catenin-mutated HCC, we utilized a novel siRNA that targets the *CTNNB1* gene, with both mouse and human specificity, encapsulated in a lipid nanoparticle (referred hereafter as LNP-CTNNB1). We first assessed whether LNP-CTNNB1 affected growth in a patient-derived HCC organoid (23277) with known mutation in *CTNNB1*[26]. After 48- and 72-h treatment with LNP-CTNNB1 at 20 nM concentration, we observed a significant decrease in both the numbers and the size of the organoids compared to treatment with the LNP-CTRL (Fig. 1a, b). Thus, LNP-CTNNB1 demonstrates efficacy in mutant-*CTNNB1* human HCC organoid cultures.

Next, to assess its pharmacodynamic effects, we first delivered LNP-CTNNB1 via tail vein intravenous (I.V.) injection to mouse livers which were transfected with human S45Y-mutant-*CTNNB1* gene (S45Y-*hCTNNB1* mice) via sleeping beauty-hydrodynamic tail vein injection (SB-HDTVi) system. We have previously reported that mouse hepatocytes overexpressing mutant-β-catenin alone via SB-HDTVi do not develop HCC[27], but require a secondary driver like hMet, mutant-Kras, or mutant-Nrf2, to induce HCC[20,27,28]. After 4 treatments at 3 mg/kg dosing in S45Y-*hCTNNB1* mice (Supplementary Fig. 1a), we observed an appreciable decrease in liver weight and liver weight to body weight ratio (LW/BW), which is consistent with the role of β-catenin in regulating liver growth and size (Supplementary Fig. 1b–d)[29,30]. Additionally, expression of two well-known β-catenin target genes GS and Cyclin D1 (CCND1) via immunohistochemistry (IHC) were absent throughout the liver lobule indicating high *mCTNNB1* gene knockdown (Supplementary Fig. 1e, f). Concomitantly, Myc-tag (present on the S45Y-*hCTNNB1* plasmid) positive cells were absent in the LNP-CTNNB1 treated mice compared to islands of Myc-tag positive cells in the LNP-CTRL mice (Supplementary Fig. 1e, f). Thus, LNP-CTNNB1 targets both endogenous mouse and mutant human *CTNNB1* with high potency and specificity in vivo.

Prior to testing efficacy of siRNA-mediated *CTNNB1* knockdown in HCC, we assessed whether there were any effects of the LNP itself on the tumor immune microenvironment (TIME). We treated mice injected with T41A-β-catenin-G31A-Nrf2 (β-N model) with either PBS, LNP-CTRL, or LNP-CTNNB1 (Supplementary Fig. 1g) utilizing a similar frequency and dosage scheme as in Supplementary Fig. 1a. β-N model has been previously shown to represent 9–12% of all human HCC[28]. Following treatment, we observed a decrease in liver weight and LW/BW in LNP-CTNNB1 treated mice (Supplementary Fig. 1h, i) with no differences in liver serum biochemistries (Supplementary Fig. 1j). Next, we performed bulk RNA-sequencing across all 3 treatment groups, and observed that PBS and LNP-CTRL treated animals are transcriptionally very similar, and distinct from the LNP-CTNNB1 treated group (Supplementary Fig. 1k). Additionally, gene set variation analysis using gene ontology (GO) pathways demonstrated that the immune phenotype is similar between PBS and LNP-CTRL treated mice, suggesting that the LNPs themselves do not influence the TIME in the CTNNB1-mutated HCC (Supplementary Fig. 1l).

### RNAi-mediated β-catenin inhibition impairs tumor growth in multiple immunocompetent CTNNB1-mutated and non-CTNNB1-mutated HCC mouse models with durable response in early-stage disease setting

We next evaluated the in vivo efficacy of LNP-CTNNB1 in *CTNNB1*-mutated and non-mutated HCC models. We first performed a dose titration study to determine the lowest efficacious dose in the β-N model. We administered once weekly I.V. injections at 3, 1, 0.3, 0.1, and 0.03 mg/kg dosages over 6 weeks of LNP-CTNNB1 starting at 5-weeks post-HDTVi, which we previously determined as a timepoint when microscopic tumor foci are already established[28] (Supplementary Fig. 2a). There were significant reductions in the tumor burden across a wide dose range of the LNP-CTNNB1 (3, 1, 0.3, and 0.1 mg/kg), as evident grossly, by liver weight, and by LW/BW (Supplementary Fig. 2b–d, Fig. 1c–f). However, at 3 mg/kg dosage, following the 4th dose, we observed mortality in one of four mice, which was likely due to the high LNP dose and frequency. Additionally, the 0.3, 0.1, and 0.03 mg/kg LNP-CTNNB1 dosages resulted in partial responses, with remnant microscopic tumor foci observed in 0.3 and 0.1 mg/kg treated animals (Supplementary Fig. 2e) and macroscopic tumor nodules present in animals treated with 0.03 mg/kg (Supplementary Fig. 2b, e). At the 1 mg/kg LNP-CTNNB1 dosage there was no morbidity or mortality observed in these mice. Additionally, there were no gross phenotypic changes to other organs including lungs, spleen, intestine, and heart. H&E of the spleens across a wide dose range did not demonstrate any microscopic changes (Supplementary Fig. 2f). We did not observe any gross neurological, gastrointestinal, genitourinary, cardiovascular, or respiratory deficits and/or distress following the once weekly treatments over 6 weeks, similar to what has been noted in rodents previously across a broad dose range, along with absence of any adverse signs or toxicity including any alterations in body weight or liver function tests[31]. Significant tumor responses were observed at 1 mg/kg LNP-CTNNB1 dosage as noted via H&E, IHC for Myc-tag and GS/Ki67, and magnetic resonance imaging (MRI) (Fig. 1g; Supplementary Fig. 3a–d). As a result, we utilized the 1 mg/kg LNP-CTNNB1 dose for treatment of β-catenin-mutated HCC preclinical models.

To extrapolate our findings to additional β-catenin-mutated HCC preclinical models that we have previously reported, we next tested LNP-CTNNB1 in the S45Y-mutant-β-catenin-Met (β-M) model, which represents 11% of human HCC[27]. Treatment was initiated at 3-weeks post-HDTVi when microscopic tumor foci are already established and based on the more aggressive tumor phenotype in this model, as determined by us previously[27]. Once weekly I.V. administration at 1 mg/kg LNP-CTNNB1 dosage over 6 weeks led to a significant decrease in tumor burden grossly (Fig. 1h–k), and histologically as observed via H&E, Myc-tag, and GS/Ki67 IHC (Fig. 1l; Supplementary Fig. 3e, f). Lastly, starting at 3-weeks post-HDTVi, a timepoint with known microscopic tumor burden[32], we tested 1 mg/kg LNP-CTNNB1 dosage

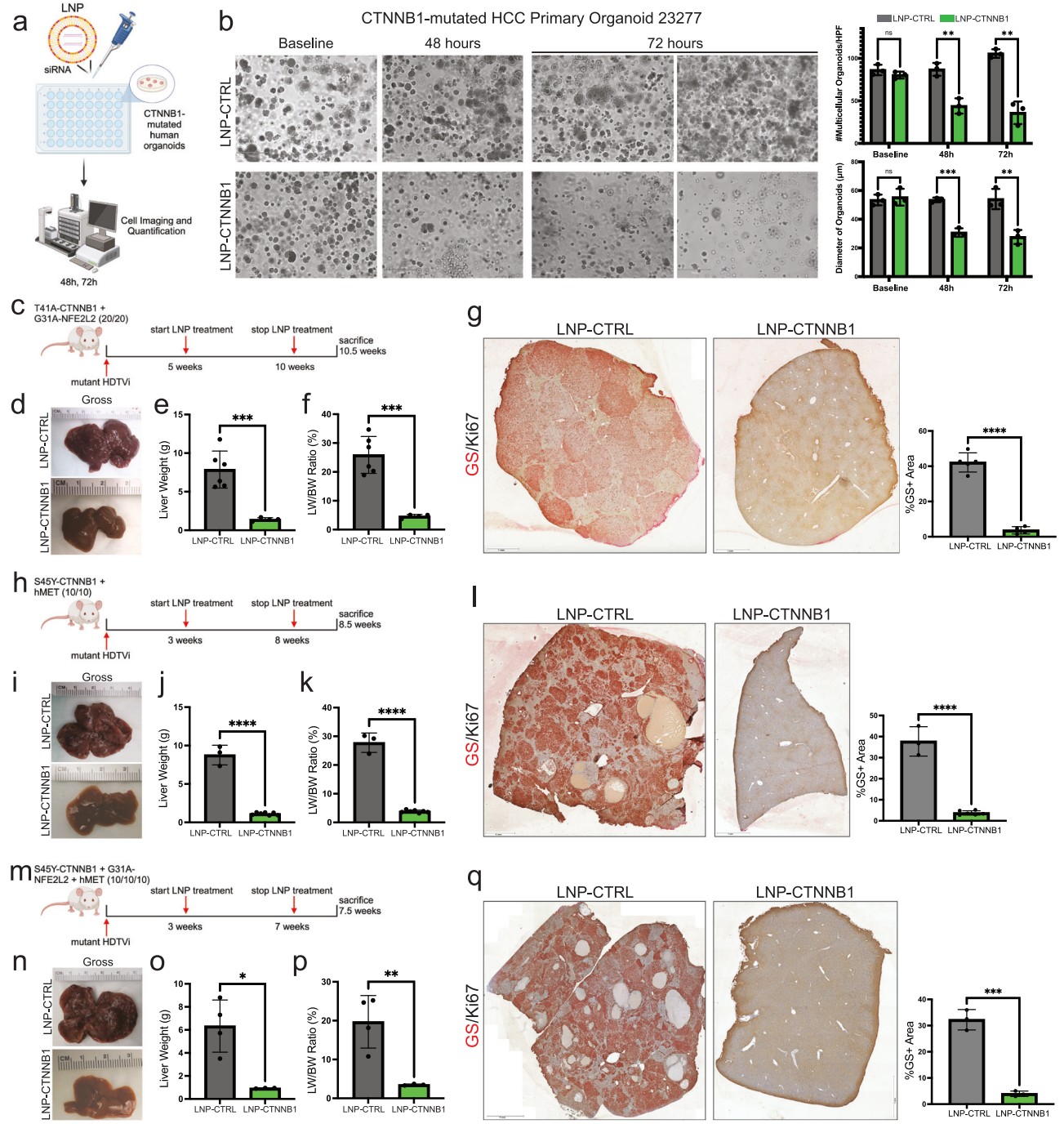

in a third *CTNNB1*-mutated model, the S45Y-mutant-β-catenin-Nrf2-Met (β-N-M) model, which represents another unique ~5% subset of human HCC[32]. Following a similar treatment protocol, we again observed significant tumor responses grossly and microscopically (Fig. 1m–q; Supplementary Fig. 3g, h).

Next, we wanted to assess response to LNP-CTNNB1 in models that were not *CTNNB1*-mutated, due to the general mitogenic function of Wnt/β-catenin signaling pathway in the liver[33]. β-Catenin suppression by LNP-CTNNB1 in the Nrf2-hMet (N-M) model at 8-weeks post-HDTVi, a timepoint with known microscopic tumor burden[32], led to a decrease in liver weight and LW/BW and in macroscopic disease (Supplementary Fig. 4a–d), although there was persistence of some microscopic nodules, which depicted inferiority in response when compared to mutant-β-catenin-driven tumors (Supplementary Fig. 4e–g). This decrease in tumor burden was observed despite HCC

nodules in this model not being homogenously positive for the bonafide Wnt target GS (Supplementary Fig. 4e). We have also previously reported that c-Met/sgAxin1 tumors require intact β-catenin to initiate tumorigenesis[34]. We also tested dependence on β-catenin in another independent non-*CTNNB1*-mutated HCC model using a genetic approach (Supplementary Fig. 4h). β-Catenin deletion in SB-HDTVi induced Akt-NRas HCC in β-catenin floxed mice through simultaneous administration of pCMV-cre or control led to a significant improvement in overall survival and less tumor burden in pCMV-Cre compared to control, although tumors still persisted (Supplementary Fig. 4i). Thus, overall, we observed that β-catenin inhibition alone for *CTNNB1*-mutated HCC is most effective in early-stage disease setting as evident through significant tumor responses in multiple models of *CTNNB1*-mutated HCC, and as partial responses in β-catenin non-mutated HCC models.

**Fig. 1 | RNAi-mediated β-catenin inhibition is efficacious in multiple immuno-competent *CTNNB1*-mutated HCC mouse models in early-stage disease setting. a** Schematic of *CTNNB1*-mutated patient-derived HCC organoid LNP treatment (20 nM). **b** (Left) Brightfield images at baseline, 48-, and 72-h. Scale bar represents 200 μm. (Right) Quantification of number of multi-cellular organoids per high powered field (HPF) and organoid diameter (*n* = 3 biological replicates per treatment and time point). **c** LNP treatment scheme in β-catenin-Nrf2 (β-N) model. Mice received once weekly intravenous (I.V.) injections at 1 mg/kg dosage starting at 5-weeks post-hydrodynamic tail vein injection (HDTVi). **d** Representative gross liver images of LNP-CTRL and LNP-CTNNB1 (1 mg/kg) treated β-N animals at 10.5-week timepoint. **e, f** Liver weights (*p* = 0.0008) and liver weight/body weight (LW/BW) (*p* = 0.0002) comparing LNP-CTRL (*n* = 6) and LNP-CTNNB1 (*n* = 4; 1 mg/kg) treated β-N animals. **g** (Left) Representative images of immunohistochemistry (IHC) for glutamine synthetase (GS)/Ki67 co-stain comparing LNP-CTRL and LNP-CTNNB1 (1 mg/kg) treated β-N animals. (Right) Quantification of %GS+ area comparing LNP-CTRL (*n* = 5) and LNP-CTNNB1 (*n* = 4). **h** LNP treatment scheme in β-catenin-hMet (β-M) model. Mice received once weekly I.V. injections at 1 mg/kg dosage starting at 3-weeks post-HDTVi. **i** Representative gross liver images of LNP-CTRL and LNP-CTNNB1 (1 mg/kg) treated β-M animals at 8.5-week timepoint. **j, k** Liver weights and LW/BW comparing LNP-CTRL (*n* = 3) and LNP-CTNNB1 (*n* = 7; 1 mg/kg) treated β-M animals. **l** (Left) Representative images of IHC for GS/Ki67 co-stain comparing LNP-CTRL and LNP-CTNNB1 (1 mg/kg) treated β-M animals. (Right) Quantification of % GS+ area comparing LNP-CTRL (*n* = 3) and LNP-CTNNB1 (*n* = 6). **m** LNP treatment scheme in β-catenin-Nrf2-hMet (β-N-M) model. Mice received once weekly I.V. injections at 1 mg/kg dosage starting at 3-weeks post-HDTVi. **n** Representative gross liver images of LNP-CTRL and LNP-CTNNB1 (1 mg/kg) treated β-N-M animals at 7.5-week timepoint. **o, p** Liver weights (*p* = 0.01) and LW/BW (*p* = 0.0098) comparing LNP-CTRL (*n* = 4) and LNP-CTNNB1 (*n* = 3; 1 mg/kg) treated β-N-M animals. **q** (Left) Representative images of IHC for GS/Ki67 co-stain comparing LNP-CTRL and LNP-CTNNB1 (1 mg/kg) treated β-N-M animals. (Right) Quantification of %GS+ area comparing LNP-CTRL (*n* = 3) and LNP-CTNNB1 (*n* = 3) (*p* = 0.0003). Created in BioRender. Lehrich (2025) https://BioRender.com/smaki7g. Lehrich (2025) https://BioRender.com/858w237. Lehrich (2025) https://BioRender.com/6ce7tob. Lehrich (2025) https://BioRender.com/9127h6y. For (**b**), (**e**–**g**), (**j**–**l**), and (**o**–**q**), data presented as mean values ± standard deviation (SD) and *P*-values calculated by unpaired two-tailed Student's *t*-test. Source data are provided as a Source Data File. For (**g**), (**l**), (**q**), scale bar indicates magnification. **p* < 0.05, ***p* < 0.01, ****p* < 0.001, *****p* < 0.0001.

Lastly, we assessed the long-term durability of the significant tumor responses and overall survival (OS) in both the β-N and β-M models following LNP-CTNNB1 treatment at 1 mg/kg dosage. Following the same treatment protocol in β-N (Fig. 1c) and β-M (Fig. 1h) models, we then withdrew LNP-CTNNB1. In the β-N model, following treatment cessation, mice were moribund by ~22.5-weeks post-LNP-CTNNB1 treatment, with gross tumor burden becoming equivalent to the tumor burden observed in LNP-CTRL treated mice at ~10.5 weeks, which is a lethal timepoint in β-N model (Supplementary Fig. 5a, b). Thus, with LNP-CTNNB1 treatment in β-N model, OS was significantly extended by ~12 weeks (*p* < 0.01) (Supplementary Fig. 5c). The nodules that reappeared at the ~22.5-week timepoint were positive for both GS and Nqo1 (Nrf2-target) (Supplementary Fig. 5d). Similarly, in the β-M model, following treatment cessation, mice were moribund by ~16.5-weeks post-LNP-CTNNB1 treatment, with gross tumor burden becoming equivalent to the tumor burden observed in LNP-CTRL treated mice at ~7.5 weeks, which is a lethal timepoint in β-M model (Supplementary Fig. 5e, f). Thus, LNP-CTNNB1 treatment in the β-M model extended OS by ~9 weeks (*p* < 0.001) (Supplementary Fig. 5g). The nodules that reappeared at ~16.5-week timepoint in β-M model were also positive for GS and V5-tag (present on hMet plasmid) (Supplementary Fig. 5h). Overall, LNP-CTNNB1 treatment as monotherapy more than doubled the OS of mice in both HCC models although tumors recurred after treatment cessation. These recurring tumors appear to be mutant-β-catenin-driven and not due to appearance of any de novo resistant clones.

**Earliest biological response to RNAi-mediated β-catenin Inhibition observed at 3-days following initial LNP-CTNNB1 treatment**
Given the robust tumor responses following LNP-CTNNB1 treatment, we investigated the earliest biological response observed following β-catenin knockdown within the tumor cells. In the β-N model, we followed mice over a 3-week treatment course (LNP-CTNNB1 injected weekly ×3) and sacrificed mice at 1-, 3-, 5-, 7-, 14-, and 21-days post the first LNP treatment (Fig. 2a). Over this 21-day treatment time course, the visible tumor foci or LW/BW progressively trended lower in the LNP-CTNNB1 group although differences were insignificant (except day 5) when compared to time-matched LNP-CTRL group (Supplementary Fig. 6a; Fig. 2b). However, at 3-days after a single LNP-CTNNB1 dose, RNA expression of *Ctnnb1*, along with Wnt target genes, *Glul*, *Ccnd1, Lect2*, and *Rgn* were significantly decreased in LNP-CTNNB1 compared to LNP-CTRL mice (Fig. 2c). Additionally, GS protein visualized via IHC was decreased in tumor nodules but retained in pericentral hepatocytes at the 3-day timepoint, while it was absent also in the pericentral hepatocytes by 14-days in the LNP-CTNNB1 group (Fig. 2d; Supplementary Fig. 6b). Ki67 and TUNEL IHC also demonstrated significantly decreased tumor cell proliferation and trend toward increased cell death, respectively, at the 3-day timepoint, which was not observed at the 1-day timepoint (Fig. 2e, f; Supplementary Fig. 6c, d). Given these results, we administered a single LNP treatment to β-M animals and sacrificed mice at 3-days post-treatment (Supplementary Fig. 7a). While there was no significant difference in gross tumor burden (Supplementary Fig. 7b), a single dose of LNP-CTNNB1 significantly decreased liver weight and LW/BW (Supplementary Fig. 7c–e), decreased intra-tumoral GS expression but retained V5-tag expression (Supplementary Fig. 7f, g). Also, there were significantly less intra-tumoral Ki67-positive cells and significantly more TUNEL-positive cells (Supplementary Fig. 7h, i). Thus, the earliest evident biological response following RNAi-mediated β-catenin inhibition in both models occurred at 3-days post-LNP treatment.

To understand the transcriptional consequences of β-catenin knockdown in HCC, we performed bulk RNA-sequencing (RNA-seq) on both the β-N and β-M models treated with either LNP-CTRL or LNP-CTNNB1 at the 3-day timepoint. Each model clustered distinctly with LNP-CTNNB1 groups for each model clustering independently from the LNP-CTRL groups as shown via PCA analysis (Fig. 2g). Differential gene expression analysis comparing LNP-CTRL vs LNP-CTNNB1 demonstrated 455 upregulated and 628 downregulated genes in the β-N model, and 608 upregulated and 634 downregulated genes in the β-M model, with 230 common downregulated and 73 common upregulated genes (Fig. 2h, i). Common downregulated genes included Wnt/β-catenin target genes and pericentral hepatocyte markers (e.g., *Glul, Axin2, Lgr5, Notum, Lect2, Ccnd1, Cyp2e1, Cyp1a2,* and *Oat*), and common upregulated genes were midzonal and periportal hepatocyte markers (e.g., *Hamp2, Cyp8b1,* and *Cyp2f2*) (Fig. 2j). From both models, gene set enrichment analysis (GSEA) using Kyoto Encyclopedia of Genes and Genomes (KEGG) pathways demonstrated relative positive enrichment of metabolic and tumor microenvironment pathways, along with relative negative enrichment of cell cycle, Wnt signaling, and xenobiotic metabolism pathways (Fig. 2k, l). Thus, we inferred β-catenin mutations in HCC confer most profound effects on tumor cell growth/proliferation, tumor metabolism, and tumor microenvironment.

**Integrated single-cell analyses reveal de novo formation of reprogrammed hepatocytes within remnant tumor nodules**
To further interrogate tumor cell-intrinsic biological effects that occurred at the 3-day timepoint, we administered LNP-CTRL or

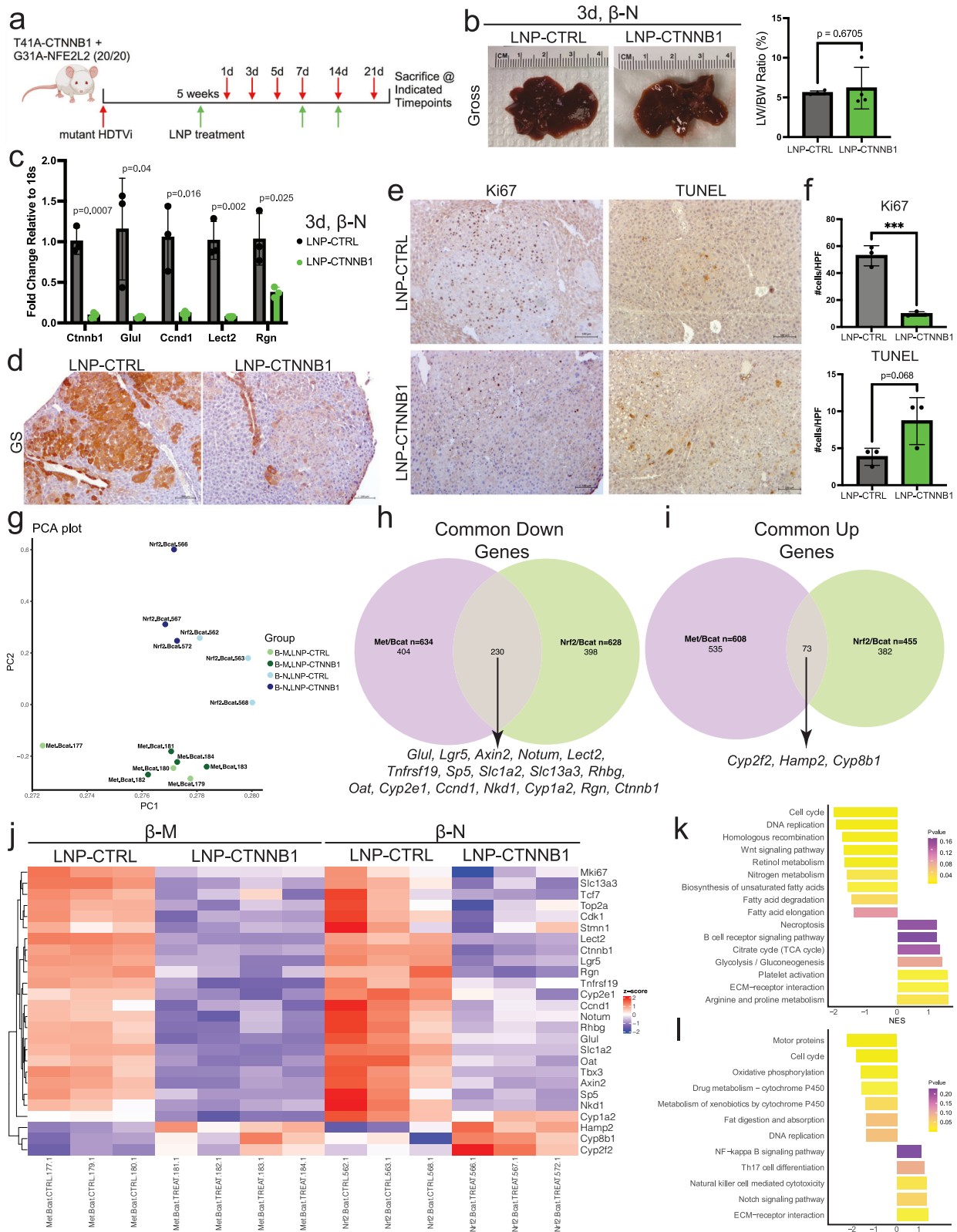

LNP-CTNNB1 at 5-weeks post-HDTVi to β-N model mice and performed single-cell RNA-sequencing (scRNA-seq) analysis on a hepatocyte-enriched single-cell population following whole liver perfusion. In total, 94,650 single cells were sequenced with 26,851 in the LNP-CTRL group and 67,799 in the LNP-CTNNB1 group. Unbiased clustering on the integrated dataset resulted in 10 unique cell populations (Supplementary Fig. 8a), annotated as (a) Dying/injured hepatocytes, (b),

Hepatic stellate cells, (c) Kupffer cells, (d) Erythroid cells, (e) Endothelial cells, (f) Low-quality hepatocytes, (g) Reprogrammed hepatocytes (expressing both zone 1 & 2 markers *Arg1, Ass1, Pck1, Hal, Hamp2*, with Nrf2 tumor targets *Prdx2, Prdx5, Gstm1, Gpx1*), (h) Zone 1 CTNNB1 WT (GS−) hepatocytes, (i) Zone 1/2 CTNNB1 MUT (GS+) hepatocytes, and (j) Zone 3 CTNNB1 WT & MUT (GS+) hepatocytes based on differential gene expression analysis per cluster (Supplementary

**Fig. 2 | Earliest biological response to RNAi-mediated β-catenin inhibition is observed 3-days following LNP treatment. a** LNP treatment scheme in β-catenin-Nrf2 (β-N) model. **b** (Left) Representative gross liver images and (Right) liver weight/body weight (LW/BW) comparing LNP-CTRL (*n* = 4) and LNP-CTNNB1 (*n* = 4; 1 mg/kg) β-N treated animals 3-days post 1st LNP treatment. **c** qPCR RNA expression levels of *Ctnnb1* and β-catenin target genes (*Glul, Ccnd1, Lect2, Rgn*) between LNP-CTRL (*n* = 3) and LNP-CTNNB1 (*n* = 3) β-N treated animals. **d, e** Representative immunohistochemistry (IHC) images for glutamine synthetase (GS), Ki67, and TUNEL comparing LNP-CTRL and LNP-CTNNB1 treated β-N animals 3-days post 1st LNP treatment. Scale bar represents 100 μm. **f** Quantification of number of Ki67- and TUNEL-positive cells across multiple high-power fields (HPF) between LNP-CTRL (*n* = 3) and LNP-CTNNB1 (*n* = 3) treated β-N animals 3-days post 1st LNP treatment (*p* = 0.0006). **g** Principal component analysis of bulk RNA-sequencing transcriptomic profiles of β-N and β-M model treated with LNP-CTRL or LNP-CTNNB1 and harvested 3-days post-LNP treatment, using all genes (*n* = 3–4 per condition and model). **h** Venn diagram highlighting number of common down-regulated differentially expressed genes (DEGs) (*n* = 230) between β-N and β-M

models treated with LNP-CTRL or LNP-CTNNB1 3-days post-LNP treatment. **i** Venn diagram highlighting number of common upregulated DEGs (*n* = 73) between β-N and β-M models treated with LNP-CTRL or LNP-CTNNB1 3-days post-LNP treatment. DEGs defined by FDR = 0.05 and fold change >1.5. **j** Heatmap of selected common downregulated and upregulated genes demonstrating normalized *z*-score expression value in each model with each LNP treatment condition from (**h**) and (**i**). **k** Kyoto Encyclopedia of Genes and Genomes (KEGG) pathway gene set enrichment analysis (GSEA) in β-N model comparing LNP-CTRL and LNP-CTNNB1 treated animals. **l** KEGG pathway GSEA in β-M model comparing LNP-CTRL and LNP-CTNNB1 treated animals. For (**k**, **l**), normalized enrichment score (NES) was normalized from ES, which was calculated by Kolmogorov–Smirnov-like statistic and the *p*-value was calculated using the one-tailed empirical permutation test procedure. Created in BioRender. Lehrich (2025) https://BioRender.com/kymtir0. For (**b**), (**c**), and (**f**), data are presented as mean values ± standard deviation (SD) and *P*-values calculated by unpaired two-tailed Student's *t*-test. Source data are provided as a Source Data File. *$p < 0.05$, **$p < 0.01$, ***$p < 0.001$, ****$p < 0.0001$.

Fig. 8b, c). KEGG pathway enrichment analysis comparing each cluster to all other clusters revealed that top pathways for Zone 3 CTNNB1 WT & MUT (GS+) hepatocytes were bile acid secretion, drug metabolism – cytochrome P450, and fatty acid metabolism, which are all known hallmarks of CTNNB1-mutated HCC (Supplementary Fig. 8d)[35]. Zone 1/2 CTNNB1 MUT (GS+) hepatocytes and Zone 1 CTNNB1 WT (GS−) hepatocytes were interestingly enriched for arginine biosynthesis and amino acid biosynthesis (Supplementary Fig. 8e, f), which are known metabolic hallmarks of zone 1 metabolism[35]. This KEGG pathway enrichment analysis reveals the metabolic heterogeneity of tumor cells along the portal-central axis.

Cell-type proportion analysis comparing LNP-CTRL and LNP-CTNNB1 demonstrated less Zone 3 CTNNB1 WT & MUT (GS+) hepatocytes along with de novo appearance of reprogrammed hepatocytes following LNP-CTNNB1 treatment (Fig. 3a, b). KEGG and GO pathway enrichment analysis on the reprogrammed hepatocytes demonstrated enrichment of pathways across all zones, including biosynthesis of cofactors (Zone 1), amino acid catabolism (Zone 1), arginine biosynthesis (Zone 1), glutamate metabolism (Zone 3), glycolysis/TCA cycle (Zone 3), along with fatty acid metabolism, a hallmark of *CTNNB1*-mutated hepatocellular cancers (Supplementary Fig. 9a–d). Cell cycle phase-specific gene expression analysis on hepatocyte clusters importantly demonstrated that tumor cells (both Zone 3 CTNNB1 WT & MUT [GS+] and Zone 1/2 CTNNB1 MUT [GS+] hepatocytes) were the most proliferative, while reprogrammed hepatocytes and Zone 1 CTNNB1 WT (GS−) hepatocytes were the least proliferative with proportionally fewer cells in S and G2M phases of the cell cycle (Fig. 3c). In fact, reprogrammed hepatocytes and Zone 1 CTNNB1 WT (GS−) were the two enriched hepatocyte populations following LNP-CTNNB1 treatment. Interestingly, Zone 1/2 CTNNB1 MUT (GS+) hepatocytes were the most proliferative tumor cell population, with the most cells in S and G2M cell cycle phases (Fig. 3c). We next performed pseudo-time analysis on all the hepatocyte populations in the dataset which demonstrated the intermediate cell state of the reprogrammed hepatocytes occurring along the trajectory of Zone 3 CTNNB1 WT & MUT (GS+) hepatocytes to Zone 1 CTNNB1 WT (GS−) hepatocytes (Fig. 3d). Thus, reprogrammed hepatocytes are an intermediate cell phenotype, reflecting tumor cell differentiation to normal hepatocyte-like cells and contributing to the rapid cell turnover observed following LNP-CTNNB1 treatment.

Next, to confirm the spatial identity of the reprogrammed hepatocyte population, we performed single-cell spatial transcriptomics using Molecular Cartography™ platform on tissue sections from the 3-day timepoint with LNP-CTRL or LNP-CTNNB1 treatment in the β-N model. The 100-gene panel consisted of markers specific for Wnt/β-catenin targets, metabolic zonation, and

nonparenchymal cell types (Supplementary Table 7). Following data pre-processing and automatic cell segmentation, in total, 19,301 single cells were sequenced from multiple regions of interest (ROIs) with 10,227 cells across 6 ROIs in LNP-CTRL group and 9074 cells across 5 ROIs in LNP-CTNNB1 group. Unbiased clustering on all 100 genes resulted in 9 unique cell populations, annotated as (a) H1: Zone 3 CTNNB1 MUT (GS+), (b) H2: Zone 3 Central Vein (CV) CTNNB1 WT (GS+), (c) H3: Zone 3 CTNNB1 WT (GS-negative), (d) H4: Zone 2–3 CTNNB1 WT (GS-negative), (e) H5: Zone 1 CTNNB1 WT (GS-negative), (f) H6: Reprogrammed hepatocytes, (g) Hepatic stellate cells (HSCs), (h) Immune cells, and (i) Endothelial cells (ECs), based on marker gene expression per cluster (Supplementary Fig. 10a–d). Clustering by treatment condition demonstrated enrichment of reprogrammed hepatocytes (46.3% vs 2.7%) with loss of H1: Zone 3 CTNNB1 MUT (GS + ) hepatocytes (1.2% vs 29.0%) in LNP-CTNNB1 group (Fig. 3e, f), similar to the scRNA-seq analysis (Fig. 3a, b). Spatial plots confirmed the tumoral origin of the H6 cluster representing the reprogrammed hepatocytes (Fig. 3g, h). In fact, spatial visualization and quantification of Wnt target genes in tumoral and non-tumoral regions, including both hepatocyte and nonparenchymal cell populatons, revealed that β-catenin-mutated tumor cells are defined by expression of bonafide Wnt targets *Glul, Tbx3, Axin2, Lgr5, Lect2*, and *Ccnd1* (Supplementary Fig. 11a–c) with their identity intimately linked to zone 3 metabolic genes (and processes), including *Cyp2e1, Cyp1a2*, and *Oat*, with exclusion of zone 1 metabolic genes (and processes), including *Cyp2f2, Ass1*, and *Arg1* (Supplementary Fig. 12a, b). However, with LNP-CTNNB1 treatment, tumor cells begin to express *Cyp2f2, Arg1, and Ass1* along with diminished expression of *Cyp2e1, Cyp1a2, Oat* and others (Supplementary Fig. 12a, b). IHC validated the sc-Spatial transcriptomic findings and confirmed decreased expression of zone 3 markers CYP2E1 and OAT, with increased expression of zone 1 markers ARG1 and CYP2F2 (Supplementary Fig. 12c). Additionally, pseudotime analysis on the sc-Spatial transcriptomic data confirmed the intermediary phenotype of the H6: reprogrammed hepatocytes (Fig. 3i), similar to the scRNA-seq data (Fig. 3d). Lastly, for validation, cell cluster quantification was performed within tumoral and non-tumoral regions (using *Glul* as tumoral landmark) (Supplementary Fig. 13a, b), which revealed a significant decrease in cell density of clusters with active β-catenin signaling, and significant increase in cell density of the reprogrammed hepaotcytes, which occurred mostly in tumoral regions following LNP-CTNNB1 treatment (Supplementary Fig. 13c). Overall, this integrated single-cell analysis revealed that β-catenin-mutated tumors are exclusively zone 3 and respond to β-catenin suppression by turning off expression of these genes while differentiating towards zone 1/2 hepatocyte-like cells, thus reprogramming their overall metabolic machinery.

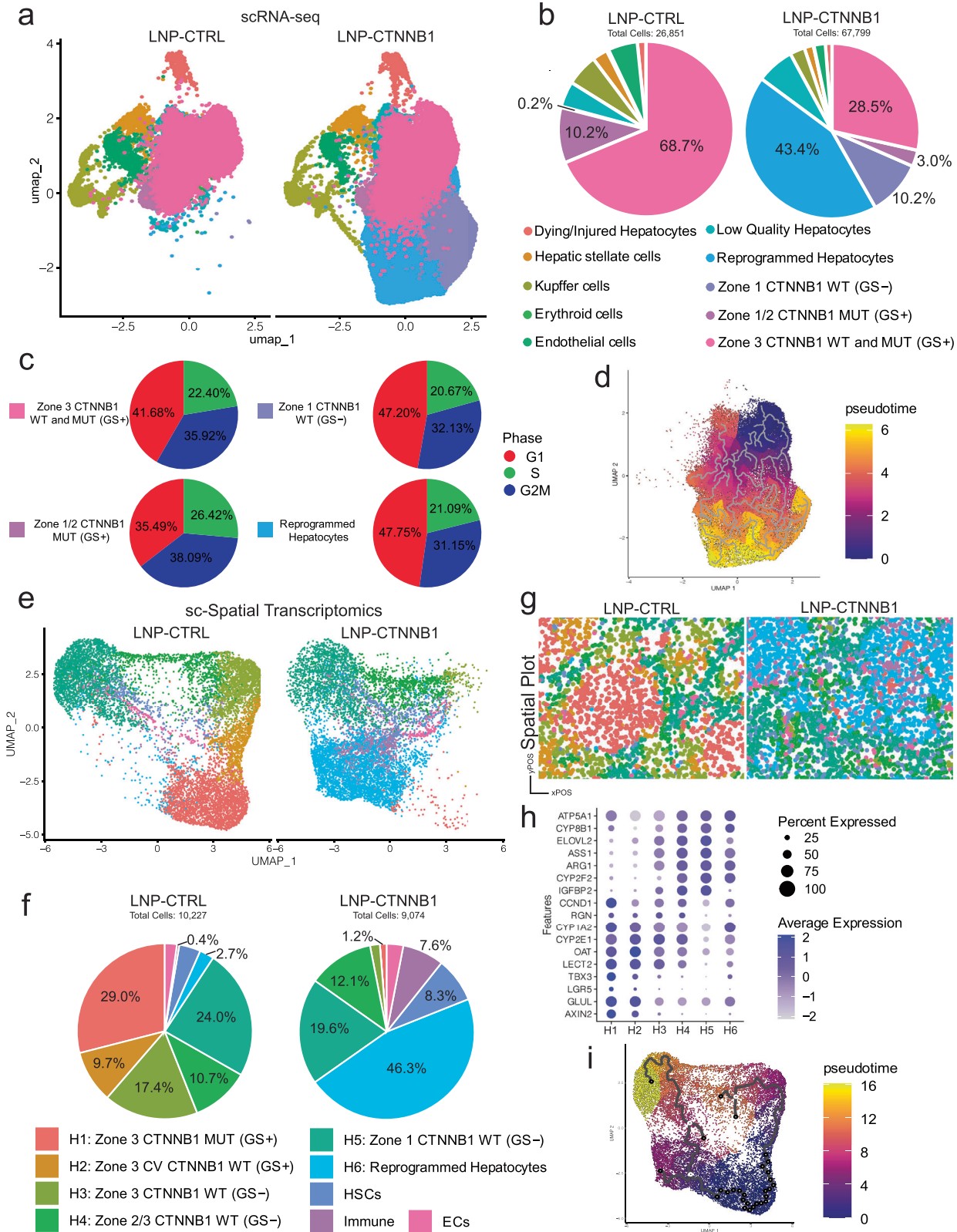

**Early β-catenin suppression induces an innate immune response characterized by type I/II interferon network signaling**

Spatial plots also revealed a significant increase in the immune cell cluster intratumorally in the LNP-CTNNB1 group compared to the LNP-CTRL group (Fig. 3g), which was also quantified (Supplementary Fig. 13c). To further investigate alterations in the immune landscape in an unbiased manner at the 3-day time point following LNP-CTNNB1

treatment, scRNA-seq was performed on an immune-enriched single-cell suspension from β-N treated animals. In total, 20,235 single cells were sequenced with 8499 cells across 3 individual biological replicates in the LNP-CTRL group and 11,736 cells across 3 individual biological replicates in the LNP-CTNNB1 group. Unbiased clustering on the integrated dataset across all cells resulted in 9 cell populations, which were annotated as: (a) T cells, (b) NK cells, (c) Myeloid cells, (d) B cells, (e) Dendritic cells,

**Fig. 3 | Integrated single-cell analyses reveal de novo formation of reprogrammed hepatocytes within remnant tumor nodules 3-days post-LNP-CTNNB1 treatment. a** Uniform manifold approximation and projection (UMAP) visualization of single-cell RNA-seq data following liver perfusion and enrichment of hepatocyte cell populations from LNP-CTRL and LNP-CTNNB1 treated β-catenin-Nrf2 (β-N) animals 3-days post-LNP treatment. UMAP split by treatment condition with 94,650 cells total across both treatment conditions. LNP-CTRL (n = 2) has 26,851 cells in the library; LNP-CTNNB1 (n = 3) has 67,799 cells in the library after data integration. **b** Pie chart of cell-type proportions between LNP-CTRL and LNP-CTNNB1 treatment conditions from (**a**). **c** Cell cycle regression scoring visualized via pie charts depicting cell cycle phase proportions in each of the indicated hepatocyte cell clusters. Each pie slice represents a group of cells colored by whether the RNA expression fits single cells belonging to G1 (red), S (green), or G2M (blue) phases of the cell cycle. **d** Pseudotime trajectory analysis on UMAP plot subset to only hepatocyte specific cell populations using the Zone 3 CTNNB1 WT and MUT (GS+) cell cluster as the root. **e** UMAP visualization of single-cell spatial

transcriptomic data via Molecular Cartogrpahy™ platform taken from frozen liver tissue sections of LNP-CTRL (n = 1) and LNP-CTNNB1 (n = 1) treated β-N animals 3-days post-treatment. UMAP generated based on expression of 100 genes. UMAP split by treatment condition with 19,301 cells total across both treatment conditions (LNP-CTRL library has n = 6 regions of interest (ROIs) with 10,227 cells total; LNP-CTNNB1 library has n = 5 ROIs with 9074 cells total). Labeled cell populations indicated by color. **f** Pie chart of cell-type proportions between LNP-CTRL and LNP-CTNNB1 treatment conditions from (**e**). **g** Spatial plots of LNP-CTRL and LNP-CTNNB1 regions of interest demonstrating visualization of certain cell populations by color from (**e**, **f**) on a virtual tissue section. **h** Dot plot visualization of various zonated marker gene expression (for all zones 1–3) for each hepatocyte cluster from (**e**, **f**). **i** Pseudotime trajectory analysis on Uniform manifold approximation and projection (UMAP) plot using the H1: Zone 3 CTNNB1 MUT (GS+) cluster as the root/origin. For (**b**), (**c**), and (**f**) labeled cell populations indicated by color and Source data are provided as a Source Data File.

(f) Stellate cells, (g) Endothelial cells, (h) Hepatocytes, and (i) Proliferative T cells, based on known marker gene expression for each of these cell types (Supplementary Fig. 14a, b). Across the 3 biological replicates for each treatment condition, the majority cell populations that were detected were T cells, B cells, NK cells, and myeloid cells (Supplementary Fig. 14c–e). We further subclustered and annotated these populations to better understand the T cell and myeloid cell functional states using marker genes previously described[36] (Supplementary Fig. 15a, b; Fig. 4a–c). The major difference observed following treatment was a 3-fold enrichment of M1-like pro-inflammatory macrophages in the LNP-CTNNB1 group (12.4%) compared to LNP-CTRL group (4.1%), which was trending toward significant enrichment when averaged across the 3 samples per LNP treatment groups (p = 0.0653) (Fig. 4b, d). However, at this 3-day time point following LNP-CTNNB1 treatment, we did not observe any significant differences in CD4+ T cell subpopulations in the β-N model from the scRNA-seq analysis (Supplementary Fig. 15c), or the sc-spatial transcriptomic analysis (Supplementary Fig. 15d, e). Additionally, in the the β-M model, IHC for CD4 did not reveal differences at the 3-day timepoint following LNP-CTNNB1 treatment (Supplementary Fig. 15f). CD8+ T cell subpopulations were not significantly altered following LNP-CTNNB1 treatment from this early scRNA-seq analysis (Supplementary Fig. 15c), although we did detect increased *CD8a* expression in tumoral compartment from the sc-Spatial Transcriptomic analysis (Supplementary Fig. 15d, e). Thus, innate immunity via myeloid cells, appears to be the predominant cell population which shifts at 3-days post-treatment (Fig. 4d).

To investigate functional changes within the M1-like macrophage population, we performed differential gene expression comparing the M1-like macrophages from LNP-CTRL and LNP-CTNNB1 treatment. GO pathway GSEA demonstrated enrichment of both response to type I/II interferon (IFN) and IFN alpha/beta pathways following LNP-CTNNB1 treatment (Fig. 4e). CellChat analysis, which suggests cell signaling pathway level changes based on gene expression of ligands and cognate receptors[37], showed enrichment of IFN-II and TNF signaling in the M1-like macrophage population following LNP-CTNNB1 treatment (Fig. 4f). Specifically, this analysis shows increased probability of cell communication via *Ifng* from proliferative T cells to *Ifngr1* and *Ifngr2* on M1-like macrophages, and other macrophage cell populations solely in the LNP-CTNNB1 group (Fig. 4g). Thus, increased type I/II IFNs released from the immune compartment (likely from T cells, macrophages, and dying tumor cells) following LNP-CTNNB1 treatment, engage with macrophages in the TIME milieu, and in part contribute towards polarizing them towards a pro-inflammatory anti-tumor phenotype. To validate our findings that a type I/II IFN response is, in part, a driver of the anti-tumor immune response following LNP-CTNNB1 treatment (Fig. 4f, g), we treated β-M mice with IFNγ 3× weekly for 5 weeks, which led to a significant decrease in tumor burden (Fig. 4h–j) accompanied by increases in S100A8/9-positive cells, a marker for M1-

like macrophages, when compared to vehicle control group (Fig. 4k). Thus, early β-catenin suppression in β-catenin-mutated tumors induces local IFN release which is likely recruiting and reprogramming intra-tumoral myeloid cells to drive an anti-tumor immune response.

## Mutated-β-catenin represses a module of transcription factors which drives immune exclusion in CTNNB1-mutated HCC

Given the amplified IFN response early after LNP-CTNNB1 treatment, we next investigated potential tumor cell-intrinsic molecular mechanisms driving this phentoype following β-catenin suppression. Also, β-catenin-mutated HCCs are well known for an immune cell excluded phenotype[7]. Analysis of TCGA-LIHC revealed *CTNNB1*-mutated HCCs downregulate pathways involved in type I/II IFN signaling, T cell activation, and chemokine signaling (Supplementary Fig. 16a, b). To identify potential mechanisms, we utilized bulk RNA-seq datasets which contained the transcriptome of multiple β-catenin-mutated HCC mouse models (GSE125336) and β-catenin knockout mouse livers (GSE68779) and performed transcription factor enrichment analysis on the 162 common genes downregulated in β-catenin-mutated HCC and upregulated in β-catenin knockout livers. We identified multiple transcription factors (TFs), including Irf2 (p = 0.0052) and Pou2f1 (p = 0.0023), as candidate TFs with known binding to the upregulated genes in β-catenin knockout livers (Fig. 5a). Interestingly, further analysis of Zone 3 CTNNB1 WT & MUT (GS+) hepatocytes from the scRNA-seq dataset (Fig. 3a) revealed upregulation of Irf2 and Pou2f1 target genes, inferred from an unbiased analysis (Fig. 5a), following LNP-CTNNB1 treatment (Fig. 5b). To further confirm hepatocytes as the cell source of Irf2 and Pou2f1, which would potentially drive an immune response upon β-catenin knockdown or loss, we investigated *IRF2/Irf2* and *POU2F1/Pou2f1* expression in both human and mouse liver scRNA-seq datasets[38] (GSE192742). We observed *IRF2/Irf2* and *POU2F1/Pou2f1* expression in hepatocyte cell populations in both human and mouse livers (Supplementary Fig. 16c, d), suggesting β-catenin-mediated IRF2/POU2F1 suppression is hepatocyte-intrinsic. Likewise, analysis of the TCGA-LIHC cohort revealed *IRF2* and *POU2F1* expression was not significantly different between patients with or without Wnt/β-catenin activity, rather, the target genes of IRF2/POU2F1 were significantly downregulated (p = 0.009) in patients with either *CTNNB1, AXIN1*, or *APC* mutations compared to other patients (Fig. 5c, d). High expression of IRF2/POU2F1 target genes was associated with improved disease-free survival (p = 0.01) in all TCGA-LIHC patients and in those with Wnt/β-catenin activating mutations (p = 0.065) (Supplementary Fig. 16e, f). Thus, we hypothesized that mutated-β-catenin is repressing a module of TFs driving immune exclusion and limiting an anti-tumor immune response.

To validate that repression of IRF2 and POU2F1 are driving immune exclusion in β-catenin-mutated HCC, we first overexpressed either pT3 (empty vector) or *Irf2* (β-M-IRF2) in the β-M model (Fig. 5e).

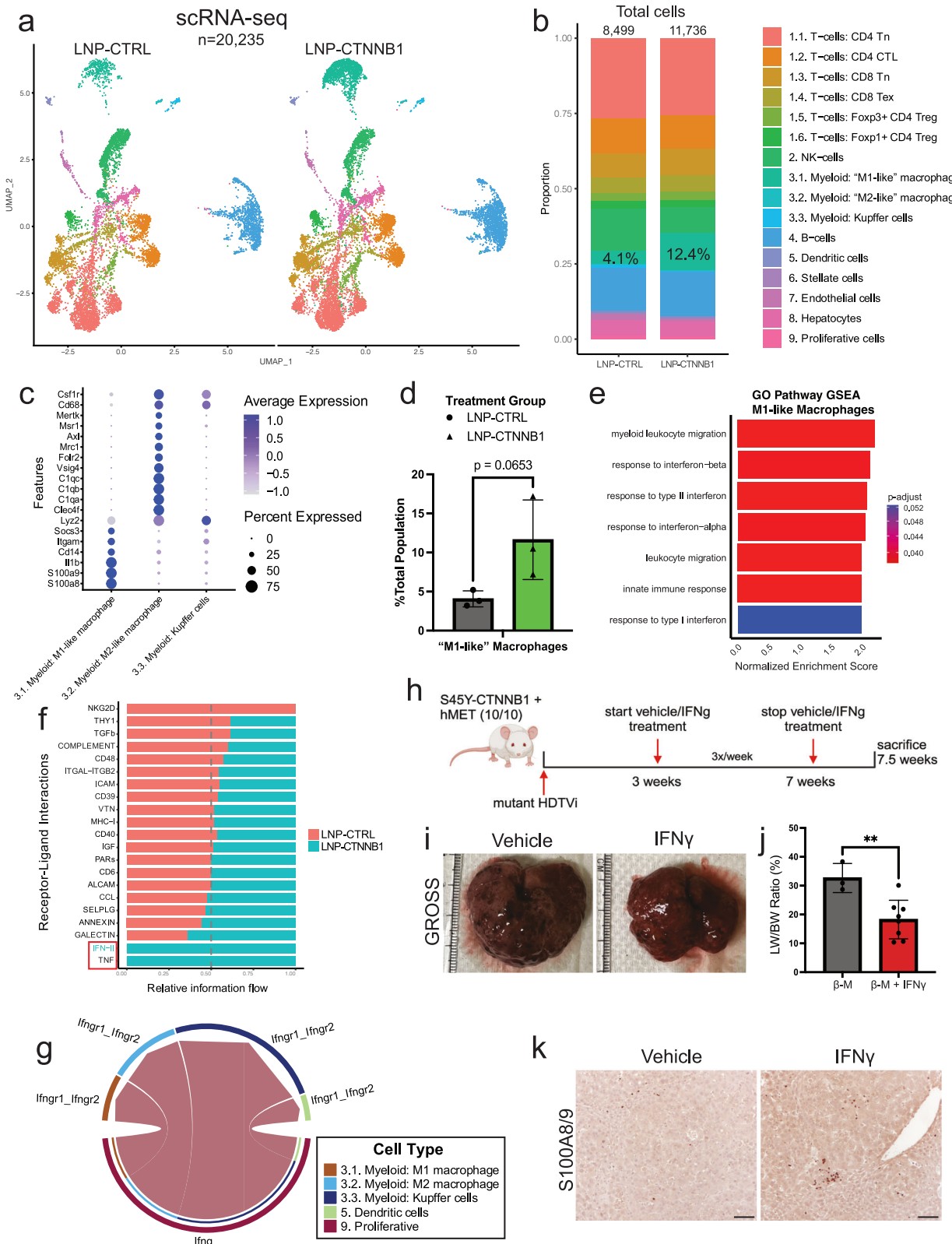

We observed a decrease in overall gross tumor burden and significant decrease in LW/BW in β-M-IRF2 mice at 7.5-weeks post-HDTVi (Fig. 5f–h). RNA-seq confirmed the overexpression of *Irf2* in the β-M-IRF2 mice at the 7.5-week timepoint where less tumor burden was evident (Supplementary Fig. 17a). Expectedly, given the known immunomodulatory roles of IRF2 and its involvement in type I/II IFN signaling[39], we observed increased immune aggregates as evident by

IHC for CD45 (Supplementary Fig. 17b). The composition of the immune infiltrate was examined with fluorescence-activated cell sorting (FACS) on isolated immune cells from β-M-pT3 and β-M-IRF2 HCC, which demonstrated significant increases in total CD4+ T cells with decreases in T regulatory cell populations in the β-M-IRF2 mice (Supplementary Fig. 17c; Supplementary Fig. 18a). Next, we overexpressed either pT3 (empty vector) or *POU2F1* (β-N-POU2F1) in the β-N model

**Fig. 4 | RNAi-mediated β-catenin inhibition induces infiltration of pro-inflammatory M1-like macrophages due to re-engaged interferon signaling.**
**a** UMAP visualization of integrated single-cell RNA-seq data following liver perfusion and enrichment of immune populations from LNP-CTRL ($n = 3$) and LNP-CTNNB1 ($n = 3$; 1 mg/kg) treated β-catenin-Nrf2 (β-N) animals 3-days post-LNP treatment. UMAP split by LNP treatment condition. **b** Stacked bar plot of cell-type proportions between LNP-CTRL ($n = 8499$ cells) and LNP-CTNNB1 ($n = 11,736$ cells) from (**a**). Labeled cell populations indicated by color for (**a**, **b**). **c** Dot plot visualization of expression of M1-like and M2-like macrophage phenotype markers. **d** Bar plot comparing percentage of M1-like macrophages amongst the total cell population in the LNP-CTRL ($n = 3$) and LNP-CTNNB1 ($n = 3$; 1 mg/kg) groups ($p = 0.0653$). **e** Gene Ontology (GO) pathway gene set enrichment analysis (GSEA) in the M1-like macrophage population comparing LNP-CTRL and LNP-CTNNB1 groups. **f** Stacked horizontal bar plot comparing relative information flow from CellChat between LNP-CTRL and LNP-CTNNB1 (1 mg/kg) groups. Boxed pathways show 100% information flow in LNP-CTNNB1 (1 mg/kg) group. IFN-II signaling highlighted in light blue show 100% enriched in LNP-CTNNB1 (1 mg/kg) group. **g** Chord diagram for IFN-II pathway in LNP-CTNNB1 (1 mg/kg) group demonstrating information flow

from proliferative T cells to macrophage populations. No information flow in LNP-CTRL treated animals. **h** IFNγ treatment schematic in β-catenin-hMet (β-M) model. Mice received multi-weekly intra-peritoneal (I.P.) injections of IFNγ at $1 \times 10^6$ IU/ml dosage or vehicle control starting at 3-weeks post-hydrodynamic tail vein injection (HDTVi). Mice were sacrificed at 7.5-weeks post-HDTVi. **i, j** Representative gross liver images and liver weight/body weight (LW/BW) comparing β-M animals treated with either vehicle control ($n = 3$) or IFNγ ($n = 8$) at 7.5-week timepoint ($p = 0.0086$). **k** Representative immunohistochemistry (IHC) images for S100A8/9 comparing Vehicle and IFNγ treated β-M animals. 10× objective lens, scale bar is 100 μm. IHC was repeated at least twice on tissue sections from multiple animals. Created in BioRender. Lehrich (2025) https://BioRender.com/9a8wkfh. For (**d**, **j**), data presented as mean values ± standard deviation (SD) and *P*-values calculated by unpaired two-tailed Student's *t*-test. For (**e**), normalized enrichment score (NES) was normalized from ES, which was calculated by Kolmogorov–Smirnov-like statistic and the adjusted *p*-value was calculated using the one-tailed empirical permutation test procedure. Source data are provided as a Source Data File. *$p < 0.05$, **$p < 0.01$, ***$p < 0.001$, ****$p < 0.0001$.

(Fig. 5i). We also observed a significant decrease in overall gross tumor burden at 10.7-weeks post-HDTVi in *POU2F1*-overexpression β-N model (Fig. 5j–l) and via histology (Supplementary Fig. 19a). These findings were also validated in the β-M model where significant reductions in tumor burden were observed at 7.7-weeks post-HDTVi (Supplementary Fig. 19b–f). IHC for CD4, CD8, and CD20 revealed T and B cells in aggregates in the TIME in the β-N-POU2F1 mice (Fig. 5m). RNA-seq confirmed the overexpression of *POU2F1* in the β-M-POU2F1 mice at the 7.7-week timepoint, along with decreased enrichment for published mutated-β-catenin gene signature (MBGS) (Supplementary Fig. 19g–i)[32]. Additionally, GSEA with GO pathways demonstrated enrichment of both T and B cell activation and proliferation (Fig. 5n). Lastly, given the less well characterized role of POU2F1 mediating an immune response, as compared to IRF2[39,40], we administered IgG/αCD3 to deplete CD3+ immune cells from β-M-POU2F1 mice (Supplementary Fig. 20a), which was confirmed via IHC for CD3 on spleens (Supplementary Fig. 20b). At 8.3-weeks post-HDTVi, there was an increase in gross tumor burden and significant increase in liver weight, LW/BW, and spleen weight in β-M-POU2F1 + αCD3 versus β-M-POU2F1 + IgG animals (Supplementary Fig. 20c–f), suggesting an immune-dependent role for POU2F1-mediated tumor regression in CTNNB1-mutated HCC. Overall, mutated-β-catenin represses IRF2, POU2F1, and likely other TFs, which limits transcription of key signaling molecules (i.e., cytokines and chemokines) important for priming lymphocyte recruitment needed for effective anti-tumor immunity.

## RNAi-mediated β-catenin inhibition impairs tumor growth in multiple immunocompetent CTNNB1-mutated HCC mouse models in advanced-stage disease with lack of response due to expansion of clones derived from secondary driver

To further assess translatability, we evaluated the in vivo activity of LNP-CTNNB1 in advanced-stage disease in β-M and β-N HCC models. First, we assessed response to LNP-CTNNB1 in the β-M model with once weekly I.V. treatments starting at 6-weeks post-HDTVi (Fig. 6a). We observed a heterogeneous response with 5/8 animals responding and 3/8 animals demonstrating minimal/poor response to LNP-CTNNB1 at 10.5-weeks post-HDTVi (Fig. 6b, c; Supplementary Fig. 21a). Next, we assessed response in the β-N model where we administered once weekly I.V. LNP treatments starting at 8-weeks post-HDTVi (Fig. 6d). Similar to the β-M model, after 6 cycles we observed a heterogeneous response with 5/8 animals responding and 3/8 animals demonstrating minimal/poor response to LNP-CTNNB1 at 13.5-weeks post-HDTVi (Fig. 6e, f; Supplementary Fig. 21b). Thus, LNP-CTNNB1 appears to be efficacious in advanced-stage disease setting, with a subset of animals responding less optimally.

To investigate the basis of the observed heterogeneous response, we first utilized the 10× Visium platform to perform unbiased spatial transcriptomics on an LNP-CTRL treated β-M HCC (β-M Control), 2 LNP-CTNNB1 treated β-M HCC showing minimal/poor response (β-M NR-1; β-M NR-2), and an LNP-CTNNB1 treated β-M HCC showing response (β-M R-1). In total, we sequenced 17,685 spots across the 4 slides, with 4461 spots in β-M Control, 4331 spots in β-M NR-1, 4842 in β-M NR-2, and 4051 spots in β-M R-1. After integrating the transcriptomes from all slides, unbiased clustering revealed 17 clusters conserved across the different conditions with changes in cluster proportions and spatial patterns of such changes (Supplementary Fig. 22a, b). Spatial plots and cluster proportion analyses revealed β-M R animals had increases in clusters 1, 2, 13, and 14, while β-M NR animals had increases in cluster 3 (Fig. 6g, h). We also performed 10× Visium spatial transcriptomics on an LNP-CTRL treated β-N HCC (β-N Control), 2 LNP-CTNNB1 treated β-N HCC showing minimal/poor response (β-N NR-1; β-N NR-2), and two LNP-CTNNB1 treated β-N HCC showing response (β-N R-1; β-N R-2) to validate findings from the β-M model. Here, we sequenced 17,130 spots across the 5 slides, with 3884 spots in β-N Control, 3624 spots in β-N NR-1, 4037 spots in β-N NR-2, 3390 spots in β-N R-1, and 2195 spots in β-N R-2. Integration and unbiased clustering across the β-N transcriptomes revealed 13 conserved clusters with changes in cluster proportions and spatial patterns of such changes (Fig. 6i; Supplementary Fig. 22c, d). β-N R animals showed increases in clusters 3, 4, and 5, while β-N NR animals showed increases in cluster 11 (Fig. 6j).

To ascertain mechanisms of response to LNP-CTNNB1, we first assessed whether there were cell cycle differences between NR and R animals in the β-M model. Utilizing gene signatures for S and G2M phases, spots belonging to specific clusters were categorized as belonging to G1, S, or G2M phases of the cell cycle (Supplementary Fig. 23a). There was a graded decrease in expression of S and G2M signature genes from control to NR to R animals (Supplementary Fig. 23b). Spatial mapping and quantification of the spots belonging to G1, S, or G2M phases revealed majority of spots corresponding to tumoral regions had increased expression of S and G2M signature genes in both β-M Control and NR animals, while β-M R animals showed majority of cell clusters to be in G1 phase (Supplementary Fig. 23c–e). Furthermore, RNA in situ expression of Mki67 and validation with IHC for Ki67 expressed throughout the cell cycle, confirmed active proliferation in tumor nodules of β-M NR animals (Supplementary Fig. 23f, g). These results were also observed in the β-N model (Supplementary Fig. 24a–g). Thus the NR phenotype in both models demonstrates active cellular proliferation likely due to insufficient β-catenin knockdown.

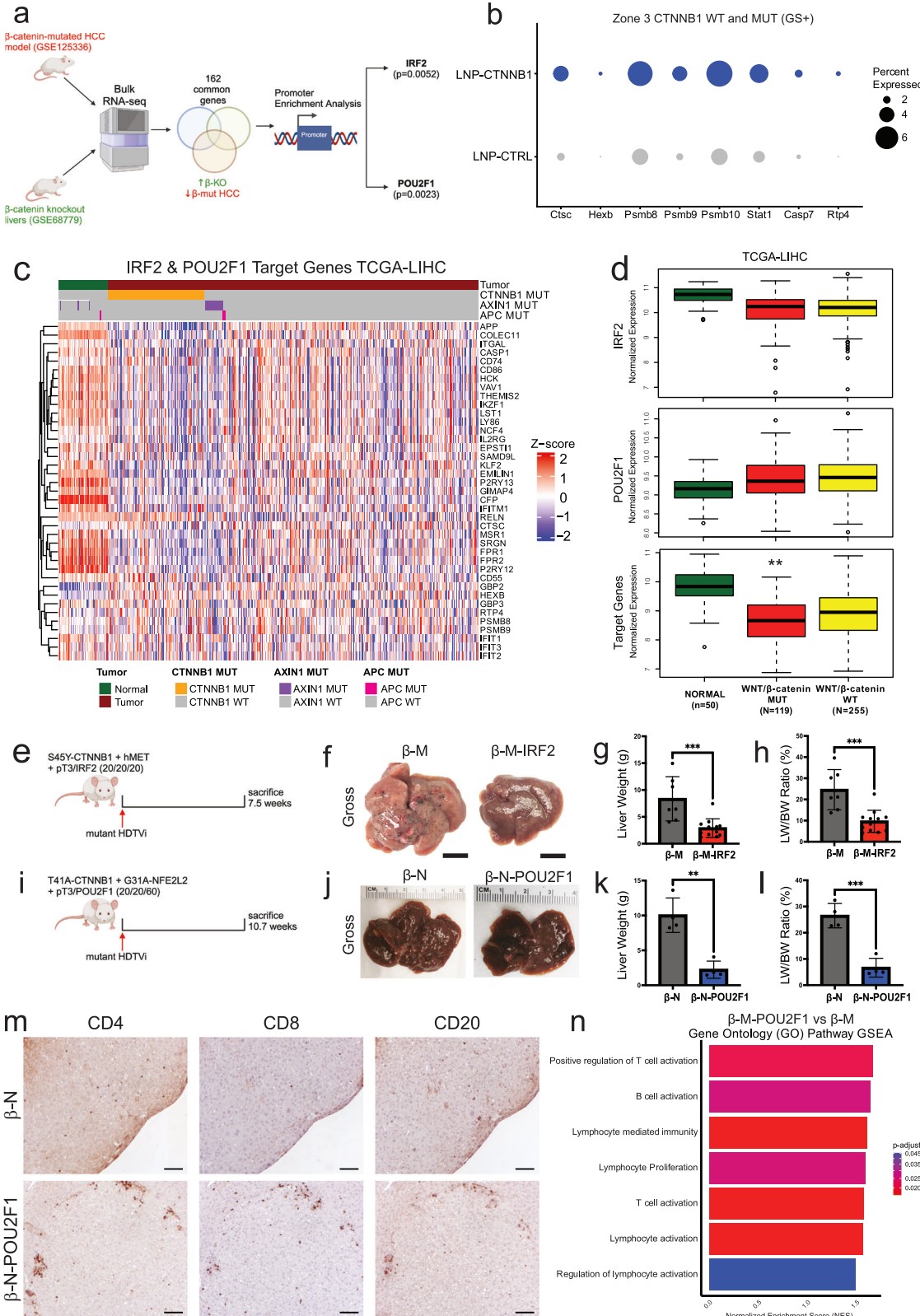

We next assessed whether the residual proliferating tumor nodules in the NR mice in both models were Wnt/β-catenin active. We evaluated expression of our previously reported mutated-β-catenin gene signature (MBGS)[32], which detects Wnt activation in whole and spatial transcriptomic datasets with high sensitivity and specificity. MBGS had graded decrease in expression from control to NR to R phenotype in both models (Supplementary Fig. 25a, b) and

highest expression in clusters 4, 9, and 12 in β-M model and in cluster 7 in β-N model (Supplementary Fig. 25c, d). As expected, we observed that the majority of tumor nodules which were proliferating in the NR phenotype were MBGS-high (Supplementary Fig. 25e, f). However, a subset of tumor nodules in the NR were MBGS-low and proliferating, implying active tumor growth despite sufficient β-catenin knock-down. These were cluster 3 in the β-M model and cluster 11 in the β-N

**Fig. 5 | IRF2 and POU2F1 repression by mutated-β-catenin is a major tumor cell-intrinsic mechanism of immune exclusion in *CTNNB1*-mutated HCC.**
**a** Schematic of pipeline comparing whole transcriptomes of β-catenin-mutated HCC to β-catenin knockout livers. **b** Dot plot highlighting *Irf2* and *Pou2f1* target genes within the Zone 3 CTNNB1 WT and MUT (GS+) cell population between LNP-CTRL and LNP-CTNNB1 from Fig. 3a, b. **c** Heatmap of *z*-scored expression values of IRF2/POU2F1 target genes in TCGA-LIHC patients ($n = 374$) and adjacent normal ($n = 50$). Data stratified by *CTNNB1*- ($n = 98$), *AXIN1*- ($n = 18$), and *APC*-mutated patients ($n = 3$). **d** Boxplots of normalized expression of *IRF2*, *POU2F1*, and IRF2/POU2F1 target genes stratified by adjacent normal ($n = 50$), Wnt/β-catenin-mutant ($n = 119$), and -wild-type ($n = 255$). **e** Schematic of β-catenin-hMet (β-M) animals co-injected with pT3 or IRF2 at time of hydrodynamic tail vein injection (HDTVi) and sacrificed at 7.5-weeks post-HDTVi. **f** Representative gross liver images from β-M-pT3 and β-M-IRF2 animals. Scale bar indicates 1 centimeter (cm). **g, h** Liver weights ($p = 0.0008$) and liver weight/body weight (LW/BW) ($p = 0.0003$) comparing β-M-pT3 ($n = 7$) and β-M-IRF2 ($n = 12$) animals at 7.5-week timepoint. **i** Schematic of β-catenin-Nrf2 (β-N) animals co-injected with pT3 or POU2F1 at time of HDTVi and sacrificed at 10.7-weeks post-HDTVi. **j** Representative gross liver images from β-N-

pT3 and β-N-POU2F1 animals. **k, l** Liver weights ($p = 0.0013$) and LW/BW ($p = 0.0005$) comparing β-N-pT3 ($n = 4$) and β-N-POU2F1 ($n = 4$) animals at 10.7-week timepoint. **m** Representative IHC images for CD4, CD8, and CD20 comparing β-N-pT3 and β-N-POU2F1 animals at 10.7-week timepoint. **n** GO pathway gene set enrichment analysis comparing β-M-POU2F1 to β-M-pT3. Created in BioRender. Lehrich (2025) https://BioRender.com/25qvbwj. Lehrich (2025) https://BioRender.com/qqvvb4x. Lehrich (2025) https://BioRender.com/ecs8omd. For (**g, h**), (**k, l**), data presented as mean values ± standard deviation (SD) and *P*-values calculated by unpaired two-tailed Student's *t*-test. For (**d**), the center line shows the median, the box limits show the interquartile range (IQR; the range between the 25th and 75th percentile) and the whiskers show 1.5× IQR. For (**d**), One-way ANOVA with Tukey-HSD post-hoc adjusted *p*-values comparing Wnt/β-catenin-mutant vs wild-type are: $p = 0.80$, $p = 0.54$, and $p = 0.009$ for *IRF2*, *POU2F1*, and IRF2/POU2F1 target gene expression, repsectively. For (**n**), NES was normalized from ES, which was calculated by Kolmogorov–Smirnov-like statistic and the adjusted *p*-value was calculated using the one-tailed empirical permutation test procedure. Source data are provided as a Source Data File. *$p < 0.05$, **$p < 0.01$, ***$p < 0.001$, ****$p < 0.0001$.

model (Fig. 6g, i). Spatial pseudotime analysis demonstrated these MBGS-low/Mki67-high nodules were insufficiently reprogrammed from zone 3 tumor to zone 1/2 hepatocytes (Supplementary Fig. 26a–d), and transcriptional trajectory indicating tumoral origin. Thus, a subset of tumor nodules in the NR phenotype may have persisted due to insufficient dosing (i.e., incomplete β-catenin suppression), yet some tumor nodules that expanded appear to be biologically distinct.

To identify biological origin of these persistent tumor nodules, we performed differential gene expression analysis across each cluster in the β-M (Supplementary Fig. 27a) and β-N models (Supplementary Fig. 28a). Cluster 3 in β-M model revealed upregulation of KEGG and GO pathways involved in PI3K-Akt signaling and actin cytoskeleton remodeling (Supplementary Fig. 29a). We then performed promoter enrichment analysis on the top DEGs from cluster 3 and identified MET as one of the top kinases predicted to be enriched (Fig. 6k). This was validated with V5-tag IHC where we observed persistence of V5-tag+ clones in NR phenotype (Supplementary Fig. 29b). Additionally, cluster 11 in the β-N model revealed upregulation of KEGG and GO glutathione metabolism and reactive oxygen species metabolic pathways (Supplementary Fig. 30a). Promoter enrichment analysis on the top DEGs from cluster 11 identified NFE2L2 as one of the top transcription factors predicted to be enriched (Fig. 6l). We also validated this finding with NQO1 IHC where we observed persistence of NQO1+ clones in NR phenotype (Supplementary Fig. 30b). Thus, MBGS-low nodules, which expanded in the NR phenotype, were likely derived from clonal expansion of tumor cells which were predominant upregulators of pathways specific to the secondary driver as a means to escape β-catenin suppression. Overall, heterogeneity in advanced-stage disease response appears to be due in part to insufficient β-catenin suppression and in part due to an escape mechanism.

### Response to RNAi-mediated β-catenin inhibition in advanced-stage disease is associated with restored adaptive immunity

Given the known role of β-catenin in promoting a non-T cell-inflamed TIME[7], we examined whether LNP-CTNNB1 sufficiently reprogrammed the adaptive immune response in advanced-stage HCC. First, we observed increased expression of T cell markers, including *Cd3d, Cd3e, Cd3g, Cd4*, and *Cd8a* in cluster 14 in the β-M model, (Supplementary Fig. 31a), which had expanded in the R phenotype. Additionally, cluster 14 in the β-M model showed increased expression of a T cell module score, which was also enriched in β-M R animals (Supplementary Fig. 31b, c). Interestingly, we also observed enrichment of these T cell markers in MBGS-high tumor-specific clusters (clusters 9 and 12 in β-M model) within R animals, demonstrating active T cell infiltration within residual tumor nodules, which was likely driving anti-tumor immunity

(Fig. 7a; Supplementary Fig. 31d). A similar amplified T cell response was observed in the β-N model with increased T cell marker expression within the MBGS-high cluster 7 in R animals (Supplementary Fig. 32a–d), although not as pronounced as the β-M model. In the β-M model, GSEA on clusters 9 and 12 comparing R to control demonstrated enrichment of GO pathways involved in leukocyte-mediated immunity, response to IFNγ, leukocyte cell adhesion, cell killing, and T cell proliferation (Supplementary Fig. 33a, b). Increased IFNγ expression was observed at the RNA and protein level following LNP-CTNNB1 treatment (Supplementary Fig. 33c, d). We validated the increased T cell infiltrate via IHC in the β-M model demonstrating increased CD3+ and CD4+ cells within tumor nodules with organization into lymphoid aggregates (LA) in the β-M R animals (Fig. 7b, c). We observed the LAs to be also composed of B cells, as noted by increased B cell markers gene expression in clusters 13 and 14, the MBGS-high tumor-specific clusters (clusters 9 and 12), specifically enriched in β-M R animals, along with increased CD20+ cells via IHC (Supplementary Fig. 34a–e). To further characterize the enhanced adaptive immune response in β-M R animals, we performed spatially enhanced CellChat[37] analysis to investigate ligand-receptor interactions between different clusters across the responders. This analysis revealed enrichment of MHC-II signaling with antigen communication from most clusters to CD4+ cells in cluster 12 specifically in β-M R animals compared to both β-M Control and β-M NR animals (Supplementary Fig. 35a–d). Moreover, an IRF2/POU2F1 target gene module score showed increased expression in cluster 12 of β-M R compared to NR and control animals (Supplementary Fig. 35e), further suggesting an IRF2-mediated IFN response upon β-catenin suppression. Overall, the R phenotype demonstrated reinvigorated and persistent adaptive immune surveillance with active T and B cell infiltration, T cell proliferation, and engaged IFN signaling in the intra-tumoral compartment, which was not observed in the NR phenotype in the advanced-stage disease setting.

### Immunotherapy enhances response to RNAi-mediated β-catenin inhibition in advanced-stage disease setting in CTNNB1-mutated HCC mouse model

Due to the known role of β-catenin promoting and IRF2 inhibiting transcription of *Cd274* (encoding PD-L1 on tumor cells)[41–43], a classical immune checkpoint mediating T cell exhaustion, we hypothesized that the NR phenotype may be driven by T cell exhaustion through increased expression of *Cd274*. Indeed, we observed increased *Cd274* and *Pdcd1* (encoding PD1 on T cells) expression in β-M NR and control compared to R animals (Fig. 7d; Supplementary Fig. 31e), implying lack of sustained and active lymphocyte effector function with increased exhaustion in NR, likely due to insufficient β-catenin suppression. This led us to hypothesize that administration of ICI would enhance

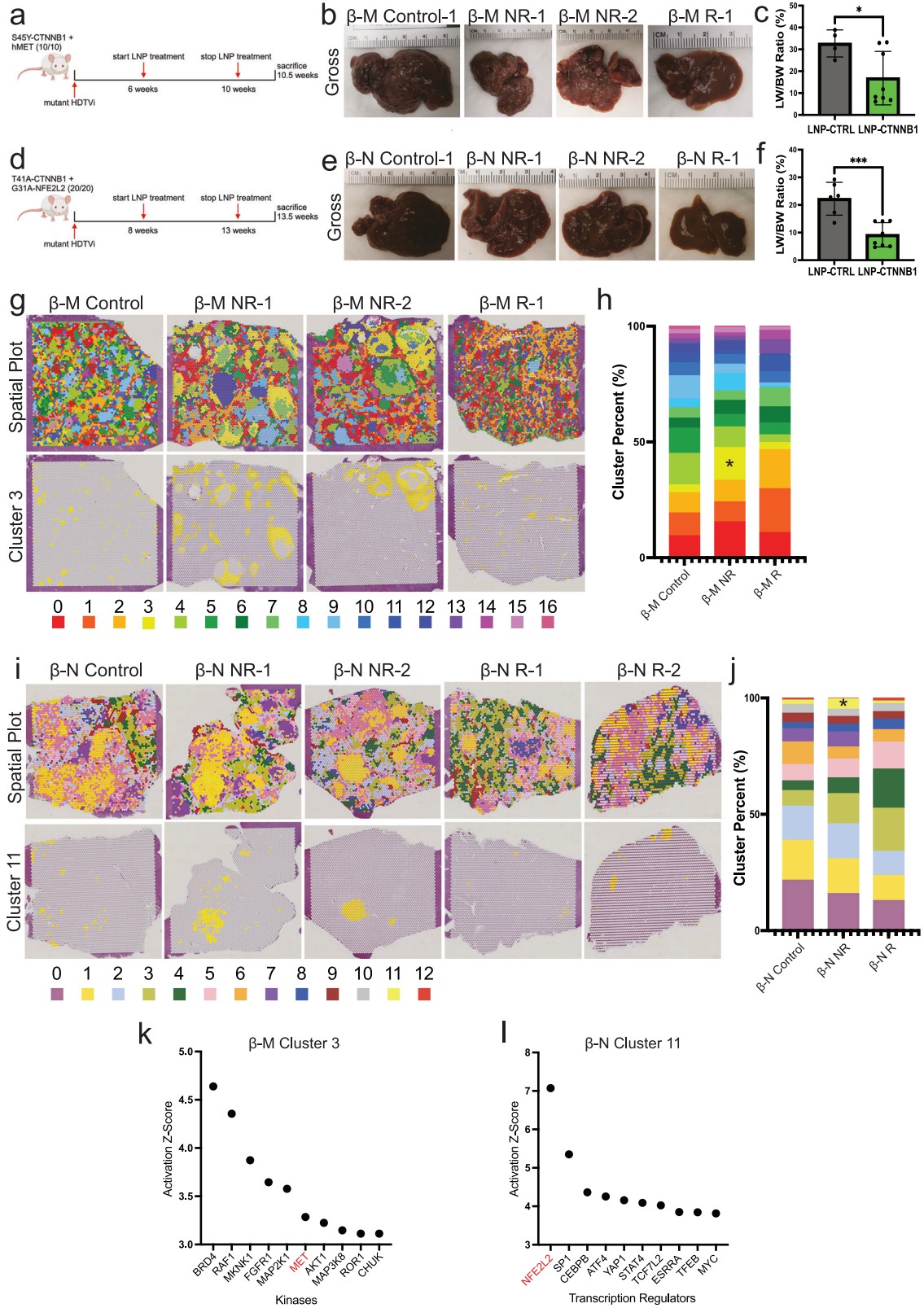

LNP-CTNNB1 response in advanced-stage disease through limiting T cell exhaustion via PD1/PD-L1 axis, thus promoting sustained anti-tumor immunity. IgG or α-PD1 was added 3-days after the first LNP dose with continued LNP treatment scheme as for the advanced-stage disease in the more aggressive β-M model. We harvested mice by 10.5-week timepoint or when moribund to assess response and performed a survival study to determine long-term anti-tumor immunity (Fig. 7e).

By 10.5-weeks, LNP-CTRL mice were all moribund with α-PD1 alone suggesting no impact on tumor burden, whereas the combination of LNP-CTNNB1 + α-PD1 resulted in enhanced efficacy with absence of any NR compared to LNP-CTNNB1 + IgG treated animals (Fig. 7f, g). Additionally, MRI demonstrated less hyperintense foci in LNP-CTNNB1 + α-PD1 compared to LNP-CTNNB1 + IgG treated mice (Fig. 7h). Decrease in *hCTNNB1*, indicative of tumor burden driven by mutant-CTNNB1, was

**Fig. 6 | Lack of response to RNAi-mediated β-catenin inhibition in *CTNNB1*-mutated HCC mouse models in advanced-stage disease setting is due to expansion of clones derived from the secondary driver. a** LNP treatment schematic in β-catenin-hMet (β-M) model for advanced-stage disease. Mice received once weekly intravenous (I.V.) injections at 1 mg/kg dosage starting at 6-weeks post-hydrodynamic tail vein injection (HDTVi) and sacrificed at 10.5-weeks post-HDTVi. **b** Representative gross liver images of LNP-CTNNB1 treated β-M animals at 10.5-week timepoint demonstrating non-responders (NR) and responders (R) compared to LNP-CTRL β-M animals when moribund at -8.5-weeks. **c** Liver weight/body weight (LW/BW) comparing LNP-CTRL ($n = 4$) and LNP-CTNNB1 ($n = 8$) treated β-M animals at 8.5-week and 10.5-week timepoint, respectively ($p = 0.0373$). **d** LNP treatment schematic in β-catenin-Nrf2 (β-N) model for advanced-stage disease. Mice received once weekly intravenous (I.V.) injections at 1 mg/kg dosage starting at 8-weeks post-HDTVi. **e** Representative gross liver images of LNP-CTNNB1 treated β-N animals at 13.5-week timepoint demonstrating NR and R compared to LNP-CTRL β-N animals when moribund at -10.5-weeks. **f** LW/BW comparing LNP-CTRL ($n = 6$) and LNP-CTNNB1 ($n = 8$) treated β-N animals at 10.5-week and 13.5-week

timepoint, respectively ($p = 0.0005$). **g** (Top) Spatial plot of all 17 clusters from the uniform manifold approximation and projection (UMAP) visualization showing the spatial localization on the H&E tissue section. (Bottom) Spatial plot of cluster 3 ident on the H&E tissue section. **h** Stacked bar chart of cluster proportions for the 3 LNP response (Control, NR, and R) groups. Asterisk denotes cluster 3. **i** (Top) Spatial plot of all 13 spot clusters from the UMAP visualization showing the spatial localization on the H&E tissue section. (Bottom) Spatial plot of cluster 11 ident on the H&E tissue section. **j** Stacked bar chart of cluster proportions for the 3 LNP response (Control, NR, and R) groups. Asterisk denotes cluster 11. **k** Waterfall plot of predicted kinases upstream of the differentially expressed genes (DEGs) in cluster 3 from the β-M model. **l** Waterfall plot of predicted transcriptional regulators upstream of the DEGs in cluster 11 from the β-N model. Created in BioRender. Lehrich (2025) https://BioRender.com/63585xs. Lehrich (2025) https://BioRender.com/6yyrkql. For (**c**, **f**), data presented as mean values ± standard deviation (SD) and *P*-values calculated by unpaired two-tailed Student's *t*-test. Source data are provided as a Source Data File. *$p < 0.05$, **$p < 0.01$, ***$p < 0.001$, ****$p < 0.0001$.

enhanced in the LNP-CTNNB1 + α-PD1 compared to LNP-CTNNB1 + IgG treated mice ($p = 0.0395$) suggesting an augmented response to LNP-CTNNB1 with α-PD1 (Fig. 7i). To determine whether there was enhanced effector T cell function in animals treated with LNP-CTNNB1 + α-PD1, we performed IHC for granzyme B (GZMB), a marker for cytotoxic T cells, which is known to be expressed at higher levels in tertiary lymphoid structures (TLS)/LA[44]. We observed an increased trend in the quantity of GZMB+LA within remnant tumor nodules in LNP-CTNNB1 + α-PD1 compared to LNP-CTNNB1 + IgG treated mice ($p = 0.0989$) (Fig. 7j, k). Concomitantly, mice receiving LNP-CTNNB1 + α-PD1 survived significantly longer than those receiving LNP-CTNNB1 + IgG ($p = 0.019$) (Fig. 7l), supporting an enhanced response to LNP-CTNNB1 with α-PD1. Overall, administration of α-PD1 augmented LNP-CTNNB1 response and restored adaptive immunity.

### TLS/LA are correlated with atezolizumab plus bevacizumab responders and inversely correlated with CTNNB1 mutational status in the IMbrave150 trial

Given the enhanced recruitment of both T and B cells into organized LA following LNP-CTNNB1 treatment in mice, we were interested in examining such a relationship between TLS/LA, *CTNNB1* mutation, and ICI response in patients. In the IMbrave150 phase III clinical trial, unresectable HCC patients received atezolizumab and bevacizumab in one arm versus sorafenib in another[6]. Atezolizumab and bevacizumab arm showed improved response rates, overall survival (OS) and progression-free survival (PFS). Accessing this information, 174 HCC patients were scored by a clinical pathologist for the presence of immune infiltration ranging from TLS, LA, diffuse infiltrate [DI], or none from the hematoxylin & eosin (H&E) slides. Overall, majority of the patients, irrespective of treatment arm, had LA ($n = 71/174$), while fewer had TLS ($n = 8/174$) or DI ($n = 8/174$) (Fig. 8a). Interestingly, among the responders, those in the atezolizumab plus bevacizumab arm tended to be enriched for presence of TLS/LA, which was not observed in the sorafenib arm (Fig. 8b). Additionally, patients with TLS/LA correlated with improved PFS and OS, which was more pronounced in the atezolizumab plus bevacizumab arm (Fig. 8c). Moreover, patients with TLS/LA had significantly increased expression of a LA-like gene signature, which consisted of previously reported marker genes for B cells, CD4 $T_{conv}$ cells, and CD4 $T_{fh}$ cells[45] compared to the patients with DI/None (Fig. 8d, e). Interestingly, high expression of the LA-like gene signature was associated with improved PFS ($p = 0.0639$) and OS ($p = 0.000697$) in the atezolizumab plus bevacizumab arm (Fig. 8f). Importantly, we observed that *CTNNB1*-mutated patients had significantly lower expression of the LA-like gene signature compared to *CTNNB1*-wild-type patients. This was also confirmed in the TCGA-LIHC cohort (Supplementary Fig. 36a–c). Thus, *CTNNB1*-mutated patients express fewer genes important for TLS/LA formation

compared to *CTNNB1*-wild-type patients, ultimately influencing response to combination ICIs.

## Discussion

We report strong in vitro and *vivo* efficacy of a novel LNP-formulated siRNA targeting *CTNNB1* mRNA transcript for treatment of β-catenin-mutated HCC as monotherapy in early-stage disease or in combination with ICI in advanced-stage disease. We identified through unbiased bulk RNA-seq, scRNA-seq, and spatial transcriptomic approaches tumor-cell-intrinsic roles of β-catenin-mediated IRF2 and POU2F1 repression driving an immune-excluded TIME and inert type I/II IFN responses in β-catenin-mutated HCC with in vivo validation. Additionally, we demonstrate upon β-catenin suppression, β-catenin-mutated tumor cells reprogram towards zone 1/2 hepatocyte-like cells, revealing the unique role of mutated-β-catenin in driving zone 3 (pericentral) tumor metabolism. Our work demonstrates that β-catenin is now directly targetable in murine HCC and supports the high impact development of clinical investigations utilizing LNP-CTNNB1 as monotherapy or in combination with ICI to achieve therapeutic benefit in HCC patients with Wnt/β-catenin activation.

β-Catenin is most active in the pericentral (zone 3) region in the hepatic lobule with hepatocytes in each of the three zones of the hepatic lobule expressing genes important for different metabolic functions, termed liver metabolic zonation[35]. Given the localization of β-catenin to zone 3, it is no surprise that β-catenin-mutated tumors preferentially originate and clonally expand from hepatocytes residing within zone 3, and these tumors share unique metabolic addictions to processes canonically identified in zone 3. In fact, we have previously shown that *CTNNB1*-mutated HCC is addicted to glutamine synthesis[46], as part of β-catenin-GS-mTOR axis[21]. Additionally, *CTNNB1*-mutated HCCs demonstrate addiction to xenobiotic metabolism through GSTM3[47]. However, surprisingly, tumors with β-catenin oncogenic activation are not glycolytic (zone 3 metabolism), but are fatty acid oxidative (zone 1 metabolism) addicted[48]. Here, we show that β-catenin-mutated tumors residing specifically in zone 3 are metabolically wired to perform canonical zone 3 metabolic processes with a focus on fatty acids as substrates, while β-catenin-mutated tumor cells in zone 1 are metabolically wired to perform canonical zone 1 metabolic processes with a focus on arginine metabolism and amino acid biosynthesis. We have also uniquely demonstrated that β-catenin-mutated tumor cells in zone 1 possess the highest proliferative capacity compared to those in zone 3, suggesting that despite β-catenin-mutated HCCs being well-differentiated, less proliferative tumors, in ectopic regions of absent Wnt signals or in presence of normal zone 1 signals, proliferation may be favored over metabolic homeostasis. Whether zone 1 β-catenin-mutated HCCs in current model are due to clonal expansion, evolution, or budding from zone 3 tumors to

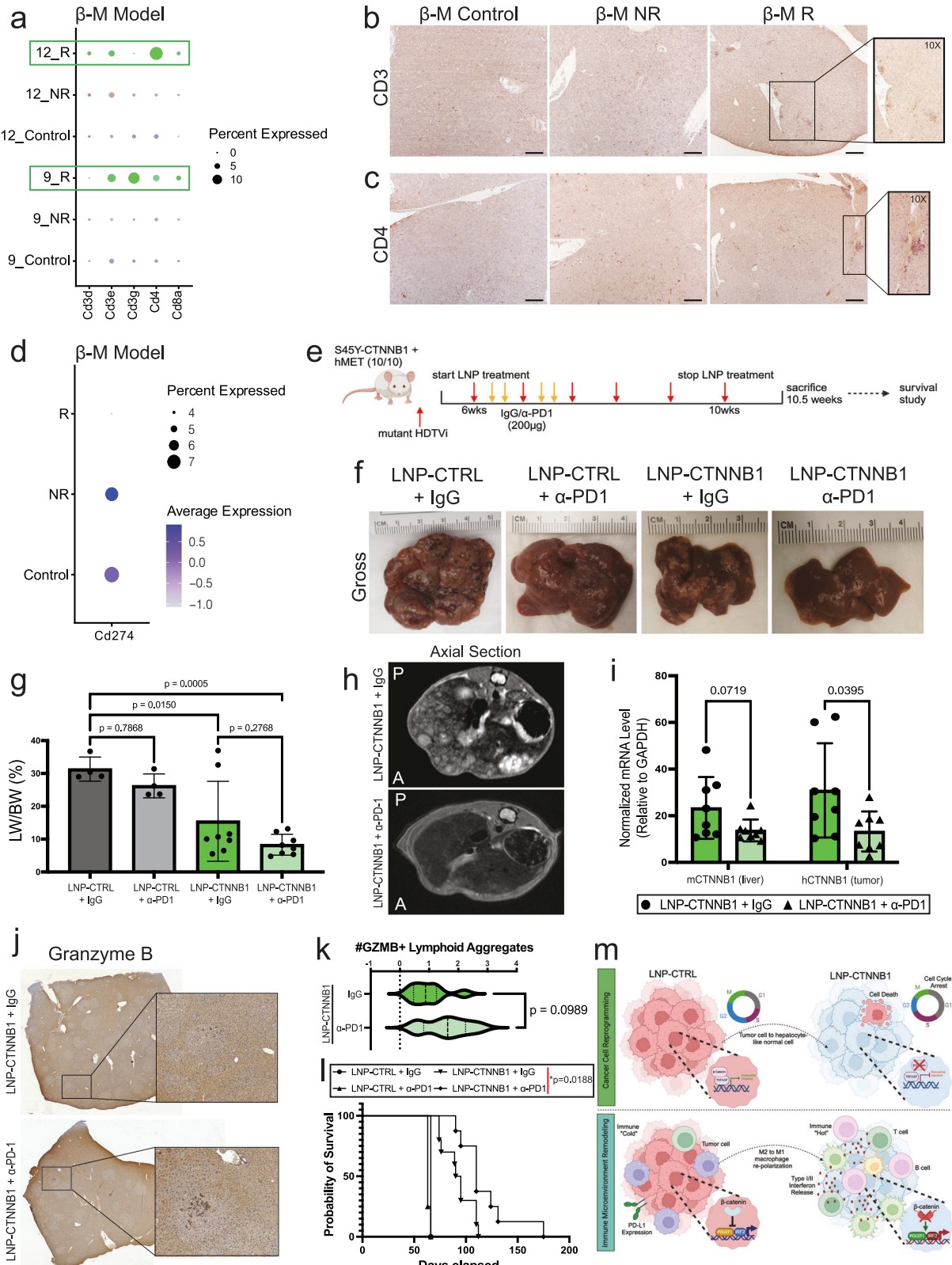

eventually establish in zone 1, or an artifact of plasmid transfection in rare hepatocytes in zone 1 requires further investigation. However, despite these tumor cell-intrinsic pathways, the overall tumor biology and metabolism may also be regulated by local zonal environment and signals. Overall, we demonstrate that suppressing β-catenin in *CTNNB1*-mutated tumors reprograms zone 3 tumors towards a zone 1/2 metabolic phenotype as early as 3-days post-LNP treatment, which

contributes to the phenotypic differentiation and metabolic rewiring, loss of tumor nodules, and normalization of hepatic parenchyma and liver mass. Such reprogramming may yield unique metabolic vulnerabilities to be exploited for additional therapies in the future.

Cancers with Wnt/β-catenin activation are considered non-T cell-inflamed across a variety of tumors, including HCC, melanoma, eso-phageal, and others[6,7,49,50]. This has been associated with mixed ICI

**Fig. 7 | ICI enhances response to RNAi-mediated β-catenin inhibition in β-M CTNNB1-mutated mouse model in advanced-stage disease setting. a** Dot plot of T cell markers (*Cd3d, Cd3e, Cd3g, Cd4, Cd8a*) in clusters 9 and 12 stratified by LNP treatment response (LNP-CTRL [blue]; non-responder [NR; red], responder [R; green]) in β-catein-hMet (β-M) from Fig. 6g, h. **b, c** Representative immunohistochemistry (IHC) images for CD3 and CD4 in β-M animals treated with LNP-CTRL or LNP-CTNNB1. 5× objective magnification. Scale bar indicates 200 μm. **d** Dot plot of *Cd274* expression by LNP treatment response in β-M. **e** Schematic of LNP + IgG/α-PD1 in β-M model. Mice received weekly LNP injections at 1 mg/kg dosage starting at 6-weeks post-hydrodynamic tail vein injection (HDTVi) with twice weekly injections of IgG/α-PD1 (200 μg) for two weeks starting 3-days after first LNP treatment. **f** Representative gross liver images of LNP-CTNNB1 ± IgG/α-PD1 treated β-M animals at 10.5-week timepoint compared to LNP-CTRL ± IgG/α-PD1 when moribund. **g** Liver weight/body weight (LW/BW) comparing LNP-CTNNB1 ± IgG/α-PD1 ($n = 8/n = 8$) treated β-M animals at 10.5-week timepoint to LNP-CTRL ± IgG/α-PD1 ($n = 4/n = 4$) when moribund. **h** Magnetic resonance images (MRI) of LNP-CTNNB1 ± IgG/α-PD1 treated β-M animals at 10.5-week timepoint. **i** RNA expression levels by qPCR of m*CTNNB1* and h*CTNNB1* in β-M animals treated with LNP-CTNNB1 ± IgG/α-PD1 ($n = 8$). **j** Representative tiled IHC images for granzyme B (GZMB) from LNP-CTNNB1 ± IgG/α-PD1 treated β-M animals at 10.5-week timepoint. Scale bar indicates magnification. **k** Violin plot showing quantification of number of GZMB+ lymphoid aggregates per tissue section in tumoral regions from LNP-CTNNB1 ± IgG/α-PD1 ($n = 8$) treated β-M animals at 10.5-week timepoint. **l** Kaplan–Meier overall survival (OS) curve comparing β-M animals treated with LNP-CTRL ± IgG/α-PD1 ($n = 3/n = 4$) or LNP-CTNNB1 ± IgG/α-PD1 ($n = 10/n = 8$) ($p = 0.0188$). **m** Schematic of two-part mechanistic working model for LNP-CTNNB1 response in β-catenin-mutated HCC. Created in BioRender. Lehrich (2025) https://BioRender.com/z4h1j67. Lehrich (2025) https://BioRender.com/hjjd4zn. For (**g, i**), data presented as mean values ± standard deviation (SD). For (**g**), *P*-values calculated by one-way ANOVA with Tukey-HSD post-hoc correction. For (**i, k**), *P*-values calculated by unpaired two-tailed Student's *t*-test. For (**l**), *P*-value calculated by log-rank test to compare difference in mean OS time between LNP-CTNNB1 + IgG and LNP-CTNNB1 + α-PD1. Source data are provided as a Source Data File. $*p < 0.05$, $**p < 0.01$, $***p < 0.001$, $****p < 0.0001$.

responses, specifically of the anti-PD1/anti-PD-L1 agent class[49]. In HCC, many chemokines are lowly expressed in *CTNNB1*-mutated patients, which directly influence immune cell recruitment, infiltration, and function[7]. For example, upon re-expression of CCL5, survival of β-catenin-mutated tumors was prolonged, along with increased antigen-specific CD8+ T cells[7]. Analogously, in KRAS-mutated colorectal cancer, where ICI is ineffective, expression of key chemokines involved in IFN network signaling, such as CXCL3, are downregulated when KRAS is mutated due to direct interaction with IRF2[39]. In the context of β-catenin-mutated tumors, previous work has described that β-catenin sequesters and inhibits p65 (a subunit of NF-κB) leading to reduced p65 binding, target gene expression, and inflammation[51–53]. Here, our unbiased bioinformatic analysis identified both IRF2 and POU2F1 as candidate TFs whose target genes are downregulated when β-catenin is active. We demonstrated that β-catenin suppression directly increases IFN signaling molecules and antigen presentation machinery components in β-catenin-mutated HCC murine models, with forced expression of either IRF2 or POU2F1 in two β-catenin-mutated HCC models sufficient to delay tumor development and convert a non-T cell-inflamed to a T cell-inflamed tumor. Thus, in β-catenin-mutated HCC, we posit that mutated-β-catenin is directly interacting with an immune-regulatory module of key TFs, which are normally active when β-catenin is absent from the nucleus, influencing expression of essential factors necessary for immune cell recruitment and function. Additionally, β-catenin has been shown to promote expression of PD-L1 on tumor cells by regulating expression of *Cd274*[41,42,54]. On the contrary, IRF2 has been shown to inhibit transcription of *Cd274* on tumor cells[43]. Adequate β-catenin loss thus prevents expression of this classical immune checkpoint mediating T cell exhaustion dually, to thus synergize with ICIs and promote anti-HCC effect which can have wider therapeutic implications. Overall, pharmacological targeting of β-catenin likely has clinical implications across a broad spectrum of tumor types to enhance ICI clinical efficacy in part through modulation of key TFs involved in priming immune cell recruitment and re-engaging global immune surveillance.

We have shown here that targeting β-catenin directly impacts both tumor cell-intrinsic biology and simultaneously reprograms the TIME from non-T cell-inflamed to T cell-inflamed, with innate immune remodeling occurring as early as 3-days post-LNP treatment. This innate immune remodeling coincided with first observed biological effect of β-catenin knockdown at 3-days. Biological effects due to siRNA knockdown are usually observed within hours in vitro[55], yet we observed a protracted time course in vivo, likely due to the systemic delivery method. Additionally, prior work has illustrated that adaptive immune surveillance begins to remodel at least 7–10 days following oncogene withdrawal, which explains the significant adaptive immune effects we observed studying advanced-stage response after 6 weeks

of LNP treatment[56]. However, the profound anti-tumor effects we observed here likely would not be so pronounced through targeting downstream effector molecules of the Wnt/β-catenin signaling pathway. Specifically, we and others have previously shown that genetic deletion or pharmacologic inhibition of downstream effectors of β-catenin-TCF/LEF interactions, such as cyclin D1 (encoded by CCND1)[57], GS[16], mTORC1[21], TBX3[17], AXIN2[58], or TNFRSF19[59] either result in partial tumor responses or compensatory negative feedback loops leading to enhanced tumorigenesis. For example, it has been shown that hepatocarcinogenesis is not dependent on cyclin D1 as β-catenin-mutated tumors induced in Ccnd1-null background mice still develop through compensatory cyclin D2 expression[57]. Additionally, conditionally deleting TBX3 or GS in mice with β-catenin-mutated HCC exacerbates tumorigenesis through YAP/TAZ inhibition or nitrogen metabolic rewiring, respectively[16,17]. Moreover, our group has previously identified metabolic addiction to β-catenin-GS-mTOR axis in β-catenin-mutated HCC and evaluated mTOR inhibitor (e.g., rapamycin, everolimus) response in multiple preclinical models of β-catenin-mutated HCC. However, response to LNP-CTNNB1 results in more consistent, robust, and durable responses in preclinical models[21,28]. Lastly, targeting solely TNFRSF19 will likely impact expression of chemokines involved in immune cell recruitment, yet there would be minimal impact on tumor cell-intrinsic biology[59]. Thus, targeting β-catenin directly is a holistic and rational strategy leading to durable anti-tumor immune responses through inhibiting multiple mechanisms hitting a truncal event, and impacting not only tumor cell-intrinsic biology, but also simultaneously remodeling the TIME architecture to promote long-lasting anti-tumor immunity.

Therapeutic targeting of Wnt/β-catenin oncogenic signaling has been pursued over the last two decades with no therapeutic agent ultimately resulting in translation to the clinic. First, given the ubiquitous role of β-catenin in many cell types, translation of many agents has been limited due to on-target, off-tumor effects[49,60]. Small-molecule inhibitors which limit interactions between β-catenin and TCF/LEF or β-catenin and cAMP response element–binding protein (CREB)–binding protein (CBP), or repurposed drugs against Wnt activity have shown in vitro inhibitory effects, yet lack strong in vivo efficacy, likely due to alternative escape mechanisms[9]. Alternative methods of Wnt/β-catenin inactivation have investigated porcupine (PORCN), tankyrase (TNKS), or Frizzled (FZD) receptor inhibitors, however, these are ineffective and far too upstream in the pathway for treating tumors with GOF *CTNNB1* mutations due to subsequent independence of Wnt/FZD receptor binding[9]. Thus, RNAi- or antisense-mediated gene silencing approaches have proven to be an effective therapeutic approach to reduce *CTNNB1* mRNA levels in tumors. Efficacy has previously been shown by our group and others across a variety of different tumor types[20,22,23,61]. Our work here builds upon

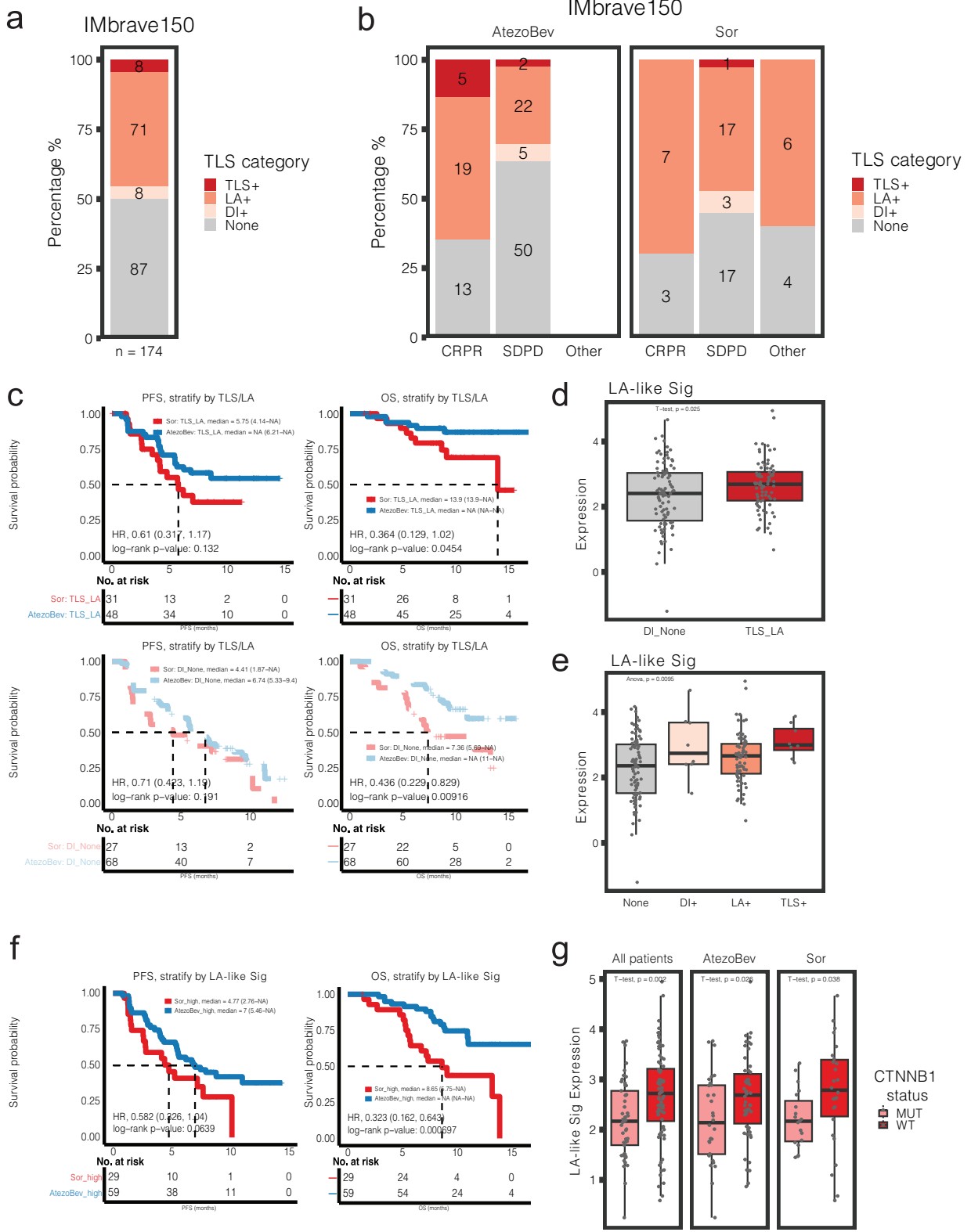

these previous findings and demonstrates that RNAi-mediated β-catenin inhibition via LNP for HCC results in minimal off-target effects with strong and durable on-target effects.

In summary, we propose a synergistic two-part working mechanism of response to RNAi-mediated β-catenin inhibition in preclinical *CTNNB1*-mutated HCC models (Fig. 7m). First, response to LNP-CTNNB1 treatment includes cessation of tumor cell proliferation

and concomitant metabolic zonal reprogramming with zone 3 tumor cells converting to zone 1/2 hepatocyte-like cells. Second, there is remodeling of the TIME with macrophages repolarizing towards a M1-like-phenotype, along with restored adaptive immune surveillance. This is driven by IRF2/POU2F1 re-engagement upon β-catenin knock-down, which act as mediators of enhanced IFN network signaling, suppression of PD-L1 expression, and priming of lymphocyte (T and

**Fig. 8 | Lymphoid aggregates are prognostic in hepatocellular carcinoma and negatively correlated with *CTNNB1* mutational status. a** Stacked bar plot of number of patients in IMbrave150 phase III trial having either tertiary lymphoid structure (TLS), lymphoid aggregate (LA), diffuse infiltrate (DI), or none (*n* = 174). **b** Stacked bar plot of number of patients in IMbrave150 phase III trial having either TLS, LA, DI, or None stratified by treatment group and clinical response: atezolizumab + bevacizumab (AtezoBev; *n* = 116) versus sorafenib (Sor; *n* = 58). **c** (Top) Kaplan–Meier progression-free survival (PFS) and overall survival (OS) curves comparing patients with TLS/LA in atezolizumab + bevacizumab versus sorafenib arms. (Bottom) Kaplan–Meier PFS and OS curves comparing patients with DI/None in atezolizumab + bevacizumab versus sorafenib arms (*n* = 174; AtezoBev, *n* = 116; Sor, *n* = 58). **d** Expression of a "LA-like" gene signature stratified by whether patients in IMbrave150 phase III trial had TLS/LA or DI/None (*n* = 174; *p* = 0.025). **e** Expression of a "LA-like" gene signature stratified by whether patients in IMbrave150 phase III trial had TLS, LA, DI, or None (*n* = 174; *p* = 0.0095). **f** (Left)

Kaplan–Meier PFS curve comparing patients with high LA-like gene signature in atezolizumab + bevacizumab versus sorafenib arms (*n* = 88; AtezoBev, *n* = 59; Sor, *n* = 29). (Right) Kaplan–Meier OS curve comparing patients with high LA-like gene signature in atezolizumab + bevacizumab versus sorafenib arms (AtezoBev, *n* = 59; Sor, *n* = 29). **g** Box plot depicting "LA-like" gene signature stratified by *CTNNB1* mutational status in all patients and within each of the two treatment arms from IMbrave150 phase III trial (*n* = 129; *p* = 0.002). For (**c**, **f**), hazard ratios (HRs) and 95% confidence intervals (CIs) were determined using a univariate Cox model. Kaplan–Meier log-rank test was used to compare differences in PFS/OS outcomes. For the boxplots in (**d**, **e**, **g**), the center line shows the median, the box limits show the interquartile range (IQR; the range between the 25th and 75th percentile) and the whiskers show 1.58× IQR. For (**d**, **g**), *P*-values were calculated using two-tailed unpaired *t*-test. For (**e**), *P*-value was calculated using one-way analysis of variance (ANOVA).

B cell) recruitment and infiltration. All these tumor cell-intrinsic and TIME remodeling mechanisms ultimately drive enhanced efficacy of LNP-CTNNB1 to α-PD1 in advanced-stage disease. Based on our findings, RNAi-mediated inhibition of β-catenin may have the potential to provide anti-tumor effects as a monotherapy in early-stage disease or in neoadjuvant setting in patients with Wnt-β-catenin active liver tumors. These proof-of-concept studies also support the clinical investigation of RNAi therapeutic approaches targeting β-catenin in combination with ICI in advanced-stage Wnt-β-catenin active-HCC patients.

## Methods

### Mice for tumor study

Approval for all animal husbandry and experimental procedures, including animal housing and diet, complies with all relevant ethical regulations, and performed under the guidelines and approval of the National Institutes of Health and the University of Pittsburgh School of Medicine Institutional Animal Use and Care Committee (IACUC). All FVB/NJ male mice were purchased from the Jackson Laboratory (Bar Harbor, ME). Ctnnb1$^{f/f}$ (on a C57Bl6 background) male and female mice were used for this study as well and purchased form the Jackson Laboratory (Bar Harbor, ME). For β-catenin-mutated HCC studies, we used 6–8-week-old FVB/NJ male mice. Male mice were utilized given the increased prevalence of liver cancer in males. Mice were fed a standard chow diet ad libitum, with access to water, enrichment, and exposed to 12 h light/dark cycles in ventilated cages. Mice were housed to individual cages with each having 3–4 mice. Body weights were monitored once weekly. As tumors developed in mice over time, they were monitored for signs of morbidity, including increased abdominal distension/girth, grimace, and body weights not to exceed 20–25% of aged-matched controls, and were appropriately euthanized and considered moribund. Mice were fasted 4–6 h prior to sacrifice. At time of sacrifice, gross images of the liver were recorded with a handheld camera, along with documenting the body and liver weights, and gross morphology of the mouse livers with a ruler measurement. At time of sacrifice, whole blood was drawn from inferior vena cava via insulin syringe and expunged into microcentrifuge tubes to be spun at 2500 × *g* for 10–15 min at room temperature to collect serum. Serum was then collected, processed, and sent to be analyzed for liver function tests, including aspartate aminotransferase (AST), alanine aminotransferase (ALT), and alkaline phosphatase via UPMC Clinical Pathology Laboratories.

### Plasmids

For mutant-β-catenin constructs, we utilized the S45Y-hCTNNB1-Myc-tag and T41A-hCTNNB1-Myc-tag plasmids, which are human constructs of the gene[27,28]. Additionally, we utilized the hMet-V5-tag and G31A-hNFE2L2 plasmids, which are also human constructs of the genes[27,28]. Both the T41A-hCTNNB1-Myc-tag and G31A-hNFE2L2 plasmids were

subcloned into the pT3-EF1αh destination vector, while the S45Y-hCTNNB1-Myc-tag and hMet-V5-tag plasmids were subcloned into the pT3-EF5αh destination vector[27,28]. We also utilized the myr-AKT and N-RasV12 plasmids known to induce Akt/NRas HCC[62]. We purchased the mouse pFUW-tetO-Irf2 plasmid from Addgene (plasmid #139814) and pENTR223-POU2F1 plasmid from DNASU (plasmid # HsCD00505520). We then subcloned these plasmids into the pT3-EF1αh plasmid using Gateway PCR cloning technology (Invitrogen, Carlsbad, CA) to make a pT3-EF1αh-Irf2 and pT3-EF1αh-POU2F1 plasmids. The pCMV-pT3, pCMV-SB transposase, and pCMV-Cre plasmids have been described previously[27]. The plasmid constructs were purified using Endotoxin-Free Maxiprep kit (NA 0410, Sigma-Aldrich, St. Louis, MO) for eventual hydrodynamic delivery. For hydrodynamic delivery, plasmids were diluted into 0.9% normal saline (NaCl) purchased from TEKNOVA (#S5815).

### HCC animal models developed via sleeping beauty-hydrodynamic tail vein injection

We utilized the sleeping beauty-hydrodynamic tail vein injection (SB-HDTVi) model, which has been extensively described by our group[20,27,28]. We utilize 5 HCC mouse models in this study: (1) T41A-hCTNNB1/G31A-hNFE2L2 (β-N), (2) S45Y-hCTNNB1/hMet (β-M), (3) S45Y-hCTNNB1/G31A-hNFE2L2/hMet (β-N-M), (4) G31A-hNFE2L2/hMet (N-M), and (5) Akt/NRas models. For the β-N model, 20 µg of pT3-EF1αh-T41A-hCTNNB1-Myc-tag and 20 µg of NFE2L2-plasmid (pT3-EF1αh-G31A-hNFE2L2) were mixed. For the β-M model, 10 µg of pT3-EF5αh-S45Y-hCTNNB1-Myc-tag and 10 µg of hMET-plasmid (pT3-EF5αh-hMet-V5-tag) were mixed. For the β-N-M model, 10 µg of pT3-EF5αh-S45Y-hCTNNB1-Myc-tag, 10 µg of NFE2L2-plasmid (pT3-EF1αh-G31A-hNFE2L2), and 10 µg of hMET-plasmid (pT3-EF5αh-hMet-V5-tag) were mixed. For the N-M model, 20 µg of NFE2L2-plasmid (pT3-EF1αh-G31A-hNFE2L2) and 20 µg of hMET-plasmid (pT3-EF5αh-hMet-V5-tag) were mixed. For the Akt/NRas model, 10 µg pT3-EF1α-HA-myr-AKT and 10 µg of pT2-Caggs-N-RasV12 were mixed with pCMV-pT3 (60 µg; empty vector; solely the backbone plasmid) or pCMV-Cre plasmids (60 µg) to delete endogenous genes in *Ctnnb1* floxed mice background. For β-M-pT3 and β-M-IRF2 models, we mixed 20 µg of pT3-EF5αh-S45Y-hCTNNB1-Myc-tag, 20 µg of hMET-plasmid (pT3-EF5αh-hMet-V5-tag), and either 20 µg of pCMV-pT3 or Irf2 plasmid (pT3-EF1αh-Irf2). For the β-M-pT3 and β-M-POU2F1 models, we mixed 20 µg of pT3-EF5αh-S45Y-hCTNNB1-Myc-tag, 20 µg of hMET-plasmid (pT3-EF5αh-hMet-V5-tag), and either 60 µg of pCMV-pT3 or POU2F1 plasmid (pT3-EF1αh-POU2F1). For the β-N-pT3 and β-N-POU2F1 models, we mixed 20 µg of pT3-EF1αh-T41A-hCTNNB1-Myc-tag, 20 µg of Nrf2-plasmid (pT3-EF1αh-G31A-hNFE2L2), and either 60 µg of pCMV-pT3 or POU2F1 plasmid (pT3-EF1αh-POU2F1). pCMV-Sleeping Beauty transposase (pCMV/SB) plasmid is mixed with each of the oncogenic plasmids at a concentration of 25:1 in 2 mL normal saline (0.9% NaCl) and filtered

through 0.22 μm filter (Millipore) for hydrodynamic injection. For hydrodynamic injection, 2 mL volume is injected in the lateral tail vein in 5–7 s.

## LNP-siRNA synthesis

Chemically modified and functionalized siRNAs for this study were synthesized and characterized by Alnylam Pharmaceuticals[63,64]. The lipid nanoparticles (LNPs) containing encapsulated siRNAs were formulated and characterized using validated techniques[65,66] with the biodegradable ionizable lipid bis(3-pentyloctyl) 9-((4-(dimethylamino)butanoyl)oxy) heptadecanedioate. The LNPs used in this study size range from 60 to 80 nm with siRNA encapsulation efficiency of ~95% (PDI = 0.045) with quality control performed by Alnylam Pharmaceuticals[31]. LNP formulations from Alnylam Pharmaceuticals have demonstrated a rapid accumulation of the parent lipids in the liver and plasma of mice with elimination from plasma by 8-h, liver by 48-h, and spleen by 96-h post systemic administration via tail vein injection[31]. Biodistribution of the LNPs in mice have also been reported to accumulate primarily in the liver and spleen[31]. A control siRNA was synthesized which is not predicted to bind to any location in the transcriptome and the lack of activity was confirmed by RNA-sequencing (RNA-seq) analysis performed by Alnylam Pharmaceuticals as part of quality control. The siRNA targeting CTNNB1 has demonstrated ability to reduce levels of both mouse and human *CTNNB1* transcripts. In this study we refer to the LNP encapsulating control siRNA as LNP-CTRL and LNP encapsulating siRNA targeting *CTNNB1* transcript as LNP-CTNNB1.

## siRNA sequences

siCTNNB1: sense – UACUGUUGGAUUGAUUCGAAA,
antisense - UUUCGAAUCAAUCCAACAGUAGC
siControl: sense ACCCAGAUAUUUUUAUCGCGU,
antisense – ACGCGAUAAAAAUAUCUGGGUCG

## Patient-derived organoid cultures

*CTNNB1*-mutated patient-derived HCC organoid 23277, which has a S33F mutation in exon 3 of *CTNNB1*, was kindly provided by Ernesto Guccione[26]. These organoids were cultured and expanded in 24-well plates with tumor organoid medium[26]. Organoids were treated with LNP-CTRL or LNP-CTNNB1 at 20 nM concentration and imaged at 48-h and 72-h post-LNP treatment. Experiments were performed in triplicate.

## Various therapeutic interventions and dosing

The therapeutic formulations of LNP-CTRL and LNP-CTNNB1 were diluted from their stock formulations into phosphate buffered saline (PBS) to reach final concentrations of either 3, 1, 0.3, 0.1, and 0.03 mg/kg dosages. The PBS/LNP-CTRL/LNP-CTNNB1 formulations were administered once weekly at the indicated time points below via lateral tail vein (intravenous; I.V.) injection using insulin syringe as the delivery method. For early-stage intervention in the β-N model, dosing started at 5-weeks post-HDTVi. For early-stage intervention in the β-M model, dosing started at 3-weeks post-HDTVi. For early-stage intervention in the β-N-M model, dosing started at 3-weeks post-HDTVi. For early-stage intervention in the N-M model, dosing started at 8-weeks post-HDTVi. For late-stage intervention in the β-N model, dosing started at 8-weeks post-HDTVi. For late-stage intervention in the β-M model, dosing started at 6-weeks post-HDTVi. For combination therapy study of LNP-CTNNB1 with α-PD1 in β-M model at late-stage, mice were randomized to receive either 200 μg of either IgG control antibody (BioXcell; InVivoMAb rat IgG2a isotype control; Catalog #BE0089) or α-PD1 antibody (BioXcell; InVivoMAb anti-mouse PD1 (CD279); Catalog #BE0146) starting 3-days after the 6-weeks post-HDTVi timepoint when given LNP-CTRL/LNP-CTNNB1 treatments. The animals received IgG/α-PD1 2×/week over 2-weeks. The IgG/α-PD-1treatments were given via intra-peritoneal (I.P.) injection. For IFNγ

(Miltenyi; #130-105-774) treatments, mice were treated at the indicated time points at a concentration of $1 \times 104$ IU starting 3-weeks post-HDTVi. For α-CD3 immune depletion studies, β-M-pT3 or β-M-POU2F1 mice were treated with either IgG isotype control (BioXcell; InVivoMAb rat IgG2a isotype control; Catalog #BE0089) or InVivoMAb α-mouse CD3 (Catalog #BE0002, clone 17A2) at 200 μg concentration at the indicated time points.

## Sample collection and processing

At the time of animal sacrifice, whole blood was collected via either drainage from IVC or retroorbital blood draw. Serum was processed through centrifugation at $3000 \times g$ for 10 min at cold 4 °C temperature (supernatant contained the serum). Liver tissue from all samples were either fixed in 10% neutralized formalin (Fisher Chemicals) at room temperature for 48–72 h, followed by transfer into 70% ethanol, dehydration, and paraffin embedded for histology analysis, or flash frozen in liquid nitrogen and stored long-term at −80 °C.

## Hematoxylin and Eosin (H&E) staining

Formalin, fixed paraffin embedded (FFPE) liver tissue was sectioned at 4 μm thick slices using a standard microtome. Liver tissue sections were deparaffinized in xylene and graded ethanol followed by Hematoxylin (Fisher Chemical Harris Modified Method Hematoxylin Stains #SH26-500D) and Eosin (Eosin Y, #23-314-63) and dehydrated, mounted, and cover-slipped. Slides were imaged on Zeiss Axioskop microscope.

## Immunohistochemistry (IHC)

**Single-stain for IHC.** FFPE sections were deparaffinized in xylene and graded ethanol (10%, 95%, 90%) and rinsed with PBS or water. For antigen retrieval, samples were placed in either Citrate Buffer (0.01 M, pH 6.0), Tris-EDTA (0.01 M, pH 9.0) or DAKO (Agilent Technologies; S236784-2) and heated in a steamer or a pressure cooker for 20 min. Next, samples were placed in 3% $H_2O_2$ solution (Fisher Chemicals) for 10 min to quench endogenous peroxide activity. After appropriates washing in saline, slides were blocked with Super Block (ScyTek Laboratories) for 10 min to prevent non-specific antibody binding. Next, the sections were incubated with the primary antibodies diluted in PBS at 4 °C overnight or at room temperature. A list of primary antibodies and the specific antigen retrieval conditions for each is provided in Supplementary Table 1. This was followed by incubation with species-specific biotinylated secondary antibodies for 15–30 min at room temperature. Next, the sections were washed with 1× PBS and incubated with ABC reagent (Vectastain ABC Elite kit, Vector Laboratories) for 5–10 min followed by washing with 1× PBS. The signal was detected using DAB Peroxidase Substrate Kit (Vector Laboratories) using the positive control to determine timing of signal. Appropriate negative controls were used. Finally, slides were counterstained with hematoxylin (Thermo Fisher Scientific), dehydrated, mounted, and cover-slipped. 5× and 10× views were imaged on Zeiss Axioskop microscope. Whole-slide tiled images were taken with a Fritz Precipoint microscope (Nikon).

**Dual-stain for IHC.** For co-staining of glutamine synthetase (GS)/Ki67 via IHC, slides were initially incubated with GS (and subsequently with species-specific biotinylated secondary antibody) and developed using ABC-AP reagent (Vectastain ABC Elite kit, Vector Laboratories) for 30 min and incubated with Vector Red reagent (Vector Laboratories) and developed in the dark for 15 min. On the second day, slides were incubated with Ki67 (and subsequently with biotinylated secondary antibody) and developed using ABC reagent (Vectastain ABC Elite kit, Vector Laboratories) for 10 min and incubated with DAB reagent (Vector Laboratories) and developed in the light for 1–2 min. Finally, slides were counterstained with hematoxylin (Thermo Fisher Scientific), dehydrated, mounted, and cover-slipped. 5× and 10× views were

imaged on Zeiss Axioskop microscope (EMD Millipore). Whole-slide tiled images were taken with a Fritz Precipoint microscope (Nikon).

**Quantification of % area for IHC.** IHC Images stained with indicated markers and quantified with %Area of indicated marker were exported as.TIFF files from the Zeiss Axioskop microscope and loaded into ImageJ. A color deconvolution plugin was downloaded (https://github.com/landinig/IJ-Colour_Deconvolution2/blob/main/colour_deconvolution2.jar) and used to deconvolute the different color channels. The brown stain channel was used to color threshold based on the intensity of the DAB staining on IHC. The same threshold parameters were used for each indicated stain comparing LNP-CTRL and LNP-CTNNB1. Raw data was exported to an Excel file and analyzed in GraphPad Prism. Technical tissue replicates were averaged across to determine the %Area for each biological replicate value. Thus, individual data points are reported as an average of the technical replicates.

**RNA extraction and cDNA preparation.** Whole liver tissue chunks were homogenized in TRIzol (Thermo Scientific) and RNA was extracted with RNeasy Micro Kit (Qiagen) following the manufacturer instructions. RNA was reverse transcribed into cDNA using SuperScript III (Invitrogen). cDNA was diluted in nuclease free water and stored long-term at −20 °C.

**Quantitative reverse transcriptase polymerase chain reaction (qRT-PCR).** Real-time PCR was performed in technical duplicates on a StepOnePlus Real-Time PCR System (Applied Biosystems) using the Power SYBR Green PCR Master Mix (Applied Biosystems). Target gene expression was normalized to the housekeeping genes Rn18s, and fold change was calculated utilizing the ΔΔ-Ct method. Primers used are listed in Supplementary Table 2.

**qRT-PCR for mouse-specific Ctnnb1 and human-specific CTNNB1.** This analysis was performed by Alnylam Pharmaceuticals. Whole liver was homogenized with a tissue grinder and approximately 10 mg of tissue was dissolved in QIAzol (Qiagen; 217061). RNA was extracted with miRNeasy 96 kit (Qiagen; 74004) following manufacturer instructions. RNA was reverse transcribed into complementary DNA (cDNA) using Taqman advanced fast polymerase chain reaction (PCR) mix (Thermo Scientific; 4374967). Real-time PCR was performed in technical duplicate on a Quantstudio quantitative PCR (qPCR) System (Thermo Scientific; A28135) using the Taqman advanced fast PCR mix (Thermo Fisher; 4444557). Target gene expression was normalized to housekeeping gene GAPDH, and fold change was calculated utilizing the ΔΔ-Ct method. Primers used are listed in Supplementary Table 3.

**IFN-γ levels in liver tissue (Enzyme-Linked Immunosorbent Assay; ELISA).** Liver tissue was weighed and homogenized in ice cold, sterile filtered 1× phosphate buffered saline (PBS), pH 7.4 (10 μL per mg tissue). Lysate was incubated on ice for 4 h and centrifuged 2500 × g for 5 min at 4 °C. Supernatant was collected and used directly. IFN-γ was measured in liver lysate following manufacturers instructions (DY485, R&D Systems, Minneapolis, MN). Briefly, Microplates (96 wells) were coated with working dilution of capture antibody overnight and blocked with 1% bovine serum albumin (BSA) diluted in sterile filtered 1× PBS pH 7.4. Samples and detection antibody were incubated for 2 h at room temperature with slight agitation. Working concentration of streptavidin-horse radish peroxidase (HRP) and color substrate mix (DY999, R&D Systems, Minneapolis, MN) were incubated at room temperature for 20 min with slight agitation. Plate was washed with 1× PBS pH 7. 4 with 0.05% Tween-20 after all steps except color substrate incubation. Product development was stopped with 1 N HCl and absorbance was measured at 450 and 540 nm on BioTek Synergy XTS Microplate Reader (Agilent Technologies, Santa Clara, CA). IFN-γ levels were calculated following manufacturer recommendation and converted into pg IFN-γ per mg of liver tissue.

**Magnetic resonance imaging (MRI) acquisition and analysis.** In vivo MRI was performed using a Bruker AV3HD 9.4 T/30 cm scanner, equipped with a BGA-12S HP gradient insert and a 40 mm transmit/receive coil, running ParaVision 6.0.1 (Bruker BioSpin, Billerica MA). Mice were anesthetized using 1–1.5% Isoflurane via a nose cone, with respiration continuously monitored and temperature maintained at 37 °C with warm air (SA Instruments, Stoney Brook, NY) during image acquisition. Following positioning and pilot scans, T2-weighted axial images of the liver were acquired using a Rapid Acquisition with Relaxation Enhancement (RARE) sequence, with the following parameters: Echo Time/Repetition Time (TE/TR) = 20/4000 ms, averages = 12, a field of view (FOV) of 36 × 24 mm, 192 × 128 matrix, 59 slices with a 0.5 mm slice thickness and a RARE factor = 8. The T2-weighted images were visualized using DSIstudio (https://dsi-studio.labsolver.org).

**Bulk RNA-sequencing and analysis pipeline.** Standard pipelines for transcriptome sequencing, quality control, data pre-processing, and mapping to mouse reference transcriptome were performed[28]. There were 5 different mouse whole transcriptome datasets analyzed in this study. First, RNA-seq analysis was performed on $n = 3$ PBS vehicle, $n = 3$ LNP-CTRL, and $n = 3$ LNP-CTNNB1 at 3 mg/kg dosage treated animals from the β-N model after 4 LNP dosages. The RNA-seq data for this analysis is deposited to Gene Expression Omnibus (GEO) under accession number: GSE290449. Second, RNA-seq analysis was performed on $n = 3$ LNP-CTRL and $n = 3–4$ LNP-CTNNB1 at 1 mg/kg dosage treated animals from each β-catenin-mutated HCC model (β-N & β-M models) after 3-days post-treatment. The RNA-seq data for this analysis is deposited to GEO under accession number: GSE270414. Third, RNA-seq analysis was performed on publicly available transcriptomic data from β-catenin-mutated HCC mouse models (GSE125336) and microarray gene expression data β-catenin knockout mouse livers (GSE68779). Fourth, RNA-seq was performed on liver tumors from β-M-pT3 ($n = 2$) and β-M-IRF2 ($n = 4$) animals. The RNA-seq data for this analysis is deposited to GEO under accession number: GSE270415. Fifth, RNA-seq was performed on liver tumors from β-M-pT3 ($n = 3$) and β-M-POU2F1 ($n = 3$) animals. The RNA-seq data for this analysis is deposited to GEO under accession number: GSE290444. For each of these different transcriptomic datasets, raw FASTQ files were first preprocessed by FastQC for quality control, Trimmomatic for filtering out low-quality reads[67], and STAR for reads alignment and gene count quantification[68]. After pre-processing data normalization was performed followed by principal component analysis (PCA). Then, based on the gene count data, differential gene expression analysis was performed using the R package DEseq2[69] to identify differentially expressed genes (DEGs). DEGs (upregulated and downregulated) were selected based on filtering criteria of absolute fold change ≥1.5 and FDR = 0.05. Volcano plots were used to depict upregulated and downregulated genes using EnhancedVolcano package in R. Gene set enrichment analysis (GSEA) with clusterProfiler R package using ranked genes was performed using Gene Ontology: Biological Processes (GO:BP) or Kyoto Encyclopedia of Genes and Genomes (KEGG) pathways from Molecular Signatures Database (MsigDB). To discover regulatory transcription factors (TFs) of common differentially expressed genes, an enrichment test was performed to select TFs with known downstream regulatory genes which overlapped with the common differentially expressed genes using the JASPAR database[70,71]. Heatmap of normalized expression values for genes was used to depict expression of individual genes between relative conditions. Waterfall plot was used to demonstrate positive or negative enrichment of certain pathways colored by $p$-value for each pathway.

**Whole liver perfusion for single-cell isolation.** Cells from the liver were isolated using a two-step collagenase perfusion[72]. Under anesthesia, a 27.5" catheter was used to circulate 0.3 mg/mL of collagenase II (Worthington, Lakewood, NJ) through the liver via the inferior vena cava. Livers were then collected in Dulbecco's modified Eagle's medium/F12 supplemented with 15 mM of HEPES (Corning, Corning, NY) + 5% fetal bovine serum (FBS; Biowest, Bradenton, FL). After manual digestion and agitation, cell suspensions were passed through a 100- then 70-μm cell strainer and washed twice using low-speed centrifugation (50 × $g$ for 5 min) to separate out nonparenchymal cells. Hepatocyte viability, determined by trypan blue staining, was typically >80%.

Hepatic lymphocytes and monocytes were enriched from the nonparenchymal fraction using a Percoll gradient combined with centrifugation[73]. Briefly, the nonparenchymal fraction from liver perfusions were centrifuged (805 × $g$ for 10 min) followed by 36% Percoll separation in Roswell Park Memorial Institute 1640 medium supplemented with +5% FBS. Erythrocytes were removed using Ammonium-Chloride-Potassium Lysing Buffer (Thermo Fisher Scientific). Isolated cells were counted for viability, which was typically >90%, using trypan blue.

**Fluorescence-activated cell sorting and gating strategy.** Enriched lymphocytes from liver perfusions were stained with fluorescent-conjugated antibodies (Supplementary Table 4) and a Fixable Dead Cell Stain (Thermo Fisher Scientific). Data was acquired using Cytek Aurora (Cytek Biosciences, Bethesda, MD) equipped with five lasers and analyzed using FlowJo software v10.1 (Treestar, Ashland, OR).

**Single-cell RNA-sequencing and analysis.** Hepatocytes isolated and purified from whole liver perfusions were sequenced along with the enriched nonparenchymal cell fraction (mostly lymphocytes/monocytes) were used for tandem scRNA-sequencing coupled with immune profiling separately. For the hepatocyte-enriched cell population sequenced (GEX; GSE270714), cells isolated from 2–3 individual animals were pooled together in each treatment condition and sequenced in separate wells on the flow cell. For the immune-enriched population (scRNA-seq with immune profiling; GSE270974), cells isolated from 6 individual animals (3 LNP-CTRL and 3 LNP-CTNNB1) were feature barcoded, hashed, and then pooled to be sequenced in one well on the flow cell. Specifically, the nonparenchymal cell fraction was feature barcoded and marked using TotalSeq™-C Mouse Universal Cocktail (Biolegend, San Diego, CA) following manufacturer's recommendations, along with BCR and TCR library preparation. At minimum 10,000 hepatocytes were sequenced (non-multiplexed wells) and 30,000 immune cells (multiplexed wells) were sequenced with at least 20,000 reads per cell in the non-multiplexed wells and 5000 reads per cell in the multiplexed wells. Multi-plexing was performed according to standard CellPlex protocols. Single-cell sequencing was performed on NovaSeq 2000 instrument and library prep with 5PrimeV2 was performed according to standard protocols. The raw FASTQ files for the pooled library were first processed using CellRanger for alignment against mm10 mouse genome using standard CellRanger multi pipeline with the 5p hashtag demultiplexing configuration file. Then, the demultiplexed aligned BAM files were re-formatted into FASTQ files per sample. Lastly, the CellRanger multi was applied to individual samples to align the data to mouse genome by integrating multi-omics data: gene expression, antibody capture, VDJ-T and VDJ-B. Ultimately, we generated the gene count matrices on a per-sample and per-cell basis for downstream analysis.

**Analysis pipeline.** We used standard Seurat[74] pipelines for quality control of the data at cell and gene level. We made sure to not include empty cells, doublets/multiplets, or dead/dying cells. This was determined by using nFeature_RNA > 500 & nFeature_RNA < 2000,

nCount_RNA ≥ 0 & nCount_RNA < 10,000, and percent.mt ≥ 0 & percent.mt <10 for analysis of the GEX dataset (GSE270714); and, nFeature_RNA > 200 & nFeature_RNA < 5000, nCount_RNA ≥ 0 & nCount_RNA < 12,000, and percent.mt ≥ 0 & percent.mt <10 for analysis of the scRNA-seq with immune profiling dataset (GSE270974). Quality control was visualized with violin plots of each of these parameters. After this step, we resulted in 26,851 high-quality cells from LNP-CTRL and 67,799 high-quality cells from LNP-CTNNB1 treatment for GEX dataset (GSE270714). Additionally, we resulted in 8499 high-quality cells from LNP-CTRL and 11,736 high-quality cells from LNP-CTNNB1 treatment for scRNA-seq with immune profiling dataset (GSE270974).

For each of the matrices, we integrated the LNP-CTRL and LNP-CTNNB1 datasets together using standard Seurat v4 pipeline of SelectIntegrationFeatures, PrepSCTIntegration, FindIntegrationAnchors, and IntegrateData using $n = 3000$ features. Then, dimensionality reduction method of principal component analysis (PCA) and uniform manifold approximation and projection (UMAP) was performed[75]. Initial cluster calls were generated using FindNeighbors and FindClusters function with dims = 1:30 and resolution = 0.2. Then, the dataset was swapped to RNA assay for visuals and variable features were identified, along with normalization and scaling of the data using FindVariableFeatures, NormalizeData, and ScaleData functions. Red blood cell contamination was inspected via Feature plots and Violin plots of *Hba-a1* and *Hbb-bs* genes. To identify specific cell types, present in each cluster, the FindAllMarkers function was used to determine differentially expressed genes per cluster compared to all other clusters. This identified 16 clusters in total for the hepatocyte-enriched dataset and 22 clusters in total for the immune cell-enriched dataset. We utilized the marker genes for different cell types present in Supplementary Table 5 and Supplementary Table 6, which is based on the literature and manual curation[36]. Clusters were then annotated and cell-type proportions per treatment condition were determined for each of the two datasets and visualized with either pie chart or stacked bar graphs.

Differential gene expression analysis, comparing each cluster to all other clusters, was performed on the integrated datasets for each cell type/cluster using the Seurat FindMarkers function to identify genes differentially expressed for each cell type irrespective of treatment condition. Differential gene expression within each cluster between treatment conditions was performed using the Seurat FindMarkers function and renaming the idents such that cells from each treatment condition were a different ident in the comparison. Differentially expressed genes (DEGs) were identified based on cut-off thresholds of adjusted *p*-value of <0.05 and absolute log2 Fold Change >0.25, or more stringent cutoff used in case of many DEGs. We utilized volcano plots to illustrate specific DEGs using the EnhancedVolcano R package with *p*-value cutoff of $p = 0.05$. Pathway enrichment analysis was performed using clusterProfiler R package using the DEGs identified and referencing GO:BP or KEGG pathways from MSigDB. Top pathways were visualized using dot plot function in clusterProfiler R package.

**Cell cycle regression analysis.** Cell cycle scoring was performed on the hepatocyte-enriched dataset using Seurat CellCycleScoring function. The expression matrix was retrieved from the source vignette and the S phase and G2M phase genes were extracted and converted to mouse orthologs from human genes. Cells that are not in S or G2M phase are classified as G1 phase. Pie charts were used to visualize distribution of percentage of cells in each phase of the cell cycle for each cluster.

**Pseudotime analysis.** Pseudotime trajectory analysis was performed on the integrated hepatocyte-enriched dataset following subsetting for only the normal hepatocyte or tumor hepatocyte clusters (Zone 3

CTNNB1 WT and MUT (GS+), Zone 1/2 CTNNB1 MUT (GS+), Zone 1 CTNNB1 WT (GS−), Reprogrammed Hepatocytes) using the monocle3 R package and standard pipelines[76]. First, a Cell Data Set (CDS) object was created, which transferred the UMAP cluster labels and cell metadata. Second, pseudotime analysis was performed and included calculations of potential cell trajectories using the published algorithm. The root node was chosen as the Zone 3 CTNNB1 WT and MUT (GS+) cell cluster. The results were visualized using the plot_cells function.

**Cell-cell interaction analysis.** Cell-to-cell interaction analysis on the immune cell-enriched population was performed using the CellChat V1 R package[37]. This package allows for analysis and inference of cell-to-cell communication using relative ligand-receptor expression. For visualization of specific communication pathways (i.e., ligand-receptor pairs), we used all the standard pipelines built in the CellChat R package[37].

**Single-cell spatial transcriptomics using Resolve Molecular Cartography™.** Mouse liver tissue was preserved frozen using OCT media following submersion in cold 2-methyl-butane. Frozen mouse liver tissue was sectioned with a cryostat (Leica, CM 1850-3-1) at a temperature of −17 °C. Tissue sections were 10 μm thick and placed on the Resolve Biosciences slide also cold at −17 °C. Following sectioning and placement, the slide was stored at −80 °C overnight, then shipped to Resolve BioSciences on dry ice for analysis. Probes were devised using Resolve BioSciences' proprietary algorithm[77]. The 100-gene panel used in this study is described in Supplementary Table 7. Resolve BioSciences performed the tissue fixation, 100-plex priming, hybridization, slide imaging, and fluorescent-tagging using their standard pipelines[77]. Then, regions of interest were imaged with fluorescent signals removed during a decolorization step to achieve a unique combinatorial code for each target gene in the panel[77]. The final images of the Molecular Cartography™ signals could then be viewed on Resolve BioSciences website which has a plugin tool for viewing the signals of each of the 100 genes at different magnification scales. We used tissue from one animal per LNP treatment group. Ultimately, 5-6 regions of interest (ROIs) were selected per tissue section for analysis. Single-cell spatial transcriptomic analysis was performed by quantifying gene counts per cell using automatic cell segmentation via QuPath software[78]. The libraries from each treatment condition (LNP-CTRL and LNP-CTNNB1) were then integrated together using the R package Seurat[74,79]. Analyzed cells included those filtered for greater than or equal to 10 gene counts per cell. Following these quality control steps, we performed unbiased clustering on all 100 genes using the dimensionality reduction method of principal component analysis (PCA) and uniform manifold approximation and projection (UMAP)[75]. Clusters corresponding to different cell types, including stellate, endothelial, and immune cells, and zones were identified and annotated from the UMAP based on expression of different landmark genes described in Supplementary Table 8[77]. The cluster proportions per treatment condition were depicted as pie charts or stacked bar plots.

Next, using Seurat, feature plots and violin plots were used to visualize gene expression across different clusters and between treatment conditions. Additionally, dot plots were used to visualize expression of specific genes within each cluster in terms of number of cells expressing that gene in each cluster and the expression level. Normalized expression values of each of the 100 genes were also visualized with heatmap across each of the different clusters. Lastly, once clusters were annotated using Supplementary Table 8 landmark genes, cells corresponding to these specific clusters were mapped back onto the virtual slide using SpatialPlot function to visualize spatial location of specific clusters using a different color for each cluster, which corresponded to the same cluster color on the UMAP plot. Moreover, pseudotime trajectory analysis to define cell

trajectories and cell states was performed on the integrated dataset using all clusters with the monocle3 R package[76]. The same pipelines for this were performed as described above in the single-cell RNA-sequencing dataset. The root node was chosen as the H1: Zone 3 CTNNB1 MUT (GS+) cell cluster. Lastly, within each ROI, tumor nodules were outlined based on the expression of *Glul* as a landmark gene for β-catenin-mutated tumor nodules. Then, gene counts and expression density (gene counts per area) were quantified within each tumor boundary and averaged across the defined regions (tumor or non-tumor). This was also performed on a cell basis to determine cell density per region (tumor or non-tumor). The single-cell spatial transcriptomic data from this analysis is deposited to Gene Expression Omnibus (GEO) under accession number: GSE270708.

**10× visium spatial transcriptomics.** Tissue sections of 10 μm thickness were cut from mouse liver tumor FFPE blocks and were deparaffinized, H&E stained, then decrosslinked using standard 10× genomics protocols. Regions of interest (ROIs) were selected by outlining ~6.5 mm$^2$ areas. Standard Visium CytAssist protocols were used for FFPE tissue for generation of sequence ready libraries. Quality control parameters were a DV200% ≥30%. For 10× Visium spatial transcriptomics bioinformatic analysis, sequencing reads from 55 μm tissue regions (spots) were first preprocessed by 10× Space Ranger software for alignment and spatial gene count quantification. The processed data were then imported into R using the package Seurat[74]. For the β-M model, spots across 4 samples (β-M LNP-CTRL [BL-131], 2 β-M LNP-CTNNB1 NR [BL-146, BL-147], and 1 β-M LNP-CTNNB1 R [BL-152]) were normalized and integrated. For the β-N model, spots across 5 samples (β-N LNP-CTRL [BL-551], 2 β-N LNP-CTNNB1 NR [BL-584, BL-587], and 2 β-N LNP-CTNNB1 R [BL-585, BL-586]) were normalized and integrated. For both datasets, following normalization and integration, dimensionality reduction via principal component analysis and uniform manifold approximation was performed. K-means clustering identified 17 clusters in the β-M model and 13 clusters in the β-N model, which were overlaid on hematoxylin and eosin sections using the SpatialPlot function in Seurat. Differential gene expression was used to identify up- and downregulated genes in each cluster using the FindAllMarkers function in Seurat with return.thresh = 0.05. We utilized volcano plots to illustrate specific DEGs (defined by adjusted *p*-value < 10e-32 and absolute log2FC > 0.5) using the EnhancedVolcano R package. Cell-type composition in each cluster was inferred based on the expression of marker genes in Supplementary Table 9 from literature and manual curation[36]. Marker genes were visualized in dot plot format across clusters and by treatment condition. Gene set enrichment analysis was performed with the clusterProfiler R package[80] using ordered ranked gene lists with gene sets from GO:BP from MSigDB collection[81], with a *p*-value threshold of 0.05. Pathway enrichment analysis was performed with the clusterProfiler R package[80] using DEGs (defined by adjusted *p*-value < 0.05 and absolute log2FC > 0.25) and gene sets from GO:BP and KEGG from MSigDB. Upstream regulator analysis was performed using Qiagen software with the DEGs from each cluster. Cell cycle regression and pseudotime analysis were performed as described in the "Single-cell RNA-sequencing and analysis" section. Version 2 of the CellChat R package[82] was used to determine probabilities of spatial cell-cell communication between clusters, based on expression of ligands and receptors in spatially proximal spots. The 10× Visium transcriptomic data from this analysis is deposited to Gene Expression Omnibus (GEO) under accession number GSE270997 (subseries: GSE270975 and GSE290445).

**TCGA-LIHC human HCC data mining.** The Cancer Genome Atlas (TCGA) RNA-seq whole exome sequencing and transcriptomic data were downloaded from Genomic Data Commons (GDC) through the R

Bioconductor package GenomicDataCommons. We selected patients from TCGA with Liver Hepatocellular Carcinoma (TCGA-LIHC) for analysis. This dataset contains 424 cases with 50 of these being adjacent normal liver tissue. Raw gene counts were normalized and used for downstream analyses. We identified that 98 out of the 374 TCGA-LIHC patients had *CTNNB1* mutations, 18 had *AXIN1* mutations, and 3 had *APC* mutations. Differential gene expression was performed to identify pathways differentially regulated between *CTNNB1*-mutated and normal patients and *CTNNB1*-mutated and *CTNNB1*-wild-type patients using gene set enrichment analysis with GO_BP pathways from MsigDB. IRF2 and POU2F1 target genes plotted in the heatmap are provided in Supplementary Table 10. The 33 IRF2 and 10 POU2F1 target genes which were intersected for common overlap and plotted (39 common genes). The 39 genes identified were inferred IRF2/POU2F1 target genes retrieved from https://maayanlab.cloud/chea3/ and which overlapped with the 162 common genes which were upregulated in β-catenin knockout livers and downregulated in β-catenin-mutated HCC mouse livers. A heatmap of z-scored values was generated to overlap patients with and without tumor, and with and without *CTNNB1, AXIN1*, or *APC* mutations, and the relative normalized expression scores of IRF2/POU2F1 target genes. Boxplots were used to visualize expression of *IRF2, POU2F1*, or a module score of the IRF2/POU2F1 target genes. Module score was calculated based on normalized mean expression of the common 39 target genes. Survival data was downloaded directly from cBioPortal website and integrated with the expression data.

**IMbrave150 phase 3 retrospective clinical trial analysis.** We retrospectively analyzed genomic and transcriptomic data (Whole Exome Sequencing and RNA-seq data) from the IMbrave150 trial[6] and associated this data with *CTNNB1* mutational status, tertiary lymphoid structure/lymphoid aggregate (TLS/LA)-like signature expression score, TLS/LA/diffuse infiltrate (DI) presence, and clinical parameters (overall and progression-free survival [OS/PFS] and clinical response using mRECIST criteria). First, TLS/LA/DI/ presence was assessed by Genentech Pathologist (author H.K.) and associated with clinical response based on mRECIST criteria (CR/PR, SD/PD, or NA)[6]. Second, TLS/LA/DI presence was associated with PFS and OS in each treatment arm (atezolizumab plus bevacizumab or sorafenib). Third, a previously described B and CD4 Tconv and CD4 Tfh cell signature expression score was merged and overlapped with genes sequenced in TCGA-LIHC, resulting in 140 genes, which was associated with TLS/LA/DI presence, clinical response based on mRECIST criteria, and *CTNNB1* mutational status[45]. The source data has been previously deposited in the European Genome-Phenome Archive under accession no. EGAS00001005503.

**Public single-cell human and mouse liver analysis.** Single-cell data from both human and mouse livers were accessed from the LiverCellAtlas website directly (https://www.livercellatlas.org/). The human data was accessed here and images of the UMAP plots were taken (https://www.livercellatlas.org/umap-humanAll.php). The mouse data was accessed here and images of the UMAP plots were taken (https://www.livercellatlas.org/umap-ststmouseAll.php). Expression of *IRF2/Irf2* and *POU2F1/Pou2f1* were queried in these datasets to investigate cell-type specific expression of these transcription factors in normal human/mouse liver.

**Statistical analysis.** The data presented throughout the manuscript is represented as mean ± standard deviation (SD) for each bar plot. The indicated statistical tests were performed in either R or Prism 10.3.1 software (GraphPad Software Inc., Boston, Massachusetts, USA, www.graphpad.com). For our study, $P < 0.05$ was considered statistically significant ($*p < 0.05$, $**p < 0.01$, $***p < 0.001$, $****p < 0.0001$).

## Reporting summary
Further information on research design is available in the Nature Portfolio Reporting Summary linked to this article.

## Data availability
For high-throughput multi-omics data, all the raw and processed files are uploaded to Gene Expression Omnibus (GEO) with accession ID GSE270977. The bulk RNA-seq CTNNB1/NFE2L2 data, CTNNB1/hMet/NFE2L2 data, CTNNB1/MET/IRF2 data, and CTNNB1/NRF2/POU2F1 data can be downloaded by GSE290449, GSE270414, GSE270415, and GSE290444, respectively; the single-cell spatial transcriptomics data by Resolve Biosciences Molecular Cartography can be accessed by GSE270708; the single-cell RNA-seq GEX and the single-cell coupled with hashtag immune profiling data can be downloaded by GSE270714 and GSE270974, respectively; and, the spatial transcriptomics data by 10× Visium platform can accessed by GSE270975 (β-M model) and GSE290445 (β-N model). All clinical, raw RNA-seq and WES data for the IMbrave150 trial are deposited in the European Genome-Phenome Archive under accession no. EGAS00001005503. Qualified researchers may request access to individual patient-level data through the clinical study data request platform (https://vivli.org/). The remaining data are available within the Source Data file. All key resources used are provided in Supplementary Table 11. Source data are provided with this paper.

## Code availability
No custom code was generated for this manuscript.

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

## Acknowledgements

This work was supported by NIH grants R01CA251155, R01CA250227, R01DK062277, and SVC Endowed Chair in Pathobiology and Therapeutics to S.P.M. This work was also supported in part by Sponsored Research Agreement to S.P.M. by Alnylam Pharmaceuticals. This work was also funded in part by T32EB001026 to B.M.L. and T.Y. This work was also funded in part by F30CA284540 to B.M.L. This work was also supported in part by the University of Pittsburgh Center for Research Computing through the resources provided and by NIH grant P30DK120531 to Pittsburgh Liver Research Center (PLRC) for services provided by the Genomics and Systems Biology Core. This work was also supported in part by UPMC Hillman Cancer Center Core grants P30CA047904 and UM1CA186690 to J.J.L.

## Author contributions

Conceptualization: B.M.L. and S.P.M.; Methodology: B.M.L., E.R.D., T.M.Y., S.L., M.T., M.M., M.R.E., Y.W., W.B., J.T. and S.P.M.; Software: B.M.L., T.M.Y., S.L. and J.-J.L.; Formal Analysis: B.M.L., E.R.D., T.M.Y., S.L., X.G., H.K., T.D., Y.W., W.B., J.T. and S.P.M.; Investigation: B.M.L., E.R.D., T.M.Y., S.L., C.C., Y.L., M.T., X.G., H.K., S.S., V.M., J.-J.L., A.S-V., Y.K., M.P., T.K.H., L.M.F., B.L., A.R., R.P.R., P.P., M.R., A.B., R.R., T.D., M.R.E., J.T. and S.P.M.; Resources: T.D., E.G., M.R.E., X.C., M.M., Y.W., W.B., J.T. and S.P.M.; Writing—Original Draft: B.M.L.; Writing—Review and Editing: T.D., J.J.L., A.L., X.C., M.M., Y.W., W.B. and S.P.M.; Visualization: B.M.L., E.R.D., T.M.Y., S.L. and M.T.; Supervision: J.T. and S.P.M.; Project Administration: S.P.M.; Funding Acquisition: B.M.L. and S.P.M.

## Competing interests

S.P.M. has received research grants from Alnylam Pharmaceuticals. He also received funding from Fog Pharmaceuticals and is a consultant on Advisory Boards for Surrozen, AntlerA, Alnylam, Mermaid Bio, Vicero Inc., and UbiquiTx, and there is no pertinent conflict of interest of these entities as relevant to the current manuscript. T.D., M.M., and W.B. are employed by Alnylam Pharmaceuticals, Cambridge, MA. X.G., H.K., and Y.G. are employed by Genentech Inc., San Francisco, CA. No other authors have any relevant conflicts of interests to declare regarding the current study.

## Additional information

Brandon M. Lehrich[1,2,3,4], Evan R. Delgado[3,5], Tyler M. Yasaka [1,2,3,4], Silvia Liu[1,2,3], Catherine Cao[1], Yuqing Liu[1], Mohammad N. Taheri[3,5,6], Xiangnan Guan[7], Hartmut Koeppen[7], Sucha Singh[1,2], Vik Meadows[1,2,3], Jia-Jun Liu[1,2,3], Anya Singh-Varma[1], Yekaterina Krutsenko[1], Minakshi Poddar[1,2], T. Kevin Hitchens[8], Lesley M. Foley [8], Binyong Liang[9], Alex Rialdi[10], Ravi P. Rai[1,2,3], Panari Patel[1,2], Madeline Riley[1,2], Aaron Bell [1,2,3], Reben Raeman[1,2,3], Tulin Dadali[11], Jason J. Luke [12], Ernesto Guccione [10], Mo R. Ebrahimkhani [3,5,6], Amaia Lujambio [10], Xin Chen [13], Martin Maier [11], Yulei Wang [7], Wendy Broom[11], Junyan Tao [1,2,3,15] ✉ & Satdarshan P. Monga [1,2,3,14,15] ✉

[1]Organ Pathobiology and Therapeutics Institute, University of Pittsburgh School of Medicine, Pittsburgh, PA, USA. [2]Department of Pharmacology and Chemical Biology, University of Pittsburgh School of Medicine, Pittsburgh, PA, USA. [3]Pittsburgh Liver Research Center, University of Pittsburgh and University of Pittsburgh Medical Center, Pittsburgh, PA, USA. [4]Medical Scientist Training Program, University of Pittsburgh, Pittsburgh, PA, USA. [5]Department of Pathology, Division of Experimental Pathology, University of Pittsburgh School of Medicine, Pittsburgh, PA, USA. [6]Department of Bioengineering, Swanson School of Engineering, University of Pittsburgh, Pittsburgh, PA, USA. [7]Translational Medicine, Genentech Inc., San Francisco, CA, USA. [8]University of Pittsburgh School of Medicine, Pittsburgh, PA, USA. [9]Hepatic Surgery Center, Department of Surgery, Tongji Hospital, Tongji Medical College, Huazhong University of Science and Technology, Wuhan, China. [10]Department of Oncological Sciences, Icahn School of Medicine at Mount Sinai, New York, NY, USA. [11]Alnylam Pharmaceuticals, Boston, MA, USA. [12]UPMC Hillman Cancer Center and University of Pittsburgh, Pittsburgh, PA, USA. [13]Cancer Biology Program, University of Hawaii Cancer Center, Honolulu, HI, USA. [14]Division of Gastroenterology, Hepatology and Nutrition, Department of Medicine, University of Pittsburgh School of Medicine, Pittsburgh, PA, USA. [15]These authors contributed equally: Junyan Tao, Satdarshan P. Monga. ✉e-mail: junyantao2010@gmail.com; smonga@pitt.edu

