## [Transparent Peer Review file · Nature Communications]

Precision targeting of β -catenin induces tumor reprogramming and immunity in hepatocellular cancers

Corresponding Author: Dr Satdarshan Monga

Version 0:

Reviewer comments:

Reviewer #1

(Remarks to the Author)

The study provides strong preclinical evidence for targeting β -catenin using LNP-siRNA (LNP-CTNNB1) in hepatocellular carcinoma (HCC), showcasing efficacy in reprogramming tumor microenvironment and enhancing immune responses. The study leverages advanced spatial and single-cell transcriptomic analyses to unravel molecular mechanisms of immune exclusion. The work suggests that β -catenin-targeting therapies could enhance responses to immune checkpoint inhibitors (ICI).

The authors use multiple elegant immunocompetent murine HCC models across both early and advanced disease stages and show that treatment with lipid nanoparticles (LNP) loaded with siRNA for CTNNB1 demonstrates promising efficacy. Transcriptomic analysis identified IRF2 and POU2F1 as key regulatory transcription factors suppressed by mutated β -catenin. The study further reveals that LNP-CTNNB1 treatment induces HCC cell reprogramming and remodels the tumor immune microenvironment. Notably, LNP-CTNNB1 also exhibits synergistic efficacy when combined with anti-PD-1 immunotherapy, which is in line with the inverse correlation between CTNNB1 mutation and lymphoid aggregates. This is a timely, extensive, and compelling study. I have several comments.

1. Did the authors observe TLS/LA in the murine HCC models following LNP-CTNNB1 treatment? If so, does the treatment affect the number of TLS/LA in these models? Given that IMbrave150 data indicates an inverse correlation between CTNNB1 mutation and lymphoid aggregates, it would be interesting to speculate on the mechanism of lymphoid aggregate formation.
2. Some more details about the LNPs, and data on PK, biodistribution and tumor accumulation would enrich the report. The information provided is very limited.
3. P9 lines 199-201. What is the rationale for the duration between the injection of HDTVi and the start of LNP treatment?
4. P8 lines 175-176. Were there any adverse effects of LNP-CTNNB1 outside the liver, such as in the lungs? The toxicity data is also limited, and may be a concern.
5. P14. Is the cluster mapping in LNP-CTNNB1 treated tumors consistent with P11 lines 243-246?
6. The paper asserts that their approach overcomes ICI resistance. The claim lacks direct evidence and should be revised to focus on enhancing ICI response rather than overcoming resistance.
7. The variability in late-stage HCC responses suggests an unclear heterogeneity in the models or experimental conditions. This requires discussion or additional analyses.
8. It would be helpful to include more data on the durability of tumor responses and immune effects post-treatment cessation.
9. The clinical trial data is extremely valuable and interesting. However, as presented, the B-cell data do not really connect well with the preclinical data.

Minor:

Line 132: "at 20nm concentration" should be corrected to "at 20nM concentration."

P14 line 310. Figure 3c, not S3c.

P17 lines 386-388. Presenting the expression of IFN γ or IFN γ receptors at the protein level in T cells or macrophages would provide stronger evidence.

P17 lines 393-394. Did the authors evaluate the increase of M1-like macrophages in tumors in b-M mice treated with IFN γ ?

P19 lines 437-439. Was there any particular lymphocyte distribution apparent in the tumors (center or edge)?

Reviewer #2

(Remarks to the Author)

CTNNB1 mutation and pathway activation is one of the major reasons for the development/progression and immunotherapy resistance of HCC. In this manuscript, the authors demonstrated the potent efficacy and potential mechanistic interpretation of a novel siRNA encapsulated in lipid nanoparticle targeting CTNNB1 (LNP-CTNNB1) in multiple preclinical HCC models. Based on single-cell and spatial transcriptomics data, they demonstrated that reprogramming of immune regulatory transcription factors IRF2 and POU2F1 in CTNNB1-mutated tumors, which is associated with type I/II interferon signaling, and alterations in both innate and adaptive immune responses, may be responsible for the therapeutic efficacy. The study is interesting, however, more data and explanation are needed to support their conclusion.

Major comments:

1. Figure 1: What adverse effects of LNP-CTNNB1 have been tested? The overall and organ weights measurement as well as the histology by H&E staining after LNP-CTNNB1 treatment to evaluate the damage of different organs should be performed.
2. Did LNP-CTNNB1 affect survival of the mice in the CTNNB1-mutated model?
3. Figure 4: The authors suggested that 'increased type I/II interferons released from the immune compartment (likely from T cells and macrophages) following LNP-CTNNB1 treatment are engaging with macrophages in the TIME milieu, and in part contributing towards polarizing them towards a pro-inflammatory anti-tumor phenotype'. While they provided evidence by IFN γ treatment (Fig 4h-j), they need to show that the IFN γ level was indeed altered in the LNP-CTNNB1 compared to LNP-CTRL treated tissues by ELISA. Moreover, after IFN γ treatment, was there change in M1/M2 macrophage proportion as anticipated?
4. Figure 4: The TME characterization by single-cell transcriptomic analysis, especially the changes in M1/M2-like macrophages in the LNP-CTNNB1 treated tumors should be validated by flow cytometry. Would modulation of these macrophages reverse the antitumor effects of LNP-CTNNB1 treatment?
5. Figure 5: What is the molecular mechanism by which mutated- β -catenin suppresses the expressions of IRF2 and POU2F1? This key link is missing in this study.
6. Is there a correlation between CTNNB1 mutation/expression and IRF2/POU2F1 expression in HCC patients?
7. Figure 7m: The cell cycle changes in cancer cells shown in the schematic diagram after LNP-CTNNB1 treatment should be validated by flow cytometry.
8. Figure 8f: "Increased Bsig expression was also observed in atezolizumab plus bevacizumab arm in patients with CR/PR and SD, while decreased Bsig expression was observed in those with PD (Figure 8f)." However, there is no obvious change in B cells upon LNP-CTNNB1 treatment as shown in Figure 4b. Thus the mouse study and human data is not concordant. No difference in TLS/LA has been shown between MUT and WT patients. The importance of B cells and TLS formation in CTNNB1 mutation is not adequately supported. Can the authors provide more direct evidence demonstrating that "formation for TLS/LA may be restricted by mutated β -catenin" (line 555-556)?
9. Summary: "Lastly, CTNNB1-mutated patients treated with atezolizumab plus bevacizumab combination had decreased presence of lymphoid aggregates, which were prognostic for response and survival." No data is found to support this statement in this manuscript.

Minor comments:

1. Figure 1b: Quantification should be performed to indicate LNP-CTNNB1 efficacy in mutant-CTNNB1 human HCC organoid cultures.
2. Tumor burden should be evaluated by the number and diameter of tumor nodules in H&E staining in addition to LW/BW.
3. Figure 7i: What is the mRNA level of CTNNB1 after single anti-PD-1 treatment?

Reviewer #3

(Remarks to the Author)

The manuscript entitled "Precision targeting of β -catenin induces tumor reprogramming and immunity in hepatocellular cancers" reported that LNP delivering si CTNNB1 is effective in treating HCC with CTNNB1 mutations by affecting intrinsic signaling within tumor cells and remodeling tumor immune surveillance. β -Catenin, considered an undruggable target, is encoded by the CTNNB1 gene, which undergoes mutations across various cancers with mutation sites differing among patients. Using a single siRNA to target multiple mutation sites is unlikely to be effective, potentially posing significant challenges to the clinical translation of this project. Although LNP delivering si CTNNB1 showed certain advantages, this work still has shortcomings in terms of novelty and solidity.

Major Comments:

1. Novelty and Conceptual Framework: The novelty and conceptual framework of the study are somewhat limited, particularly regarding material selection. The effects of CTNNB1 siRNA on hepatocellular carcinoma (HCC) have been explored in previous studies. Additionally, the lipid nanoparticle (LNP) composition and synthesis methods employed here are relatively conventional. This raises concerns regarding the originality of the approach and its potential for advancing therapeutic strategies.
2. Evaluation of LNP Properties: The manuscript provides insufficient characterization of the LNP system employed. Detailed information on the composition and physicochemical properties of the LNPs, such as particle size, charge, and stability, is lacking. For instance, the particle size of LNP-siRNA is reported as 61.3 nm, but it is unclear whether this measurement is based on number-based or intensity-based distribution. Furthermore, the stability of the LNP system under

storage conditions should be evaluated to ensure its viability for clinical translation.

3. **Statistical Rigor:** Several experiments, including those depicted in Figure 3a, suffer from inadequate sample sizes. Specifically, the LNP-CTRL group (n=2) and LNP-CTNNB1 group (n=3) are insufficient to yield reliable conclusions, particularly given that the observed cell volume differences between the two groups are substantial (up to 2.5-fold). A more robust statistical design, with increased sample sizes and uniform group numbers, is recommended to enhance the reliability of the findings.
4. **Quantification of Delivery Efficiency and Specificity:** While the study demonstrates the anti-tumor effects of LNP-CTNNB1 in vivo, the delivery efficiency and cell specificity of the LNP system have not been sufficiently quantified. The manuscript would benefit from a more thorough analysis of the distribution of LNP-CTNNB1 particles in various tissues and tumors post-administration. This is essential for evaluating the therapeutic potential and specificity of the delivery system.
5. **Experimental Rationale in Figure 1b:** The rationale for selecting a 20 nM concentration of LNP-CTNNB1 and a 72-hour incubation period in Figure 1b remains unclear. The authors should provide a more detailed justification for these experimental conditions, citing relevant studies or experimental data to support the chosen dose and time points.
6. **Tumor Resistance Mechanism:** In Figure 6, the manuscript mentions a tumor model that did not respond to treatment, but the underlying mechanism of resistance remains unexplored. To gain a deeper understanding of this phenomenon, it is recommended that single-cell RNA sequencing and immunohistochemical analysis be conducted on the resistant models. These approaches may provide insights into the molecular mechanisms responsible for treatment failure.
7. **Role of IRF2 and POU2F1:** The finding that overexpression of IRF2 and POU2F1 in the β -catenin mutant model leads to a significant reduction in liver tumor growth suggests that these factors play critical roles in β -catenin signaling. However, the mechanisms by which IRF2 and POU2F1 influence β -catenin signaling in HCC require further investigation. More detailed studies are needed to elucidate how these factors modulate liver growth and tumorigenesis.
8. **Sample Size in Omics Analysis:** Although extensive omics analyses were employed to examine the effects of RNAi-mediated β -catenin inhibition on intrinsic signaling and immune surveillance in HCC, the number of animal samples analyzed (n=2,3,4) is relatively small. To enhance the statistical power and reliability of the results, a larger sample size should be used, and the study should consider additional validation at both the cellular and molecular levels.
9. **Clarity of Images:** Several figures in the manuscript, such as Figure 1b, Figure 2e, and Figure 5m, lack sufficient resolution, which hampers the ability to discern the significant differences highlighted in the text. Retaking these images with higher clarity is recommended to ensure that the key findings are clearly visible and well-supported by the visual data.
10. **Lack of Citation for Statement on RNAi:** The manuscript states that "RNAi-mediated gene silencing has proven to be an excellent tool for targeting the traditionally 'undruggable,' especially in hepatic tissue," but no citation is provided to support this assertion. It is crucial to include appropriate references to substantiate this claim and place the statement within the broader context of RNAi-based therapeutics.

Reviewer #4

(Remarks to the Author)

The manuscript tries to address a relevant topic in hepatocellular carcinoma (HCC), focusing on the role of β -catenin mutations in immune exclusion and tumor progression. The authors investigate the effects of targeting β -catenin mutations with a siRNA encapsulated in lipid nanoparticles (LNP-CTNNB1) in HCC mouse models, with a particular emphasis on immune modulation and tumor response. The study aims to better understand the immune microenvironment in HCC and explore therapeutic strategies to overcome immune exclusion. However, several aspects of the study and its design raise concerns about the rationality and clinical applicability of the findings.

1. The author used siRNA to target the mutant β -catenin in a humanized hydrodynamic HCC mouse model. This approach only inhibits tumor intrinsic β -catenin signal. However, the tumorigenic effects of β -catenin signaling in human cancers also involve immune suppression via exogenous factors like TGF- β . In the current mouse model, the partial humanization prevents the siRNA from directly impacting immune cells, which weakens the translation of the results to human conditions. Thus, this model does not fully reflect the complexity of the immune microenvironment in human HCC. This limited the design of the study.
2. The mouse model employed here is not an established HCC treatment model but rather an intervention model for cancer development. The method does not align with the intended purpose of evaluating therapeutic strategies in established HCC as the author claimed. This discrepancy undermines the study's design and its relevance to clinical HCC treatment.
3. The author hypothesizes that β -catenin mutations lead to immune exclusion through the inhibition of IRF2 and POU2F1, but this claim lacks clear experimental validation. (1) Figure 5c shows heterogeneity in IRF2 and POU2F1 expression across liver cells. The manuscript does not explore whether there is a correlation between the expression of these factors and β -catenin mutations. It would be crucial to establish whether high or low expression of IRF2 and POU2F1 correlates with β -catenin mutation status, which is not clearly addressed. (2) In Figure 5M, the infiltration of lymphocytes in tumors overexpressing POU2F1 appears to form clusters at certain boundaries, which seems inconsistent with the manuscript's hypothesis of throughout chemokine upregulation. This pathological structure of this location more closely resembles the gaps between different small cancer lesions during tumor formation, different from those in control groups. (3) The manuscript has not confirmed whether POU2F1 overexpression impacts T cell function or activity at the protein level. (4) Additionally, no results regarding the overexpression of IRF2 are shown, which is essential to strengthen the claims made in the study.
4. While RNA interference (RNAi) technology has been approved for clinical use, the lipid nanoparticle (LNP) delivery system employed in this study remains a preclinical tool and is not yet mature enough for clinical application. The lack of clinical applicability of this delivery system significantly limits the practical relevance of the findings.
5. Figure 1b shows data of a human organoid model for one experiment. More replicates would be necessary to validate the findings.

Version 1:

Reviewer comments:

Reviewer #1

(Remarks to the Author)

The authors have responded to all the issues raised. I have no additional comments.

Reviewer #2

(Remarks to the Author)

The revised manuscript has addressed most of our questions. However, the mechanism by which CTNNB1 influences IRF2/POU2F1 has not been clearly elucidated, and the authors have chosen not to address this aspect through supplemental experiments in the current study. This likely represents the most significant limitation of this work.

Reviewer #3

(Remarks to the Author)

The authors present a well-executed study employing a clinically relevant CTNNB1-mutant HCC model to investigate β -catenin targeting via siRNA encapsulated in lipid nanoparticles (synthesized from commercial company). Through multi-omics analyses, the study provides insights into the reprogramming of tumor cells and microenvironments by β -catenin and examines the impact of its inhibition on tumor biology and therapeutic strategies. However, the overall novelty and conceptual advancement of this work in both material design and target selection are very limited. The oncogenic role of CTNNB1 mutations and therapeutic potential of β -catenin inhibition in HCC have been extensively documented in prior literatures (e.g., Nat Commun. 2022; Cell Death Dis. 2023; Curr Cancer Drug Targets. 2013). While the combination strategy of β -catenin inhibition with anti-PD-L1/CTLA-4 therapies is scientifically sound, it primarily constitutes an incremental extension of existing paradigms rather than a transformative conceptual advance. The study does not provide fundamentally new biological insights or demonstrate a clear therapeutic advantage beyond what has already been reported in the context of β -catenin modulation in HCC immunotherapy. Given these considerations, the findings may be better suited for a more specialized journal rather than Nature Communications.

RESPONSE TO REVIEWER' COMMENTS

Reviewer #1 (Remarks to the Author): with expertise in HCC, cancer immunology

The study provides strong preclinical evidence for targeting β -catenin using LNP-siRNA (LNP-CTNNB1) in hepatocellular carcinoma (HCC), showcasing efficacy in reprogramming tumor microenvironment and enhancing immune responses. The study leverages advanced spatial and single-cell transcriptomic analyses to unravel molecular mechanisms of immune exclusion. The work suggests that β -catenin-targeting therapies could enhance responses to immune checkpoint inhibitors (ICI).

The authors use multiple elegant immunocompetent murine HCC models across both early and advanced disease stages and show that treatment with lipid nanoparticles (LNP) loaded with siRNA for CTNNB1 demonstrates promising efficacy. Transcriptomic analysis identified IRF2 and POU2F1 as key regulatory transcription factors suppressed by mutated β -catenin. The study further reveals that LNP-CTNNB1 treatment induces HCC cell reprogramming and remodels the tumor immune microenvironment. Notably, LNP-CTNNB1 also exhibits synergistic efficacy when combined with anti-PD-1 immunotherapy, which is in line with the inverse correlation between CTNNB1 mutation and lymphoid aggregates. This is a timely, extensive, and compelling study. I have several comments.

Response: We thank the reviewer for their comment. The reviewer well summarized the main points of our “extensive” studies asserting the “strong preclinical evidence” for testing efficacy of LNP-siRNA and demonstrating reprogramming of the tumor microenvironment and enhanced innate and adaptive immunity. We also utilized “advanced spatial and single-cell transcriptomic analyses” to investigate underlying mechanisms. We have described the major revisions to the manuscript based on the reviewer comments below.

Major comments:

Point 1. Did the authors observe TLS/LA in the murine HCC models following LNP-CTNNB1 treatment? If so, does the treatment affect the number of TLS/LA in these models? Given that IMbrave150 data indicates an inverse correlation between CTNNB1 mutation and lymphoid aggregates, it would be interesting to speculate on the mechanism of lymphoid aggregate formation.

Response: We thank the reviewer for their comment. We had quantified the number of GZMB+ lymphoid aggregates (Fig.7k), which is known to be expressed at higher levels in TLS/LA. We observed a trending increase in the quantity of GZMB+ LA in LNP-CTNNB1 + anti-PD1 compared to LNP-CTNNB1 + IgG treated mice. We have more clearly outlined this rationale in the results text of the manuscript. Additionally, we suspect that these lymphoid aggregates form in response to upregulation of various signaling molecules (e.g., chemokines, cytokines) that may be downstream of transcription factors (TFs), such as IRF2 or POU2F1, that are active when β -catenin is suppressed. We have ongoing work in the lab

in this area re-expressing various chemokines downstream of mutant- β -catenin. We collaboratively reported on CCL5 re-expression in β -catenin-Myc model [ref1], which enhanced T cell activity, although we did not quantify presence of lymphoid aggregates. We have extensively edited the discussion section describing role of β -catenin in immune surveillance extensively, with some commentary on this hypothesis, supported by previous literature, but which requires further mechanistic investigations. Overall, in HCC, we posit that β -catenin is directly influencing expression of essential cytokines/chemokines necessary for immune recruitment and function through modulation of an immune-regulatory module of key TFs which are normally active when β -catenin is absent from the nucleus. Previous work has described sequestration and inhibition of p65 (a subunit of NF- κ B) by β -catenin directly leading to reduced binding, target gene expression, and inflammation [refs 2-4]. Thus, here we have described similar findings, yet with 2 additional TFs, IRF2 and POU2F1. Further ongoing work in the lab is aimed to determine how β -catenin interacts with both, which chemokines downstream of IRF2 and POU2F1 are essential for immune response in HCC, and which immune subsets are most affected by re-engagement of these TFs. This is a follow-up study in itself. Here, we describe that efficacy of LNP-CTNNB1 is mediated in part by promoting re-engagement of these TFs and target gene expression.

References

1. Ruiz de Galarreta M, Bresnahan E, Molina-Sanchez P, et al. beta-Catenin Activation Promotes Immune Escape and Resistance to Anti-PD-1 Therapy in Hepatocellular Carcinoma. *Cancer Discov.* 2019;9(8):1124-1141.
2. Du Q, Zhang X, Cardinal J, et al. Wnt/beta-catenin signaling regulates cytokine-induced human inducible nitric oxide synthase expression by inhibiting nuclear factor-kappaB activation in cancer cells. *Cancer Res.* 2009;69(9):3764-3771.
3. Deng J, Miller SA, Wang HY, et al. beta-catenin interacts with and inhibits NF-kappa B in human colon and breast cancer. *Cancer Cell.* 2002;2(4):323-334.
4. Nejak-Bowen K, Kikuchi A, Monga SP. Beta-catenin-NF-kappaB interactions in murine hepatocytes: a complex to die for. *Hepatology.* 2013;57(2):763-774.

Point 2. Some more details about the LNPs, and data on PK, biodistribution and tumor accumulation would enrich the report. The information provided is very limited.

Response: We thank the reviewer for their comment. In regard to the LNP characterization, in the Methods section under 'LNP-siRNA synthesis' we have currently referenced manuscripts which have described the synthesis and formulation of the LNP. We have also included the siRNA sequences for the siControl and siCTNNB1. The LNP formulation we are using is biosimilar to what is currently being used for treating patients with hereditary amyloidosis (Onpattro/patisiran) and is manufactured by Alnylam Pharmaceuticals. The LNP used in this study is in Ph1 clinical trial (NCI-2024-08329/NCT06600321) currently and we have focused this manuscript mostly on therapeutic efficacy of this agent which supports the ongoing effort to target β -catenin-mutated HCC. We have included the following statement in the Methods section:

“The LNPs used in this study size range from 60-80nm with siRNA encapsulation efficiency of ~95% (PDI = 0.045) with characterization and quality control performed as previously described [ref 1]. LNP formulations from Alnylam have demonstrated a rapid accumulation of the parent lipids in the liver and plasma with elimination from plasma by 8-hours, liver by 48-hours, and spleen by 96-hours post systemic administration via tail-vein injection [ref 1]. Biodistribution of the LNPs have also been described to accumulate primarily in the liver and spleen [ref 1].”

Moreover, we have provided below some single-cell spatial analysis using Resolve Molecular Cartography showing cell-type specific knockdown effects of CTNNB1 *in vivo* to describe tumor, hepatocyte, and non-parenchymal cell accumulation of the LNP. Demonstrated below are four β -catenin target genes, *Cyp2e1*, *Glul*, *Rgn*, and *Tbx3*, which are decreased in tumor cells, hepatocytes, and immune cells, with minimal impact to HSCs and ECs.

References

1. Maier MA, Jayaraman M, Matsuda S, et al. Biodegradable lipids enabling rapidly eliminated lipid nanoparticles for systemic delivery of RNAi therapeutics. *Mol Ther*. 2013;21(8):1570-1578.

Point 3. P9 lines199-201. What is the rationale for the duration between the injection of HDTV_i and the start of LNP treatment?

Response: We thank the reviewer for their comment. We have developed the 3 β -catenin-driven models (Tao et al. 2016. *Hepatology*, Tao et al. 2021. *Hepatology*; Lehrich et al. 2024. *JHep Reports*, refs 1-3 below) and extensively characterized these models and determined the timepoints where there is evidence of both microscopic and gross macroscopic tumors. The timepoints chosen were 3-weeks, 5-weeks, or 3-weeks for microscopic tumor formation in β -catenin-Met, β -catenin-Nrf2, β -catenin-Nrf2-Met models, respectively, and 6-weeks or 8-weeks for gross macroscopic tumor formation in β -catenin-Met and β -catenin-Nrf2 models, respectively. These stages represent what we defined as early versus advanced-stage disease treatment windows for the 3 β -catenin-driven models.

The Nrf2-Met model is a slower growing disease, as we had previously reported (ref 3), thus starting treatment at 8-weeks post-injection was still a microscopic disease treatment window. We had treated the mice to known morbidity timepoint in the different models.

We have provided these sentences to justify the use of these time points more clearly in the results section of the manuscript.

References

1. Tao J, Xu E, Zhao Y, et al. Modeling a human hepatocellular carcinoma subset in mice through coexpression of met and point-mutant beta-catenin. *Hepatology*. 2016;64(5):1587-1605.
2. Tao J, Krutsenko Y, Moghe A, et al. Nuclear factor erythroid 2-related factor 2 and beta-Catenin Coactivation in Hepatocellular Cancer: Biological and Therapeutic Implications. *Hepatology*. 2021;74(2):741-759.
3. Lehrich BM, Tao J, Liu S, et al. Development of Mutated β -catenin Gene Signature to identify CTNNB1 mutations from whole and spatial transcriptomic data in patients with HCC. *JHEP Reports*. 2024:101186.

Point 4. P8 lines175-176. Were there any adverse effects of LNP-CTNNB1 outside the liver, such as in the lungs? The toxicity data is also limited, and may be a concern.

Response: We thank the reviewer for their comment. In regards to the related safety profile, we have already described serum liver function tests/biochemistries (i.e., ALT/AST/ALKP) at the 3mg/kg dosage in the manuscript (Fig.S1j). We have also described that at the 3mg/kg dosage, we noted mortality in one of four mice tested. This is consistent with previous reports in rodent models tested with LNP from Alnylam [ref 1]. We have also performed bulk RNAseq on livers at the 3mg/kg dosage after 4 treatments demonstrating no immune alterations with LNP treatment (current Fig. S1k-l). In regard to the 1mg/kg dosage, which was used for therapeutic efficacy studies, we have additionally included the following statement in the main text of the manuscript:

“At the 1mg/kg LNP dosage, there was no morbidity or mortality observed in these mice. Additionally, there were no gross phenotypic changes to other organs, including the lungs, spleen, intestine, and heart. H&E of the spleens across a wide dose range did not demonstrate any microscopic findings (Fig. S2f). Moreover, we did not observe any

neurological, gastrointestinal, genitourinary, cardiovascular, or respiratory deficits and/or distress following the once weekly treatments over 6 weeks. These phenotypic observations have also been previously described across a broad dose range with no adverse clinical signs or toxicologically significant alterations to body weight or liver function tests [ref 1].”

Overall, Alnylam has performed studies and published on LNP formulations determining no off-target toxicity. Similar platform is used by Alnylam for FDA approved Onpattro, a formulation to decrease TTR in liver to decrease amyloid deposits. It should also be noted that a clinical trial has already been initiated using LNP-siCTNNB1 and in patient recruitment phase (NCT06600321).

References

1. Maier MA, Jayaraman M, Matsuda S, et al. Biodegradable lipids enabling rapidly eliminated lipid nanoparticles for systemic delivery of RNAi therapeutics. *Mol Ther.* 2013;21(8):1570-1578.

Point 5. P14. Is the cluster mapping in LNP-CTNNB1 treated tumors consistent with P11 lines 243-246?

Response: We thank the reviewer for their comment. Yes, the cluster mapping and expression of Glul is consistent. We performed cluster annotation of the single-cell RNAseq and single-cell spatial transcriptomic datasets using known marker genes based on the integrated datasets of the LNP-CTRL and LNP-CTNNB1 treated samples. Thus, cluster identities are shared and preserved between both treatment conditions, and cluster proportions provide insights into relative abundance in each condition. As can be noted in the violin plot below from the single-cell spatial transcriptomic dataset, Glul is retained in Zone 3: CV CTNNB1 WT (GS+), which is consistent with the IHC (P11 243-246): “GS protein expression visualized via IHC was decreased within tumor nodules, but retained in hepatocytes around central veins, at this 3-day timepoint”. We have included this data in the supplement of the new version of the manuscript.

Point 6. The paper asserts that their approach overcomes ICI resistance. The claim lacks direct evidence and should be revised to focus on enhancing ICI response rather than overcoming resistance.

Response: We thank the reviewer for their comment. We have toned down the statements throughout the manuscript where we originally asserted ICI resistance and synergy and have modified the manuscript to reflect our data that support that ICI enhances/augments LNP-CTNNB1 response. In fact, these observations are the basis of the clinical trial (NCT06600321).

Point 7. The variability in late-stage HCC responses suggests an unclear heterogeneity in the models or experimental conditions. This requires discussion or additional analyses.

Response: We thank the reviewer for their comment. We have performed significant additional analysis on the 10X Visium data for the late-stage HCC treatment responses. In addition to further characterization of the β -catenin-Met model, we have also performed 10X Visium analysis on the β -catenin-Nrf2 model to validate and support any claim we have made from the one model. Based on the new analyses, heterogeneity in response is two-fold: 1) NR phenotype may be driven in part by insufficient dosing, and 2) driven in part by biological differences where there is expansion of clones that are more predominant in the secondary driver (e.g., Met or Nrf2 clonal expansion following β -catenin suppression in some select mice; i.e., escape).

To summarize the findings with relevant data below from the 10X Visium platforms from both models.

1) We observed heterogeneous responses following LNP-CTNNB1 monotherapy treatment in advanced-stage disease. 10X Visium allowed us to explore spatially the transcriptome between the control, NR, and R tumors in both models. Performing unbiased clustering revealed many different cell populations (17 in β -catenin-Met and 13 in β -catenin-Nrf2) and proportions of these in the models and across NR and R (Fig6g-h, i-j).

2) In R animals in both models, we observed a decrease in expression of genes involved in cell cycle S and G2M phases, with relative proportional increase of cells in G1 phase, illustrating lack of cell proliferation with response to LNP-CTNNB1 and role of β -catenin in regulating cell cycle (β -catenin-Met on Left and β -catenin-Nrf2 on right) (FigS23a-g, S24a-g).

3) We then investigated expression of our previously reported mutated- β -catenin gene signature (MBGS) (Lehrich et al. *JHep Reports*, 2024), which includes expression of 13 genes downstream of oncogenic Wnt/ β -catenin signaling in HCC which we have shown to correlate with CTNNB1 mutational status, ICI responses, and spatially mapped Wnt/ β -catenin activation in 10X Visium spatial transcriptomics (ref 1). We first saw a decrease in MBGS expression from control to NR to R. Additionally, MBGS mapped spatially showed persistence of MBGS tumors in NR despite suppression of β -catenin with LNP-CTNNB1. This suggested to us that we observed heterogeneity in response driven in part by insufficient dosing in terms of quantity or frequency (FigS25a-f).

4) We also saw increased expression of MBGS in specific tumor clusters. We then wanted to investigate tumor clusters that were low in MBGS expression but expanded/persisted in the NR animals (cluster 3 in β -catenin-Met and cluster 11 in β -catenin-Nrf2 models). Interestingly, we observed that cluster 3 in β -catenin-Met model was upregulating pathways involved in PI3K-Akt signaling (downstream of Met activation) and pathways involved in actin cytoskeleton (downstream of Met activation). Additionally, cluster 11 in β -catenin-Nrf2 model was upregulating pathways involved in glutathione metabolism, reactive oxygen species, and antioxidant stress (which are downstream of NFE2L2 activation). This provided evidence to us that biological differences where there is expansion of clones that are more predominant from the secondary driver (e.g., Met or Nrf2 clonal expansion) which were driving the NR phenotype (FigS29a-b, S30a-b). These findings were validated via immunohistochemistry (IHC) as well.

References

1. Lehrich BM, Tao J, Liu S, et al. Development of Mutated β -catenin Gene Signature to identify CTNNB1 mutations from whole and spatial transcriptomic data in patients with HCC. JHEP Reports. 2024:101186.

Point 8. It would be helpful to include more data on the durability of tumor responses and immune effects post-treatment cessation.

Response: We have already provided data regarding the durability of tumor responses/overall survival post treatment cessation. The data is in FigS5 demonstrating a durable tumor response of 9-12 weeks post treatment cessation from the indicated timepoints, depending on the mode indicated. We have also characterized that the tumors that re-appear at this timepoint are β -catenin driven given the simultaneous expression of GS and the secondary driver (NQO1 or V5-tag for MET plasmid). Thus, the TIME is likely to be reverted to a Wnt driven TIME. It is unclear the rationale to include more IHC on the immune microenvironment post-treatment cessation. We had not presented any immune-related data at this point in the manuscript where this data is described, nor is this one of the main findings of the manuscript.

Point 9. The clinical trial data is extremely valuable and interesting. However, as presented, the B-cell data do not really connect well with the preclinical data.

Response: As described above, we observed an overall increase in quantity of GZMB+ lymphoid aggregates (LA) in LNP-CTNNB1 + anti-PD1 compared to LNP-CTNNB1 + IgG treated mice. GZMB+ T cells are enriched in tertiary lymphoid structures (TLS)/LA. TLS/LA are composed predominantly of T and B cells, along with macrophages, dendritic cells, and fibroblasts. From our 10X Visium analysis, we observed increases in both T and B cells in the same tumor clusters, which was confirmed spatially in situ and with IHC for T and B cell markers. We have de-emphasized the role of solely B cells in the manuscript, and rather focus on the re-engaged adaptive immunity due to LNP-CTNNB1. In the revised manuscript, we have utilized instead a combined T and B cell gene signature, which we call a "LA-like" gene signature, and correlated this with TLS/LA status and CTNNB1 mutation. We found high concordance between LA-like signature and TLS/LA presence (Fig8d-e) and clinical responses (Fig.8f). Finally, we found an inverse correlation between CTNNB1 mutational status and LA-like gene signature expression (Fig.8g). Overall, this highlighted to us two points: 1) HCC patients with CTNNB1 mutations have low presence of TLS/LA (based on low expression of LA-like signature, which correlates with TLS/LA presence), and 2) suboptimal response and PFS/OS in CTNNB1-mutated patients can be explained in part through low expression of genes important for TLS/LA formation and presence. Much work has determined that presence of TLS/LA is important for anti-tumor immunity, but it is poorly understood mechanisms that do not elicit TLS/LA formation. We show here that β -catenin mutation in HCC is one such mechanism, along with having prognostic importance in HCC. Overall, we provided this data to suggest that upon LNP-CTNNB1 + anti-PD1 treatment, we are inducing TLS/LA presence, which is lacking in CTNNB1 mutated patients, and this may be a good indicator for assessing response to ICI combinations in the future.

Minor:

Point 10. Line 132: “at 20nm concentration” should be corrected to “at 20nM concentration.”

Response: We thank the reviewer for their comment. We have changed this in the revised manuscript. “72-hour treatment with LNP-CTNNB1 at 20nM concentration led to a notable decrease in both the number and size of the organoid compared to treatment with a LNP-CTRL (Figure 1a-b).”

Point 11. P14 line 310. Figure 3c, not S3c.

Response: We thank the reviewer for their comment. We have changed this in the revised manuscript. “...were the least proliferative with proportionally fewer cells in G2M phase of the cell cycle (Figure 3c).”

Point 12. P17 lines 386-388. Presenting the expression of IFN γ or IFN γ receptors at the protein level in T cells or macrophages would provide stronger evidence.

Response: We thank the reviewer for their comment. We do not have IFN γ or IFN γ receptor expression at protein level in T cells or macrophages from the CITE-seq panel. We also do not have additional LNP reagent to re-perform this experiment due to clinical trial testing of this agent. We have toned down as best we can the conclusions that can be derived from such bioinformatic analyses from RNA expression without the protein validation.

Point 13. P17 lines 393-394. Did the authors evaluate the increase of M1-like macrophages in tumors in b-M mice treated with IFN γ ?

Response: We thank the reviewer for their comment. We did not perform single-cell RNAseq or FACS for M1/M2 markers in β -catenin-Met mice treated with IFN γ . The purpose of this experiment was to validate that increased IFN γ , which was observed following LNP-CTNNB1 treatment in the single-cell RNAseq data, would lead to tumor responses. We observed a noticeable and statistically significant decrease in tumor burden following IFN γ treatment in β -catenin-Met mice. We have provided IHC for S100A8/9, a M1 macrophage marker, and toned down the conclusion that IFN γ is solely driving the repolarization from M2 to M1, since we have not performed a full characterization panel.

Point 14. P19 lines 437-439. Was there any particular lymphocyte distribution apparent in the tumors (center or edge)?

Response: We thank the reviewer for their comment. We noted the majority of lymphocytes appeared at the periphery/edge of the tumor at the timepoint analyzed. We are well aware that immune infiltration is a dynamic process and lymphocytes may have engaged with in the tumor core at an earlier timepoint than which we performed the IHC analysis. Thus, we do not want to make any definitive conclusions on the localization without a proper time course study and inducible delivery of the TF (i.e., IRF2/POU2F1), which is current ongoing work, and outside the current scope of the study.

Reviewer #2 (Remarks to the Author): with expertise in HCC, cancer immunology

CTNNB1 mutation and pathway activation is one of the major reasons for the development/progression and immunotherapy resistance of HCC. In this manuscript, the authors demonstrated the potent efficacy and potential mechanistic interpretation of a novel siRNA encapsulated in lipid nanoparticle targeting CTNNB1 (LNP-CTNNB1) in multiple preclinical HCC models. Based on single-cell and spatial transcriptomics data, they demonstrated that reprogramming of immune regulatory transcription factors IRF2 and POU2F1 in CTNNB1-mutated tumors, which is associated with type I/II interferon signaling, and alterations in both innate and adaptive immune responses, may be responsible for the therapeutic efficacy. The study is interesting; however, more data and explanation are needed to support their conclusion.

Response: We thank the reviewer for their comment. The reviewer well summarized the main points that we “demonstrated the potent efficacy” and “mechanistic interpretation of a novel siRNA encapsulated in lipid nanoparticle targeting CTNNB1 (LNP-CTNNB1) in multiple preclinical HCC models”. We also utilized “single-cell and spatial transcriptomic data” to investigate mechanism. We have outlined a plan below to address the reviewer comments.

Major comments:

Point 1. Figure 1: What adverse effects of LNP-CTNNB1 have been tested? The overall and organ weights measurement as well as the histology by H&E staining after LNP-CTNNB1 treatment to evaluate the damage of different organs should be performed.

Response: We thank the reviewer for their comment. In regard to the related safety profile, we have already described serum liver function tests/biochemistries (i.e., ALT/AST/ALKP) at the 3mg/kg dosage in the manuscript (Fig.S1j). We have also described that at the 3mg/kg dosage, we noted mortality in one of four mice tested. This is consistent with previous reports in rodent models tested with LNP from Alnylam [ref 1]. We have also performed bulk RNAseq on livers at the 3mg/kg dosage after 4 treatments demonstrating no immune alterations with LNP treatment (Fig. S1k-l). Regarding the 1mg/kg dosage, which was used for therapeutic efficacy studies, we have additionally included the following statement in the main text of the manuscript:

“At the 1mg/kg LNP dosage, there was no morbidity or mortality observed in these mice. Additionally, there were no gross phenotypic changes to other organs, including the lungs, spleen, intestine, and heart. H&E of the spleens across a wide dose range did not demonstrate any microscopic findings (Fig. S2f). Moreover, we did not observe any neurological, gastrointestinal, genitourinary, cardiovascular, or respiratory deficits and/or distress following the once weekly treatments over 6 weeks. These phenotypic observations have also been previously described across a broad dose range with no adverse clinical signs or toxicologically significant alterations to body weight or liver function tests [ref 1].”

Overall, Alnylam has performed studies and published on LNP formulations determining no off-target toxicity. Similar platform is used by Alnylam for FDA approved Onpattro, a formulation to decrease TTR in liver to decrease amyloid deposits. It should also be noted

that a clinical trial has already been initiated using LNP-siCTNNB1 and in patient recruitment phase (NCT06600321).

References

1. Maier MA, Jayaraman M, Matsuda S, et al. Biodegradable lipids enabling rapidly eliminated lipid nanoparticles for systemic delivery of RNAi therapeutics. *Mol Ther.* 2013;21(8):1570-1578.

Point 2. Did LNP-CTNNB1 affect survival of the mice in the CTNNB1-mutated model?

Response: We assessed both the immediate effect of treatment intervention (Fig.1) and survival/durability of response (Fig.S5) in both the β -catenin-Met and β -catenin-Nrf2 models. As described in Fig.S5, this data demonstrated a durable tumor response of 9-12 weeks, depending on the model compared to untreated controls following treatment cessation. Thus, LNP-CTNNB1 improved survival of CTNNB1 mutated HCCs notably.

Point 3. Figure 4: The authors suggested that ‘increased type I/II interferons released from the immune compartment (likely from T cells and macrophages) following LNP-CTNNB1 treatment are engaging with macrophages in the TIME milieu, and in part contributing towards polarizing them towards a pro-inflammatory anti-tumor phenotype’. While they provided evidence by IFN γ treatment (Fig 4h-j), they need to show that the IFN γ level was indeed altered in the LNP-CTNNB1 compared to LNP-CTRL treated tissues by ELISA. Moreover, after IFN γ treatment, was there change in M1/M2 macrophage proportion as anticipated?

Response: Through single-cell RNA-seq, we observed increased IFN γ expression in T cells, which was shown to communicate with macrophages via CellChat analysis. As stated by the reviewer, we then validated that IFN γ induced tumor response. We have now also shown that IFN γ increased M1 macrophages in TIME via S100a8/9 IHC. We did also perform an ELISA for IFN γ from whole liver lysate with LNP treatment, however, did not observe a difference at the 3d timepoint. We attribute this to single-cell analyses being able to detect more granular findings, given the cell-type specificity of our findings, and the early timepoint we queried. We hypothesize the latter due to us being able to detect increased IFN γ in the advanced-stage disease setting which after 6 cycles of LNP.

Point 4. Figure 4: The TME characterization by single-cell transcriptomic analysis, especially the changes in M1/M2-like macrophages in the LNP-CTNNB1 treated tumors should be validated by flow cytometry. Would modulation of these macrophages reverse the antitumor effects of LNP-CTNNB1 treatment?

Response: We have utilized advanced single-cell transcriptomics to profile the transcriptional landscape of various macrophage populations which shifted following LNP treatment. However, we do not have additional LNP reagent to perform this experiment due to PhI/II clinical trial that is ongoing with this agent. It is standard practice to halt preclinical studies once clinical studies have started. We have shown functional consequence of increased M1 polarization through IFN γ studies in β -catenin-mutated HCC model illustrating effects on anti-tumor immunity.

Point 5. Figure 5: What is the molecular mechanism by which mutated- β -catenin suppresses the expressions of IRF2 and POU2F1? This key link is missing in this study.

Response: We have provided extensive evidence with validation of two transcription factors (IRF2/POU2F1) that are repressed when β -catenin is active. We have demonstrated this suppression for each of these in both murine and human models, their clinical relevance to CTNNB1 mutations (TCGA), their *in vivo* functional validation and modulation, and consequence on tumor immune microenvironment with both RNA-seq and IHC. While we understand it would be important to query the mechanism underlying suppression, this work is ongoing and requires additional time and resources. This additional data would be suitable for a follow-up manuscript.

We posit that β -catenin is directly influencing expression of essential cytokines/chemokines necessary for immune recruitment and function through modulation of an immune-regulatory module of key TFs which are normally active when β -catenin is absent from the nucleus. Previous work has described sequestration and inhibition of p65 (a subunit of NF- κ B) by β -catenin directly leading to reduced binding, target gene expression, and inflammation [refs 1-3]. Thus, here we have described similar findings, yet with 2 additional TFs, IRF2 and POU2F1. Further ongoing work in the lab is aimed to determine how β -catenin interacts with both, which cytokines/chemokines downstream of IRF2 and POU2F1 are essential for immune response in HCC, and which immune subsets are most affected by re-engagement of these TFs. Here, we describe that efficacy of LNP-CTNNB1 is mediated in part by promoting re-engagement of these TFs and target gene expression.

References

1. Du Q, Zhang X, Cardinal J, et al. Wnt/beta-catenin signaling regulates cytokine-induced human inducible nitric oxide synthase expression by inhibiting nuclear factor-kappaB activation in cancer cells. *Cancer Res.* 2009;69(9):3764-3771.
2. Deng J, Miller SA, Wang HY, et al. beta-catenin interacts with and inhibits NF-kappa B in human colon and breast cancer. *Cancer Cell.* 2002;2(4):323-334.
3. Nejak-Bowen K, Kikuchi A, Monga SP. Beta-catenin-NF-kappaB interactions in murine hepatocytes: a complex to die for. *Hepatology.* 2013;57(2):763-774.

Point 6. Is there a correlation between CTNNB1 mutation/expression and IRF2/POU2F1 expression in HCC patients?

Response: We have investigated this and observed that CTNNB1 mutated and wild-type patients do not have significantly different expression of IRF2 or POU2F1, rather it is the target genes of these transcription factors which are differentially expressed (Fig.5d). We assert that β -catenin activation is repressing a module of transcription factors, through yet an undetermined mechanism, leading to inert immune responses in β -catenin-mutated HCC. This is similar to what we have shown with β -catenin regulation of p65 subunit of NFkB (PMID: 28474571; 34609282). We have provided data to also show high expression of IRF2/POU2F1 target genes was associated with improved disease-free survival in those with Wnt/ β -catenin activating mutations ($p=0.065$) (Fig.S16e-f).

Point 7. Figure 7m: The cell cycle changes in cancer cells shown in the schematic diagram after LNP-CTNNB1 treatment should be validated by flow cytometry.

Response: We have provided robust analyses of cell cycle changes via single-cell RNAseq cell cycle regression analysis. We have also validated the single-cell RNAseq with two 10X Visium datasets in the late-stage using similar methodology to score cells in each phase of the cell cycle: G1, S, G2M. This is based on gene signatures for each of the phases of the cell cycle. This assigns cell cycle scores for each phase in each cell type. We believe flow cytometry would not be able to reach this sensitivity since collagenase perfusion to isolate cells, itself impacts cell cycle in hepatocytes because of loss of extracellular matrix. Additionally, the granularity that can be gained here but cell type specification would not be achievable with flow cytometry. Again, we do not have additional LNP reagent to perform this experiment due to clinical testing of this agent, but have sufficiently shown this result already across multiple transcriptomic assays.

Point 8. Figure 8f: “Increased Bsig expression was also observed in atezolizumab plus bevacizumab arm in patients with CR/PR and SD, while decreased Bsig expression was observed in those with PD (Figure 8f).” However, there is no obvious change in B cells upon LNP-CTNNB1 treatment as shown in Figure 4b. Thus the mouse study and human data is not concordant. No difference in TLS/LA has been shown between MUT and WT patients. The importance of B cells and TLS formation in CTNNB1 mutation is not adequately supported. Can the authors provide more direct evidence demonstrating that “formation for TLS/LA may be restricted by mutated β -catenin” (line 555-556)?

Response: We have taken emphasis off of B cells in the manuscript, but rather focused on the re-engaged adaptive immune pool, both T and B cells. This is reflected in the new analysis in Fig.8, which uses a T and B cell gene signature, which we have called “LA-like” gene signature, to reflect the two major cell populations composing TLS/LA. The reviewer points out that B cells were not changed upon LNP treatment in Fig.4b due to the early timepoint (3d post LNP treatment), but rather alterations in B cells were observed in advanced-stage disease setting, which is normal for adaptive immunity to follow innate immune responses.

Point 9. Summary: “Lastly, CTNNB1-mutated patients treated with atezolizumab plus bevacizumab combination had decreased presence of lymphoid aggregates, which were prognostic for response and survival.” No data is found to support this statement in this manuscript.

Response: We have rephrased this sentence to fit the data in the manuscript. We have changed the sentence in the abstract to the following: “Lastly, a “LA-like” gene signature expression was inversely correlated with CTNNB1-mutational status, which was associated with response and survival to atezolizumab plus bevacizumab in IMbrave150 trial.”

Minor comments:

Point 10. Figure 1b: Quantification should be performed to indicate LNP-CTNNB1 efficacy in mutant-CTNNB1 human HCC organoid cultures.

Response: We thank the reviewer for their comment. We have now performed additional experiments and provided quantification of efficacy of LNP-CTNNB1 in CTNNB1-mutated organoid cultures via quantity and size of the organoids.

Point 11. Tumor burden should be evaluated by the number and diameter of tumor nodules in H&E staining in addition to LW/BW.

Response: Our group has pioneered use of SB-HDTV_i model for modeling CTNNB1-mutated liver tumors (over 30 publications as senior corresponding or contributing author including in *Gastroenterology*, *Hepatology*, *Cell Metabolism*, *Cancer Discovery*, *Nature Communications*, *Nature*, *Nature Biotechnology* and many others). Compared to other co-mutations with HDTV_i model, the β -catenin-mutated HCC models are well differentiated HCC where the entire liver becomes tumor tissue by advanced disease. Surface nodules to assess tumor nodule number or size are not visible, as is seen in more undifferentiated tumors (e.g. Myc models or β -catenin/Yap hepatoblastoma models). We firmly believe that LW/BW is the most relevant surrogate for tumor burden in these models. We have provided scanned tiled images of large liver sections as evidence of response. We have also included MRI in a select few cases as another indicator of tumor burden and improvement. We have, however, now quantified the area of tumor by GS staining as well to report in Fig1.

Point 12. Figure 7i: What is the mRNA level of CTNNB1 after single anti-PD-1 treatment?

Response: We don't have any mouse tissue banked from a single anti-PD-1 treatment and unsure of the significance of this to our study. It is unclear how a single anti-PD-1 treatment would impact Ctnnb1 levels.

Reviewer #3 (Remarks to the Author): with expertise in nanoparticles, HCC therapy

The manuscript entitled “Precision targeting of β -catenin induces tumor reprogramming and immunity in hepatocellular cancers” reported that LNP delivering si CTNNB1 is effective in treating HCC with CTNNB1 mutations by affecting intrinsic signaling within tumor cells and remodeling tumor immune surveillance. β -Catenin, considered an undruggable target, is encoded by the CTNNB1 gene, which undergoes mutations across various cancers with mutation sites differing among patients. Using a single siRNA to target multiple mutation sites is unlikely to be effective, potentially posing significant challenges to the clinical translation of this project. Although LNP delivering si CTNNB1 showed certain advantages, this work still has shortcomings in terms of novelty and solidity.

Response: We thank the reviewer for their comment. The reviewer well summarized the main points that we demonstrated “LNP delivering si CTNNB1 is effective in treating HCC with CTNNB1 mutations by affecting intrinsic signaling within tumor cells and remodeling tumor immune surveillance”. We have clearly demonstrated that the siRNA utilized suppresses CTNNB1 mRNA levels in multiple humanized mouse models with different point mutations in β -catenin (S45, T41, etc). Thus, this LNP-siRNA is clinically translatable, and a clinical trial has already been initiated and patients are being recruited (NCT06600321). We have outlined a plan below to address the reviewer comments.

Major Comments:

Point 1. Novelty and Conceptual Framework: The novelty and conceptual framework of the study are somewhat limited, particularly regarding material selection. The effects of CTNNB1 siRNA on hepatocellular carcinoma (HCC) have been explored in previous studies. Additionally, the lipid nanoparticle (LNP) composition and synthesis methods employed here are relatively conventional. This raises concerns regarding the originality of the approach and its potential for advancing therapeutic strategies.

Response: We would like to emphasize that the study is not focused on chemical formulation of the LNP or siRNA. These agents were provided by Alnylam Pharmaceuticals based on their robust platform and existing pipeline of drugs using similar formulations that are routinely given to patients such as for porphyria and amyloidosis. The current LNP is similar to what is used in their FDA approved agent Onpattro. Our goal here is to characterize CTNNB1 mutated HCCs, their unique biology, and how β -catenin suppression in these tumors impacts tumor biology for therapeutic exploitation. This agent that we have used in our current study has successfully received IND designation by FDA and a clinical trial has already been initiated and patients are being recruited (NCT06600321). The reviewer mentions previous work describing LNPs formulated with siRNA to CTNNB1, in which we have also reported on. There are several distinctions from our prior work compared to the current manuscript.

1. We previously therapeutically tested a LNP-siRNA (from Dicerna Pharmaceuticals) in mutated β -catenin-Kras (β -K model) and initiated treatment very early in tumorigenesis. The models used in the current study β -catenin-Nrf2, β -catenin-Met, and β -catenin-Nrf2-Met are more clinically relevant models representing ~25-30% of all HCC. Co-alterations of β -catenin and Kras are

infrequently seen in HCC. Thus, for advancing clinical utility of novel agents, the models utilized in this study represents a significant advance from the prior work to demonstrate efficacy of β -catenin targeting in clinically relevant CTNNB1-mutated HCC models. Additionally, in this study we not only assess response to LNP-CTNNB1 in early-stage disease, but also advanced-stage disease in combination with ICI, which hasn't been demonstrated previously.

2. The current study provides an in-depth novel investigation into the tumor biology following β -catenin knockdown using sophisticated single-cell and spatial approaches at various timepoints and in different models to assess response. To our knowledge, there hasn't been as thorough of a characterization of these β -catenin mutated models, which these new omics resources could be used by others for subsequent investigations into tumor biology of CTNNB1-mutated HCC.
3. Through these single-cell and spatial approaches, we systematically investigated changes in tumor cell-intrinsic biology and effects on TIME, which was not described in our previous work in 2017. Specifically, the novelty is based on impact on zonation and tumor cell-intrinsic impact on macrophages and lymphocytes, which provided opportunity to synergize with immune checkpoint inhibitors, which are the standard of care for HCC treatment now and were not FDA approved for use in 2017, when our previous work was published.

Overall, our current manuscript is a significant advance over the previous work and describes multiple roles of β -catenin in HCC biology that are both tumor cell-intrinsic (cell proliferation, cell survival, zonation) and TIME specific, especially regarding macrophage and lymphocyte recruitment and function. All these studies have led to initiation of precision-medicine based phase I/II clinical trial (NCT06600321) led by Alnylam Pharmaceuticals for CTNNB1-mutated and β -catenin active HCC, which is a demonstrable advance in the field where personalized therapies are absent. This is the first precision-medicine trial of its kind in HCC space.

Point 2. Evaluation of LNP Properties: The manuscript provides insufficient characterization of the LNP system employed. Detailed information on the composition and physicochemical properties of the LNPs, such as particle size, charge, and stability, is lacking. For instance, the particle size of LNP-siRNA is reported as 61.3 nm, but it is unclear whether this measurement is based on number-based or intensity-based distribution. Furthermore, the stability of the LNP system under storage conditions should be evaluated to ensure its viability for clinical translation.

Response: The focus of our work is not characterization of the LNP-siRNA which is the expertise of Alnylam, but impact of β -catenin inhibition on tumor biology. Several novel aspects of impact of β -catenin inhibition have been presented and are the foundation of clinical studies which is using the same agent. In regard to the LNP characterization, in the Methods section under 'LNP-siRNA synthesis' we have currently referenced manuscripts which have described the synthesis and formulation of the LNP. We have also included the siRNA sequences for the siControl and siCTNNB1. The LNP formulation we are using is biosimilar to what is currently being used for treating patients with hereditary amyloidosis

(Onpattro/patisiran) and is currently manufactured by Alnylam Pharmaceuticals. The LNP used in this study is in Ph1 clinical trial (NCI-2024-08329/NCT06600321) currently and we have focused this manuscript mostly on therapeutic efficacy of this agent which supports the ongoing effort to target β -catenin-mutated HCC. We have however included the following statement in the Methods section to provide additional information:

“The LNPs used in this study size range from 60-80nm with siRNA encapsulation efficiency of ~95% (PDI = 0.045) with characterization and quality control performed as previously described [ref 1]. LNP formulations from Alnylam have demonstrated a rapid accumulation of the parent lipids in the liver and plasma with elimination from plasma by 8-hours, liver by 48-hours, and spleen by 96-hours post systemic administration via tail-vein injection [ref 1]. Biodistribution of the LNPs have also been described to accumulate primarily in the liver and spleen [ref 1].”

References

1. Maier MA, Jayaraman M, Matsuda S, et al. Biodegradable lipids enabling rapidly eliminated lipid nanoparticles for systemic delivery of RNAi therapeutics. *Mol Ther*. 2013;21(8):1570-1578.

Point 3. Statistical Rigor: Several experiments, including those depicted in Figure 3a, suffer from inadequate sample sizes. Specifically, the LNP-CTRL group (n=2) and LNP-CTNNB1 group (n=3) are insufficient to yield reliable conclusions, particularly given that the observed cell volume differences between the two groups are substantial (up to 2.5-fold). A more robust statistical design, with increased sample sizes and uniform group numbers, is recommended to enhance the reliability of the findings.

Response: We would like to clarify the sample sizes used in our study and the robustness of our models. First, our β -catenin-mutated HCC models are highly reproducible, have high penetrance, and have been validated by others as well since our original publications of these models: β -catenin-Nrf2 model (2021, Tao et al. *Hepatology*), β -catenin-hMet (2016, Tao et al. *Hepatology*), β -catenin-Nrf2-hMet & Nrf2-hMet (2024, Lehrich et al. *JHep Reports*). In fact, our group has published over 30 studies using these models in journals like *Cell metabolism*, *Gastroenterology*, *Hepatology*, *Nature*, and others, and the studies are highly consistent and reproducible because of optimization. In our early-stage disease model testing of LNP, treatment groups with LNP-CTNNB1 were n=7/group and in late-stage disease model testing of LNP, treatment groups with LNP-CTNNB1 were n=8/group. Control groups of LNP-CTRL in some cases were n=3-5/group; however, the high reproducibility and penetrance of these models in our hands demonstrates the adequacy of the N used for each group in each treatment scenario. Additionally, the fact that we have observed consistent results across multiple β -catenin-mutated HCC models provides rationale as to the validity of our results and generalizability. In terms of the single-cell analyses discussed by the reviewer, performing single-cell RNA-seq on more samples would be extremely costly. Additionally, we have consulted bioinformaticians regarding the cell volume differences between the conditions and it is standard practice to normalize the cell proportions to total number of cells in the sample, which we have done, in instances of this. This difference in cell volume is likely due to technical perfusion differences in tumor models.

Point 4. Quantification of Delivery Efficiency and Specificity: While the study demonstrates the anti-tumor effects of LNP-CTNNB1 *in vivo*, the delivery efficiency and cell specificity of the LNP system have not been sufficiently quantified. The manuscript would benefit from a more thorough analysis of the distribution of LNP-CTNNB1 particles in various tissues and tumors post-administration. This is essential for evaluating the therapeutic potential and specificity of the delivery system.

Response: We thank the reviewer for their comment. We have provided below some single-cell spatial analysis using Resolve Molecular Cartography showing cell-type specific knockdown effects of CTNNB1 *in vivo* to describe tumor, hepatocyte, and non-parenchymal cell accumulation of the LNP. Demonstrated below are four β -catenin target genes, *Cyp2e1*, *Glul*, *Rgn*, and *Tbx3*, which are decreased in tumor cells, hepatocytes, and immune cells, with minimal impact to HSCs and ECs.

In regards to the 1mg/kg dosage, which was used for therapeutic efficacy studies, we have additionally included the following statement in the main text of the manuscript:

“At the 1mg/kg LNP dosage, there was no morbidity or mortality observed in these mice. Additionally, there were no gross phenotypic changes to other organs, including the lungs, spleen, intestine, and heart. H&E of the spleens across a wide dose range did not demonstrate any microscopic findings (Fig. S2f). Moreover, we did not observe any neurological, gastrointestinal, genitourinary, cardiovascular, or respiratory deficits and/or

distress following the once weekly treatments over 6 weeks. These phenotypic observations have also been previously described in rodents across a broad dose range with no adverse clinical signs or toxicologically significant alterations to body weight or liver function tests [ref 1].”

References

1. Maier MA, Jayaraman M, Matsuda S, et al. Biodegradable lipids enabling rapidly eliminated lipid nanoparticles for systemic delivery of RNAi therapeutics. *Mol Ther.* 2013;21(8):1570-1578.

Point 5. Experimental Rationale in Figure 1b: The rationale for selecting a 20 nM concentration of LNP-CTNNB1 and a 72-hour incubation period in Figure 1b remains unclear. The authors should provide a more detailed justification for these experimental conditions, citing relevant studies or experimental data to support the chosen dose and time points.

Response: We selected the 20 nM concentration and 72-hour time point based on data provided by Alnylam testing their LNP *in vitro* in HepG2 spheroids (unpublished and not presented in this manuscript). **See confidential data below to be redacted.**

[figure redacted]

Point 6. Tumor Resistance Mechanism: In Figure 6, the manuscript mentions a tumor model that did not respond to treatment, but the underlying mechanism of resistance remains unexplored. To gain a deeper understanding of this phenomenon, it is recommended that single-cell RNA sequencing and immunohistochemical analysis be conducted on the resistant models. These approaches may provide insights into the molecular mechanisms responsible for treatment failure.

Response: We thank the reviewer for their comment. We have performed significant additional analysis on the 10X Visium data for the late-stage HCC treatment responses. In addition to further characterization of the β -catenin-Met model, we have also performed 10X Visium analysis on the β -catenin-Nrf2 model to validate and support any claim we have made from the one model. Based on the new analyses, heterogeneity in response is two-

fold: 1) NR phenotype may be driven in part by insufficient dosing, and 2) driven in part by biological differences where there is expansion of clones that are more predominant in the secondary driver (e.g., Met or Nrf2 clonal expansion following β -catenin suppression in some select mice; i.e., escape).

To summarize the findings with relevant data below from the 10X Visium platforms from both models.

1) We observed heterogeneous responses following LNP-CTNNB1 monotherapy treatment in advanced-stage disease. 10X Visium allowed us to explore spatially the transcriptome between the control, NR, and R tumors in both models. Performing unbiased clustering revealed many different cell populations (17 in β -catenin-Met and 13 in β -catenin-Nrf2) and proportions of these in the models and across NR and R (Fig6g-h, i-j).

2) In R animals in both models, we observed a decrease in expression of genes involved in cell cycle S and G2M phases, with relative proportional increase of cells in G1 phase, illustrating lack of cell proliferation with response to LNP-CTNNB1 and role of β -catenin in regulating cell cycle (β -catenin-Met on Left and β -catenin-Nrf2 on right) (FigS23a-g, S24a-g).

3) We then investigated expression of our previously reported mutated- β -catenin gene signature (MBGS) (Lehrich et al. *JHep Reports*, 2024). which includes expression of 13 genes downstream of oncogenic Wnt/ β -catenin signaling in HCC which we have shown to correlate with CTNNB1 mutational status, ICI responses, and spatially mapped Wnt/ β -catenin activation in 10X Visium spatial transcriptomics (ref 1). We first saw a decrease in MBGS expression from control to NR to R. Additionally, MBGS mapped spatially showed persistence of MBGS tumors in NR despite suppression of β -catenin with LNP-CTNNB1. This suggested to us that we observed heterogeneity in response driven in part by insufficient dosing in terms of quantity or frequency (FigS25a-f).

4) We also saw increased expression of MBGS in specific tumor clusters. We then wanted to investigate tumor clusters that were low in MBGS expression but expanded/persisted in the NR animals (cluster 3 in β -catenin-Met and cluster 11 in β -catenin-Nrf2 models). Interestingly, we observed that cluster 3 in β -catenin-Met model was upregulating pathways involved in PI3K-Akt signaling (downstream of Met activation) and pathways involved in actin cytoskeleton (downstream of Met activation). Additionally, cluster 11 in β -catenin-Nrf2 model was upregulating pathways involved in glutathione metabolism, reactive oxygen species, and antioxidant stress (which are downstream of NFE2L2 activation). This provided evidence to us that biological differences where there is expansion of clones that are more predominant from the secondary driver (e.g., Met or Nrf2 clonal expansion) which were driving the NR phenotype (FigS29a-b, S30a-b). These findings were validated via immunohistochemistry (IHC) as well.

5) Lastly upstream regulator analysis revealed MET was one of the top kinases predicted to regulate the DEGs in cluster 3 in β -catenin-hMet model and NFE2L2 was one of the top transcription factors predicted to regulate the DEGs in cluster 11 in β -catenin-Nrf2 model.

We added more discussion on the heterogeneity based on all this additional analysis of the spatial transcriptomic data. We have significantly edited the results to reflect these new novel findings. We have presented these findings in the supplement related to Fig6 and Fig7.

References

1. Lehrich BM, Tao J, Liu S, et al. Development of Mutated β -catenin Gene Signature to identify CTNNB1 mutations from whole and spatial transcriptomic data in patients with HCC. *JHEP Reports*. 2024;101186.

Point 7. Role of IRF2 and POU2F1: The finding that overexpression of IRF2 and POU2F1 in the β -catenin mutant model leads to a significant reduction in liver tumor growth suggests that these factors play critical roles in β -catenin signaling. However, the mechanisms by which IRF2 and POU2F1 influence β -catenin signaling in HCC require further investigation. More detailed studies are needed to elucidate how these factors modulate liver growth and tumorigenesis.

Response: We have provided extensive evidence in preclinical models and patients that IRF2/POU2F1, two transcription factors regulating immune cell responses when β -catenin is absent, are repressed when β -catenin is active. We have demonstrated for each of these in both murine and human models, their clinical relevance to CTNNB1 mutations, their *in vivo* functional validation and manipulation, and consequence on tumor immune microenvironment with both RNA-seq and IHC. We believe investigating the mechanism of how β -catenin represses these transcription factors would be suitable for a follow-up manuscript and is outside the scope of the current study.

Previous work by us and others has described sequestration and inhibition of p65 (a subunit of NF- κ B) by β -catenin directly leading to reduced binding, target gene expression, and inflammation [refs 1-3]. Thus, here we have described similar findings, yet with 2 additional TFs, IRF2 and POU2F1. Further ongoing work in the lab is aimed to determine how β -catenin interacts with both TFs, which cytokines/chemokines downstream of IRF2 and POU2F1 are essential for immune response in HCC, and which immune subsets are most affected by re-engagement of these TFs. Here, we describe that efficacy of LNP-CTNNB1 is mediated in part by promoting re-engagement of these TFs and target gene expression.

References

1. Du Q, Zhang X, Cardinal J, et al. Wnt/beta-catenin signaling regulates cytokine-induced human inducible nitric oxide synthase expression by inhibiting nuclear factor-kappaB activation in cancer cells. *Cancer Res*. 2009;69(9):3764-3771.

2. Deng J, Miller SA, Wang HY, et al. beta-catenin interacts with and inhibits NF-kappa B in human colon and breast cancer. *Cancer Cell*. 2002;2(4):323-334.

3. Nejak-Bowen K, Kikuchi A, Monga SP. Beta-catenin-NF-kappaB interactions in murine hepatocytes: a complex to die for. *Hepatology*. 2013;57(2):763-774.

Point 8. Sample Size in Omics Analysis: Although extensive omics analyses were employed to examine the effects of RNAi-mediated β -catenin inhibition on intrinsic signaling and immune surveillance in HCC, the number of animal samples analyzed (n=2,3,4) is relatively small. To enhance the statistical power and reliability of the results, a larger sample size should be used, and the study should consider additional validation at both the cellular and molecular levels.

Response: We respectfully disagree with the reviewer comments here. We have performed validation at the cellular and molecular levels for each of the major findings of the omics analyses despite limited numbers due to excessive costs of the technologies being employed in our study. To summarize, we have performed bulk RNA-sequencing on 4 models, 2 separate single-cell RNA-seq analyses on 1 model, single-cell spatial transcriptomics on 1 model, and 10X Visium spatial transcriptomics on 2 models. This is 9 omics analyses overall in this study, which is truly unprecedented to characterize these models deeply, and these resources can be used by others for future work.

We have validated all our major findings from the omics analyses as well. For example, qRT-PCR and IHC of zonation targets from the bulk RNAseq and single-cell spatial transcriptomics via Molecular Cartography has been performed and validated the findings; IFN γ treatment in β -catenin-Met model validated the single-cell profiling results in macrophages; and, IHC staining for immune cells validated the immune effects observed from the 10X Visium platform, along with others. Overall, increasing the N of the omics would not provide additional data that hasn't already been validated in multiple samples by traditional molecular biology techniques. It is pretty standard to use limited samples in single cell spatial transcriptomic studies as long as key results are validated, which is what we have included throughout our study.

Point 9. Clarity of Images: Several figures in the manuscript, such as Figure 1b, Figure 2e, and Figure 5m, lack sufficient resolution, which hampers the ability to discern the significant differences highlighted in the text. Retaking these images with higher clarity is recommended to ensure that the key findings are clearly visible and well-supported by the visual data.

Response: None of the other reviewers or Editor(s) had concerns about resolution. The resolution may be lost during download of the figures into PDF. High-resolution figures can be provided.

Point 10. Lack of Citation for Statement on RNAi: The manuscript states that “RNAi-mediated gene silencing has proven to be an excellent tool for targeting the traditionally ‘undruggable,’ especially in hepatic tissue,” but no citation is provided to support this assertion. It is crucial to include appropriate references to substantiate this claim and place the statement within the broader context of RNAi-based therapeutics.

Response: We have provided a citation here for this point. See ref below.

References

Setten RL, Rossi JJ, Han SP. The current state and future directions of RNAi-based therapeutics. *Nat Rev Drug Discov.* 2019;18(6):421-446.

Reviewer #4 (Remarks to the Author): with expertise in HCC, cancer immunology

The manuscript tries to address a relevant topic in hepatocellular carcinoma (HCC), focusing on the role of β -catenin mutations in immune exclusion and tumor progression. The authors investigate the effects of targeting β -catenin mutations with a siRNA encapsulated in lipid nanoparticles (LNP-CTNNB1) in HCC mouse models, with a particular emphasis on immune modulation and tumor response. The study aims to better understand the immune microenvironment in HCC and explore therapeutic strategies to overcome immune exclusion. However, several aspects of the study and its design raise concerns about the rationality and clinical applicability of the findings.

Response: We have outlined a plan below to address the reviewer comments. We would like to emphasize that the clinical applicability of the study is that our data in the current study was used for successful filing of IND status for this agent with the FDA. Also, this LNP-siRNA against β -catenin will be used for a clinical trial already initiated using this agent (**NCT06600321**; <https://clinicaltrials.gov/study/NCT06600321>). Our studies in the current manuscript were critical and formed the basis of these phase I/II trials, the first of its kind that will be precision-medicine based and for CTNNB1-mutated subclass of HCC.

Major Comments:

Point 1. The author used siRNA to target the mutant β -catenin in a humanized hydrodynamic HCC mouse model. This approach only inhibits tumor intrinsic β -catenin signal. However, the tumorigenic effects of β -catenin signaling in human cancers also involve immune suppression via exogenous factors like TGF- β . In the current mouse model, the partial humanization prevents the siRNA from directly impacting immune cells, which weakens the translation of the results to human conditions. Thus, this model does not fully reflect the complexity of the immune microenvironment in human HCC. This limited the design of the study.

Response: β -catenin mutations in hepatocytes are truncal events and one of the three major drivers of HCC. It is widely accepted that tumor cell-intrinsic mutations in hepatocytes are driving the disease. Having published >30 manuscripts using CTNNB1 mutation driven HCC models and demonstrating high transcriptomic expression overlap between our preclinical model and subsets of human HCCs with CTNNB1 mutations, we and others have validated the relevance of these mutations and models. The fact that tumors can be generated by expressing mutant- β -catenin (along with another relevant event that is also evident in patients like Met or Myc or Nrf2 activation) in hepatocytes is itself a proof that these mutations are sufficient to drive tumorigenesis. And that inhibition of β -catenin when tumors are established can have a profound impact through impact on tumor proliferation, survival, metabolism, angiogenesis, and tumor microenvironment. We also respectfully disagree with the reviewer's comments. The LNP-siRNA utilized targets the endogenous mouse and humanized forms of *Cttnb1*/CTNNB1 introduced. Thus, as presented, there is impact on immune cells as evident by single-cell data presented. Demonstrated below are four β -catenin target genes, *Cyp2e1*, *Glul*, *Rgn*, and *Tbx3*, which are decreased in tumor cells, hepatocytes, and immune cells, with minimal impact to HSCs and ECs.

Thus, our findings in the mouse models encapsulates holistic understanding of how precision targeting of β -catenin in both tumoral and immune compartments impacts tumor development and progression.

Point 2. The mouse model employed here is not an established HCC treatment model but rather an intervention model for cancer development. The method does not align with the intended purpose of evaluating therapeutic strategies in established HCC as the author claimed. This discrepancy undermines the study's design and its relevance to clinical HCC treatment.

Response: We respectfully disagree that this is not an established HCC treatment model. Several labs including ours have used SB-HDTVi models for HCC development. These studies have been peer reviewed, funded by NIH, funded by pharma, and have now become the basis of clinical trials. Major trunk mutations in HCC are mutations in TERT promoter, mutations in TP53, and mutations in CTNNB1 (encoding β -catenin). β -catenin mutation alone does not induce HCC in mice using HDTVi method. Common co-mutations/alterations with β -catenin include NFE2L2 (encoding Nrf2), ARID1/2, APOB, MLL2, and Met overexpression. We believe that by characterizing response in both β -catenin-Nrf2 and β -catenin-Met models, and then in aggressive β -catenin-Nrf2-Met triple HCC model at both early and advanced-disease stages demonstrates generalizability and robustness of the proposed therapeutic strategy. As mentioned in point 1, we have previously shown in 3 independent publications (Tao et al. 2016, Tao et al. 2021, and Lehrich

et al. 2024) that β -catenin-Nrf2 HCC, β -catenin-Met HCC, and β -catenin-Nrf2-Met triple HCC models have been validated for clinical relevance by examining transcriptomic analysis of these HCC models and comparing them to human HCC subsets with corresponding molecular perturbations. We have demonstrated 70-80% similarity to human HCC subsets with similar perturbations. Combined, these models represent ~30% of all HCC. In contrast, using a PDX model with patient-derived tissue would require immunodeficient mouse background, providing a limitation in terms of testing synergy of LNP-siRNA with ICI in CTNNB1-mutated HCC. There are no other alternatives for assessing role of β -catenin in HCC via mouse models that have been extensively validated as much as the SB-HDTV1 models for HCC. In summary, we have used multiple clinically relevant models of HCC to demonstrate efficacy of inhibiting β -catenin for treatment of these subsets of liver tumors.

References

1. Tao J, Xu E, Zhao Y, et al. Modeling a human hepatocellular carcinoma subset in mice through coexpression of met and point-mutant beta-catenin. *Hepatology*. 2016;64(5):1587-1605.
2. Tao J, Krutsenko Y, Moghe A, et al. Nuclear factor erythroid 2-related factor 2 and beta-Catenin Coactivation in Hepatocellular Cancer: Biological and Therapeutic Implications. *Hepatology*. 2021;74(2):741-759.
3. Lehrich BM, Tao J, Liu S, et al. Development of Mutated β -catenin Gene Signature to identify CTNNB1 mutations from whole and spatial transcriptomic data in patients with HCC. *JHEP Reports*. 2024:101186.

Point 3. The author hypothesizes that β -catenin mutations lead to immune exclusion through the inhibition of IRF2 and POU2F1, but this claim lacks clear experimental validation.

(1) Figure 5c shows heterogeneity in IRF2 and POU2F1 expression across liver cells. The manuscript does not explore whether there is a correlation between the expression of these factors and β -catenin mutations. It would be crucial to establish whether high or low expression of IRF2 and POU2F1 correlates with β -catenin mutation status, which is not clearly addressed.

Response: We have investigated this and observed that CTNNB1 mutated and wild-type patients do not have significantly different expression of IRF2/POU2F1, rather it is the target genes of these transcription factors which are differentially regulated, and which we have reported on in Fig.5c-d. We posit that β -catenin is repressing a module of transcription factors, through yet undetermined mechanisms, leading to inert immune responses in β -catenin-mutated HCC. This is similar to what we have shown with β -catenin regulation of p65 subunit of NF κ B (PMID: 28474571; 34609282). How mutated β -catenin is repressing IRF2 and POU2F1 is a topic of intense research in the lab but we do expect this to take substantial time and resources and hence would be a follow-up study.

(2) In Figure 5M, the infiltration of lymphocytes in tumors overexpressing POU2F1 appears to form clusters at certain boundaries, which seems inconsistent with the manuscript's hypothesis of throughout chemokine upregulation. This pathological structure of this location more closely resembles the gaps between different small cancer lesions during tumor formation, different from those in control groups.

Response: We believe that the tumors are forming ectopic lymphoid aggregates upon overexpression of POU2F1 through distinct cytokine/chemokine upregulation. The exact chemokines which are upregulated upon β -catenin suppression and POU2F1 activation is an active area of research in the lab but we do expect this to take substantial time and resources and hence would be a follow-up study.

(3) The manuscript has not confirmed whether POU2F1 overexpression impacts T cell function or activity at the protein level.

Response: We thank the reviewer for their comment. We have shown increases in CD4 and CD8 upon POU2F1 overexpression. We have demonstrated sufficiently that POU2F1 has an immune dependent role through anti-CD3 immune depletion studies (Fig.S20). We have also shown via RNAseq that T cell activity is modulated with POU2F1 overexpression in murine models. Moreover, we have performed flow cytometry for IRF2 overexpression demonstrating modulation of Tregs (Fig.S17).

(4) Additionally, no results regarding the overexpression of IRF2 are shown, which is essential to strengthen the claims made in the study.

Response: We kindly point to Fig.5 which demonstrates the IRF2 overexpression data.

Point 4. While RNA interference (RNAi) technology has been approved for clinical use, the lipid nanoparticle (LNP) delivery system employed in this study remains a preclinical tool and is not yet mature enough for clinical application. The lack of clinical applicability of this delivery system significantly limits the practical relevance of the findings.

Response: We respectfully disagree with the reviewer here. The studies outlined here in this manuscript have led to successful filing of IND to FDA for the LNP delivery system. This agent is now in PhI/II clinical trial evaluating LNP-CTNNB1 in HCC patients with confirmed Wnt activation in a first-in-class precision-medicine trial for HCC (NCT06600321). Thus, this study is highly clinically relevant and applicable with important translational implications. Moreover, Alnylam has utilized similar LNP delivery systems for their other clinically available drugs on the market including ONPATTRO, which is a highly analogous LNP to what is being used in our study.

Point 5. Figure 1b shows data of a human organoid model for one experiment. More replicates would be necessary to validate the findings.

Response: We have performed additional organoid experiments and quantified the efficacy of LNP-CTNNB1 in organoid cultures with sufficient replicates via organoid quantity and size.

Point by Point Responses

REVIEWER COMMENTS:

Reviewer #1 (Remarks to the Author):

Point 1: The authors have responded to all the issues raised. I have no additional comments.

Response: We thank the reviewer for their time and review of our manuscript to improve the quality.

Reviewer #2 (Remarks to the Author):

Point 1: The revised manuscript has addressed most of our questions. However, the mechanism by which CTNNB1 influences IRF2/POU2F1 has not been clearly elucidated, and the authors have chosen not to address this aspect through supplemental experiments in the current study. This likely represents the most significant limitation of this work.

Response: We have made a significant advance in this manuscript describing efficacy of innovative LNP-siRNA targeting CTNNB1 for treating subset of Wnt/ β -catenin active HCC. The impact of β -catenin on tumor cell reprogramming, metabolism and immune microenvironment has been identified using single-cell and spatial transcriptomics and validated through appropriate mechanistic studies. The role of β -catenin influencing IRF2/POU2F1 signaling remains an area of ongoing research in the lab and will be included in a follow-up manuscript.

Reviewer #3 (Remarks to the Author):

Point 1: The authors present a well-executed study employing a clinically relevant CTNNB1-mutant HCC model to investigate β -catenin targeting via siRNA encapsulated in lipid nanoparticles (synthesized from commercial company). Through multi-omics analyses, the study provides insights into the reprogramming of tumor cells and microenvironments by β -catenin and examines the impact of its inhibition on tumor biology and therapeutic strategies. However, the overall novelty and conceptual advancement of this work in both material design and target selection are very limited. The oncogenic role of CTNNB1 mutations and therapeutic potential of β -catenin inhibition in HCC have been extensively documented in prior literatures (e.g., Nat Commun. 2022; Cell Death Dis. 2023; Curr Cancer Drug Targets. 2013). While the combination strategy of β -catenin inhibition with anti-PD-L1/CTLA-4 therapies is scientifically sound, it primarily constitutes an incremental extension of existing paradigms rather than a transformative conceptual advance. The study does not provide fundamentally new biological insights or demonstrate a clear therapeutic advantage beyond what has already been reported in the context of β -catenin modulation in HCC immunotherapy. Given these considerations, the findings may be better suited for a more specialized journal rather than Nature Communications.

Response: We consider our study as a major conceptual advance in the field. Despite years of attempting to target CTNNB1 in β -catenin-mutated HCC, this is the first agent to move to a clinical trial as a precision-medicine therapeutic. This has broad implications to treat nearly 40-50% of all HCC with additional ramifications in treatments of colorectal cancer, melanoma, and endometrial cancer. The mechanistic studies on tumor cell reprogramming, regulation of immune microenvironment and eventually on tumor biology using innovative modalities, provide sufficient rationale for use of LNP-siRNA against β -catenin as a single agent or in combination with immune checkpoint inhibitor for the treatment of HCC. Additionally, the datasets generated during the study including bulk RNA-seq, single-cell RNA-seq, and spatial transcriptomics on multiple SB-HDTV_i models with β -catenin mutations, have all been made publicly available and hence a potential resource to mine for any additional discoveries.